



# Non-Redfield carbon model for the Baltic Sea (ERGOM version 1.2) – Implementation and Budget estimates

Thomas Neumann[1], Hagen Radtke[1], Bronwyn Cahill[1], and Martin Schmidt[1]

[1]Leibniz Institute for Baltic Sea Research Warnemünde, Seestr. 15, 18119 Rostock, German

**Correspondence:** Thomas Neumann (thomas.neumann@io-warnemuende.de)

**Abstract.** Redfield stoichiometry based marine biogeochemical models suffer from underestimating carbon fixation by primary production. Most pronounced indication of this is the overestimation of the dissolved inorganic carbon concentration and, consequently, the partial pressure of carbon dioxide in surface waters. The reduced production of organic carbon will impact most biogeochemical processes.

We propose a marine biogeochemical model allowing for a non-Redfield carbon fixation. The updated model is able to reproduce observed partial pressure of carbon dioxide and other variables of the ecosystem, like nutrients and oxygen, reasonably well. The additional carbon uptake is realized in the model by an extracellular release of dissolved organic matter from phytoplankton. Dissolved organic matter is subject to flocculation and the sinking particles remove carbon from surface waters. This approach is mechanistically different from existing non-Redfield

models, which allow for flexible element ratios for the living cells of the phytoplankton itself. The performance of the model is demonstrated as an example for the Baltic Sea.

Budget estimates for carbon illustrate that the Baltic Sea acts as a carbon sink. For alkalinity, the Baltic Sea is a source due to internal alkalinity generation by denitrification. Owing to the underestimated model alkalinity, there exists still an unknown alkalinity source or underestimated land based fluxes.

## 15   1   Introduction

We introduce the non-Redfieldish carbon uptake implemented in the biogeochemical model ERGOM 1.2. In a previous publication (Neumann et al., 2021), the optical model of ERGOM 1.2 is described. In this paper, we focus on the non-Redfieldish carbon uptake in ERGOM 1.2. We decided to split the description of ERGOM 1.2 into two parts because we think both parts could be used separately in other models as well.

Models for the marine carbon cycle often fail if carbon fixation by autotrophs is restricted to the elemental Redfield ratio (Redfield et al., 1963). As an example, the surface $CO_2$ partial pressure ($spCO_2$) for the Baltic Sea can hardly be represented correctly (Omstedt et al., 2014). This is especially obvious by an elevated simulated $spCO_2$ compared to observations during summer, when primary production is limited by depleted nutrients. Fransner et al. (2018) demonstrated the considerable improvement by introducing non-Redfieldish dynamics. Established methods





for implementing a non-Redfieldish carbon fixation in ecosystem models are the cell quota model by Droop (1973) and/or additional carbon uptake due to dissolved organic matter (DOM) production (Fransner et al., 2018).

DOM in the ocean is one of Earth's major carbon reservoirs (Hansell et al., 2009). Many production, degradation and consumption processes control its dynamics. An excellent review of DOM dynamics is given by Carlson and Hansell (2015). We will summarize some facts from this review which we think are important to guide model

development: Main producer of DOM is phytoplankton within the euphotic zone due to extracellular release (ER). Two common models exist to explain mechanisms for ER: (i) The overflow model and (ii) the passive diffusion model. The overflow model assumes an active DOM release by healthy cells. This process is directly coupled to primary production (PP) and regulates the frequently mismatching availability of irradiation and nutrients. The active ER will be used to dissipate energy from the photosynthetic machinery and protect it from damage. In the

passive diffusion model, ER is controlled by different concentrations of DOM inside and outside of the cell. The concentration gradient forces an ER across the cell membrane. This process is stronger coupled to phytoplankton biomass instead of primary production. For both models, experimental evidence exists and it is possible that both are valid and depending on environmental conditions, one or the other process is more active.

Although ER is coupled to PP in the overflow model, there is not a constant fraction of produced DOM. In fact,

fractionation depends on nutrient availability and phytoplankton composition (Carlson et al., 1998). Phytoplankton ER consists to up to 80% of carbohydrates which are important precursors for the formation of transparent exopolymer particles (TEP). TEP are sticky and aggregate into larger particles which may sink down (Engel et al., 2004) and are methodically often be counted as particulate organic carbon (POC) (Carlson and Hansell, 2015). Therefore, not considering TEP production results in underestimating ER (Wetz and Wheeler, 2007).

Several studies prove that the stoichiometry of healthy phytoplankton cells follow the Redfield ratio. Ho et al. (2003) show in an experimental setup for marine phytoplankton that the biomass composition is generally close to the Redfield ratio. In situ data of particulate organic matter (POM) by (Martiny et al., 2016) show only moderate deviations from Redfield ratio. Considering that POM constitutes not only of phytoplankton, other particles like heterotrophs or detritus may impact the observed ratios. Sharoni and Halevy (2020) show with the aid of model

experiments that variations in POM stoichiometry are best explained by taxonomic composition of phytoplankton compared to phenotypic plasticity. That is, phytoplankton with a minimum flexibility of the nutrient cell quota, but a variation between adapted groups, fits best the observed elemental ratio variations on a global scale. Engel (2002) state that "the fundamental need for N and P for biomass synthesis does not allow large deviations from Redfield".

Considering the fact that biogeochemical models for the Baltic Sea with a Redfieldish carbon fixation are not able

to reproduce the observed carbon cycle (see also Fig 1) and a strong observational evidence for an ER of DOM, we develop a model able to fix carbon beyond the classical Redfield ratio. In this study, we introduce a non-Redfiedish carbon uptake by maintaining Redfieldish composition of living biomass, but allowing ER of highly carbon-enriched DOM in the model ERGOM 1.2 and show selected budgets derived from the model simulations.

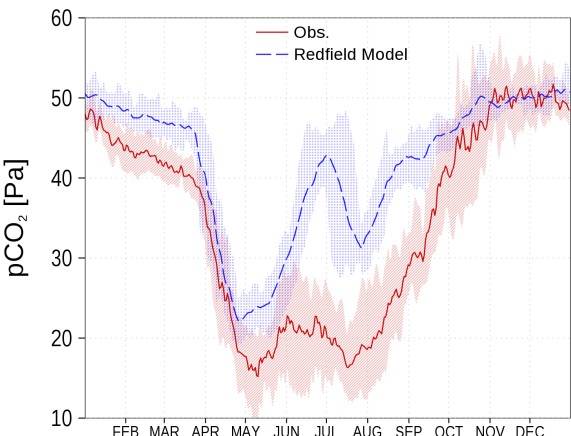

**Figure 1.** $spCO_2$ in the central Baltic Sea from a previous, Redfield stoichiometry version of ERGOM (blue) and observations (red) as climatology (2003–2016) at station BY15 (Fig.5). Shaded areas show the range between 10th and 90th percentile. Observations are available from SOCAT (see *code and data availability*).

## 2 Methods and data

### 2.1 Biogeochemical model

We start with explaining the biogeochemical model ERGOM (Leibniz Institute for Baltic Sea Research, 2015), which describes cycles of the elements nitrogen, phosphorus, carbon, oxygen, and partly sulfur.

Primary production, forced by photosynthetically active radiation (PAR), is provided by three functional phytoplankton groups (large cells, small cells, and cyanobacteria). The chlorophyll concentration used in the optical model is estimated from the phytoplankton groups (Neumann et al., 2021). Dead particles accumulate in the detritus state variable. A bulk zooplankton grazes on phytoplankton and is the highest trophic level considered in the model. Phytoplankton and detritus can sink down in the water column and accumulate in a sediment layer. In the water column and in the sediment, detritus is mineralized into dissolved inorganic nitrogen and phosphorus. Mineralization is controlled by water temperature and oxygen concentration. Oxygen is produced by primary production and consumed due to all other processes, e.g., metabolism and mineralization.

The stoichiometry in all organic carbon components of the model is confined to the classical Redfield ratio (Redfield et al., 1963). The advantage of this approach is the model's simplicity. However, observations of the carbon cycle in the Baltic Sea reveal the shortcomings of this kind of models (Fransner et al., 2018). A prominent disagreement is the overestimated $spCO_2$ in Redfield ratio based models (e.g. Kuznetsov et al., 2011). We show in Fig. 1 the $spCO_2$ climatology in the central Baltic Sea from observations and from a previous, Redfield version of the model ERGOM. There is clear observational evidence that carbon fixation continues after the depletion of nitrate during the spring





bloom period, which has been termed post-nitrate production (Schneider and Müller, 2018). As this production cannot be sustained in a strictly Redfield-defined parameterization, the simulated $spCO_2$ strongly deviates from the onset of nitrate-depletion, which in the central Gotland Sea usually starts by mid-April (Fig. 1, see also Schneider

and Müller (2018), their Figure 5.13). The $spCO_2$ overestimation vanishes in fall when primary production subsides and deeper mixing occurs. Consequently, the model primary production fixes considerably less carbon compared to *in situ* conditions. The missing organic carbon impacts all biogeochemical processes of the ecosystem. However, the relatively large freedom in calibration allows to tune the models to match observed variables like nutrient concentrations. Based on these findings, specifically the underestimation of carbon fixation, we decided to extend

our model by introducing an non-Redfield stoichiometry in carbon fixation. The aim of this extension is to not limit carbon fixation ultimately by the availability of nutrients.

Our basic idea is that the element composition in vegetative phytoplankton cells remains at the Redfield ratio and under certain circumstances, extracellular dissolved organic matter (DOM) is produced. This extracellular DOM has a fairly flexible elemental ratio. The produced DOM is subject to flocculation (TEP formation) with a certain

rate and eventually sinks down as particulate organic matter (POM). In order to realize the elemental flexibility in DOM, we introduce three different DOM state variables together with the POM counterparts. We call the DOM state variables dissolved organic carbon (DOC), dissolved organic nitrogen (DON), and dissolved organic phosphorus (DOP). The model considers DOC as polysaccharides $(COH_2)$, and DON and DOP as DOC with additional nitrogen (N) and phosphorus (P), respectively. In DON and DOP the elemental ratio is fixed to the Redfield ratio and they

are counted in units of N and P: DON - $(COH_2)_{106/16}N$ and DOP - $(COH_2)_{106}P$. Altogether, model DOM has a flexible elemental ratio with the restriction that the carbon fraction never is below the Redfield ratio. That is, DOM usually is enriched by carbon compared to the Redfield ratio. One could also have used one DOM state variable with a completely free elemental ratio. However, we used the different DOM compartments because we may consider a different fate for DOC, DON, and DOP later.

The production of DOC, DON, and DOP by phytoplankton is controlled by light availability and nutrient concentrations. Under optimal conditions, primary production increases phytoplankton biomass. When nutrients become limiting, DOM will be produced by photosynthesis instead of phytoplankton biomass. A schematic is shown in Figure 2. In case of N limitation, DOP is produced and under P limitation, DON is produced. If both N and P are limiting, DOC is produced. We have to note that only phytoplankton is able to produce DOM. That means,

if phytoplankton biomass decreases because a net growth is not possible due to e.g. nutrient limitation, the DOM production will decrease as well. In particular, the DOM production is controlled by a reversal of the phytoplankton nutrient limitation. The phytoplankton gross growths in our model is:

$$\frac{dPY}{dt} = r_0 \cdot PY \cdot \min(l_N, l_P, l_L) \cdot l_T \tag{1}$$

$PY$ is the phytoplankton biomass, $r_0$ the maximum uptake rate, and $l_n$ are limitation functions ranging between

zero and one. Subscripts $N$, $P$, and $L$ are for nitrogen, phosphorus, and light. $l_T$ is a (possible) temperature impact




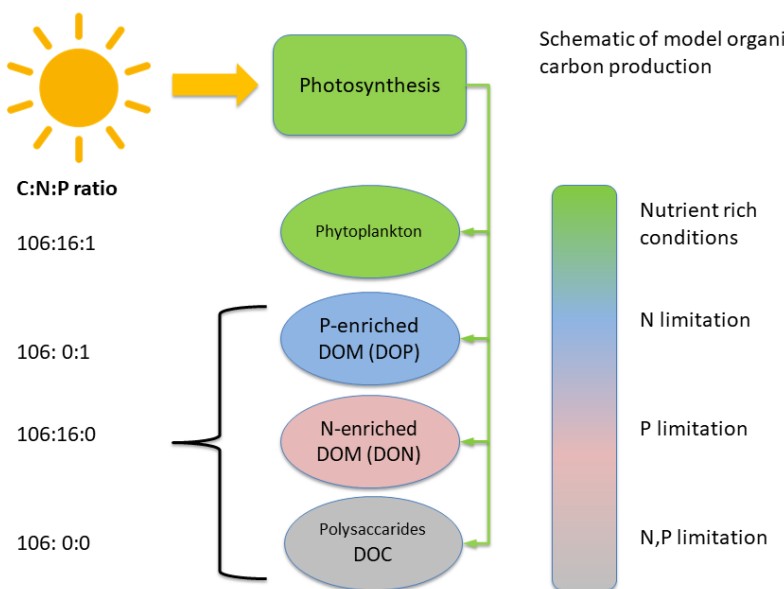

**Figure 2.** Schematic of DOM production. In case of sufficient nitrients nitrogen (N) and phosphorus (P), phytoplankton biomass is produced. If N becomes exhausted, DOP is produced and by depleated P, DON is produced. If both N and P are depleted, DOC is produced.

on uptake. For nutrient limitation $(l_N, l_P)$, we use a squared Monod kinetik (Monod, 1949; Neumann et al., 2002). Light limitation $(l_L)$ follows Steele (1974) and for temperature control $(l_T)$, a Q10 rule is applied (Eppley, 1972) meaning a doubling of rates within a 10 Kelvin temperature increase. For the temporal development of the DOM compartments we formulate:

$$\frac{\mathrm{d}DON}{\mathrm{d}t} = r_0 \cdot PY \cdot \min(1 - l_P, l_N, l_L)\, l_T \tag{2}$$

$$\frac{\mathrm{d}DOP}{\mathrm{d}t} = r_0 \cdot PY \cdot \min(l_P, 1 - l_N, l_L)\, l_T \tag{3}$$

$$\frac{\mathrm{d}DOC}{\mathrm{d}t} = r_0 \cdot PY \cdot \min(\max(1 - l_P, 1 - l_N), l_L)\, l_T \tag{4}$$

The dependence of nutrients uptake in relation to carbon uptake on nutrient concentrations is shown in Fig. 3. For this purpose, we divide the nutrient assimilation for nutrients N and P by the carbon assimilation. The assimilation consists of phytoplankton growth (Redfield ratio) and ER defined in equations 1 to 4. The nutrient concentrations are normalized by the half saturation constant from the Monod kinetics. A value of one in Fig. 3 denotes a carbon uptake in the Redfield ratio while smaller values is an excess carbon uptake. In case of low N concentrations, the



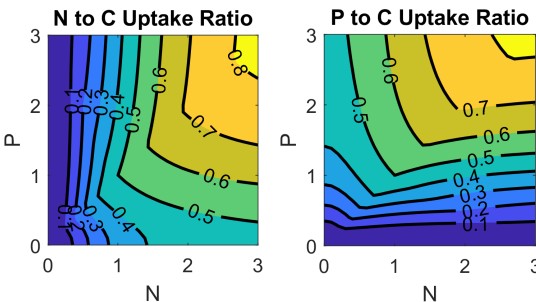

**Figure 3.** Nutrients (N, P) to carbon uptake ratios in dependence of nutrient concentrations. Nutrient concentrations are normalized by the half saturation constant in the limitation function and the uptake is normalized by the Redfield ratio. A ratio of one means uptake in the classical Redfield ratio and values less then one describe an excess carbon uptake.

N:C uptake ratio declines to zero. The P:C uptake ratio in this case depends on P concentrations and asymptotically approaches 0.5 for high P concentrations. That is, ER consists of DOC and DOP in equal shares.

Extracellular DOM eventually forms particles (POC, PON, POP) which constitutes transparent exopolymer particles (TEP). Engel (2002) shows a linear relation between dissolved inorganic carbon (DIC) uptake and TEP production implying a direct transfer from DOM to TEP. Therefore, we chose a simple rate equation for DOC flocculation:

$$\frac{\mathrm{d}POC}{\mathrm{d}t} = rf \cdot DOC \tag{5}$$

$rf$ is a constant rate for POC formation. The same equation applies for DON and DOP forming their counterparts PON and POP.

For particle sinking, we apply a Martin curve (Martin et al., 1987) which means a linear increase of the sinking speed with depth.

$$w = a \cdot z \tag{6}$$

$w$ is the sinking speed, $a$ a constant, and $z$ the depth. This approach is investigated by e.g. Kriest et al. (2012) and yields good results for the deep ocean. In the Baltic Sea application, we could improve the simulated oxygen concentrations by using the non-constant sinking speed.

A schematic of ERGOM is shown in Fig.4. Ellipses are for state variables and rectangles for processes. The complete set of equations is given in appendix B.





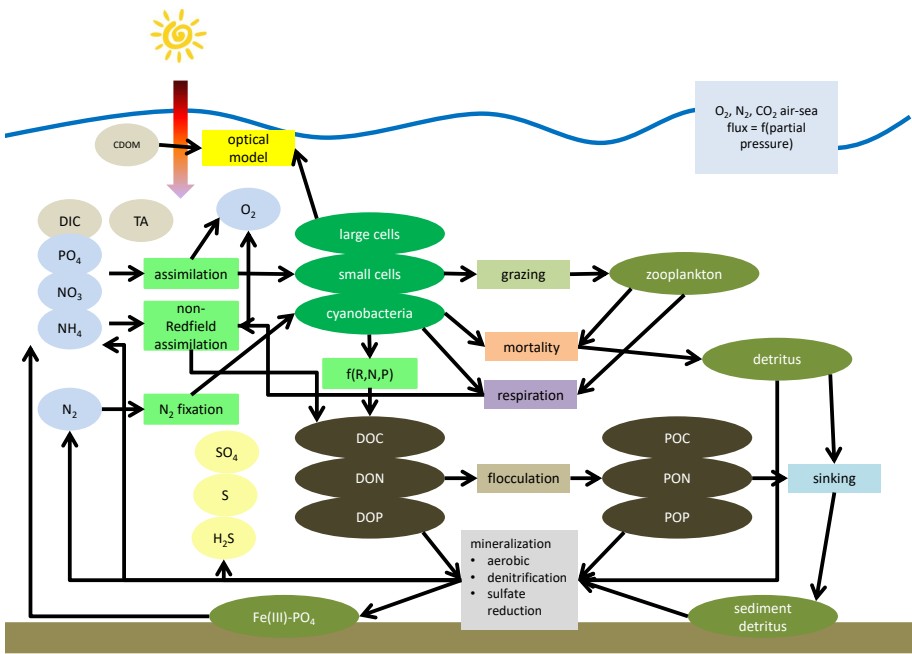

**Figure 4.** Simplified schematic of the ERGOM model. State variables are shown as ellipses and processes as rectangles. State variables are explained in Tab. 1.

The relation between model state variables and observed dissolved organic carbon ($DOC_{obs}$) in carbon units is:

$$DOC_{obs} = DOC + \frac{106}{16} DON + 116 DOP \tag{7}$$

Taking into account that model state variables DON and DOP are counted in nitrogen and phosphorus units (see Tab. 1), they correspond to the observed nitrogen and phosphorus in DOM. We have to note that our model DOM (DOC, DON, DOP) constitutes only the labile part of DOM existing in the ocean. Usually, the refractory DOM fraction, not considered in the model, is much larger than the labile fraction.

## 2.2 Model setup and simulations

For model testing, we use a coupled system of circulation and biogeochemical model similar to that in Neumann et al. (2021). The circulation model is MOM5.1 (Griffies, 2004) adapted for the Baltic Sea. The horizontal resolution is 3 nautical miles. Vertically, the model is resolved into 152 layers with a layer thickness of 0.5m at the surface and gradually increasing with depth up to 2 m. The circulation model is coupled with a sea ice model Winton (2000) accounting for ice formation and drift. The biogeochemical model ERGOM, described in Sec. 2.1, is coupled with the circulation model via the tracer module which is part of the MOM5.1 code.





**Table 1.** State variables of the biogeochemical model ERGOM shown in Fig. 4.

| Symbol | State Variable | Units <element> [mol kg$^{-1}$] |
|---|---|---|
| $O_2$ | dissolved oxygen | dioxygen |
| $N_2$ | dissolved nitrogen | dinitrogen |
| CDOM | colored dissolved organic matter | carbon |
| DIC | dissolved inorganic carbon | carbon |
| TA | total alkalinity | molar equivalent |
| $NH_4$ | ammonium | nitrogen |
| $NO_3$ | nitrate | nitrogen |
| $PO_4$ | phosphate | phosphorus |
| $SO_4$ | sulfate | sulfur |
| S | sulfur | sulfur |
| $H_2S$ | hydrogen sulfide | sulfur |
| large cells | large cell phytoplankton | nitrogen |
| small cells | small cell phytoplankton | nitrogen |
| cyanobacteria | cyanobacteria | nitrogen |
| zooplankton | bulk zooplankton | nitrogen |
| detritus | detritus | nitrogen |
| DOC | dissolved organic carbon | carbon |
| DON | DOC with additional nitrogen | nitrogen |
| DOP | DOC with additional phosphorus | phosphorus |
| POC | particulate organic carbon | carbon |
| PON | POC with additional nitrogen | nitrogen |
| POP | POC with additional phosphorus | phosphorus |
| sediment detritus | detritus accumulated in the sediment layer | nitrogen [mol m$^{-2}$] |
| Fe(III) − PO$_4$ | phosphate adsorbed to iron-3 minerals in the sediment | phosphorus [mol m$^{-2}$] |

Sediment state variable units are mol m$^{-2}$.

The code for the biogeochemical model is generated automatically. Fundamentals are a set of text files describing the biogeochemistry independently of programming language and the host system. Code templates describe physical
and numerical aspects and are specific for a certain host, e.g., a circulation model. All necessary ingredients (the code generation tool, text files, and templates for several systems) can be downloaded from Leibniz Institute for Baltic Sea Research (2015). The same technique is used for example in Neumann et al. (2021).

We run the model for about 70 years (1948–2019) after a spin-up of 50 years. The long simulation time allows for assessing the model performance under different forcing conditions, as for example the eutrophication of the Baltic
Sea in the 1970s and the nutrient load reduction beginning in 1990.





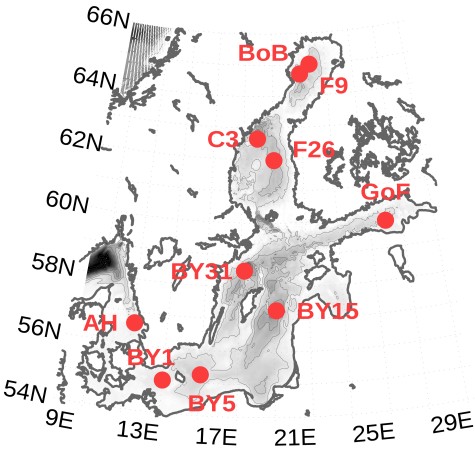

**Figure 5.** Model domain and bathymetry used for this model study. Red dots denotes stations to which we will refer later in the text. Bathymetry contour lines have a distance of 50 m. The map was created using the software package GrADS 2.1.1.b0 (http://cola.gmu.edu/grads/, last access: 14 December 2021), using published bathymetry data (Seifert et al., 2008).

**Table 2.** Average alkalinity concentration and loads in runoff for different Baltic Sea basins and from different authors. BP: Baltic Proper, GR: Gulf of Riga, GF: Gulf of Finland, BS: Bothnian Sea, BB: Bay of Bothnia. HS: Hjalmarsson et al. (2008), GS: Gustafsson et al. (2014b), NM: this study.

| Basin | Concentration | | | Load | |
|---|---|---|---|---|---|
| | HS | GS | NM | GS | NM |
| BP | 3244 | 1910 | 3156 | 203 | 340 |
| GR | 3117 | 3140 | 3638 | 92 | 117 |
| GF | 835 | 689 | 786 | 73 | 89 |
| BS | 467 | 271 | 240 | 27 | 17 |
| BB | 136 | 164 | 174 | 19 | 20 |
| total | | 904 | 1165 | 453 | 606 |

Alkalinity concentration in $\mu$mol kg$^{-1}$ and loads in Gmol a$^{-1}$ .

## 2.3 Data

The model has been forced by meteorological data from the coastDat-2 dataset (Geyer and Rockel, 2013). Nutrient loads to the Baltic Sea due to riverine discharge and atmospheric deposition have been compiled based on data from HELCOM assessments (e.g. HELCOM, 2018). Riverine alkalinity follows data provided in Hjalmarsson et al. (2008). 165 In Tab. 2, we compare riverine alkalinity concentration and loads with published data from Hjalmarsson et al. (2008)



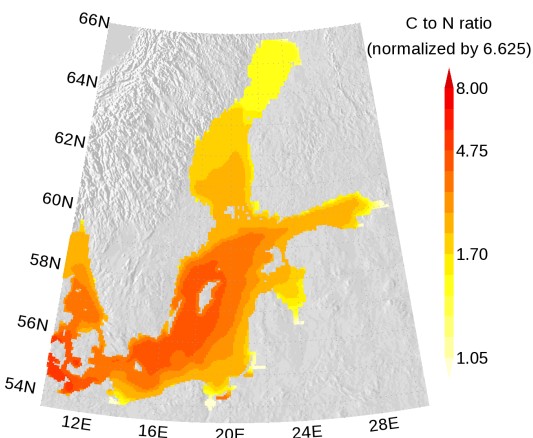

**Figure 6.** Mean elemental carbon to nitrogen ratio in surface organic matter. The ratio is normalized and a ratio of one refers to the classical Redfield ratio. The map was created using the software package GrADS 2.1.1.b0 (http://cola.gmu.edu/grads/, last access: 14 December 2021), using published topography data (Seifert et al., 2008).

and Gustafsson et al. (2014b). The data are relatively similar with the exception of the Baltic Proper. Gustafsson et al. (2014b) use considerably lower values, which impacts the total load. Our mean concentrations differ slightly form Hjalmarsson et al. (2008). We use the basin-wide and constant concentration values given in Hjalmarsson et al. (2008) and assigned the data to our model rivers which show inter annual runoff variability. This results in mean
concentration deviations. Loads given in Tab. 2 result from runoff and river specific concentrations.

    $sp$CO$_2$ for model validation have been extracted from the SOCAT (Surface Ocean CO$_2$ Atlas) data base (https://www.socat.info/). The majority of data are from the voluntary observing ship (VOS) Finnmaid between Lübeck-Travemünde and Helsinki. VOS Finnmaid is a component of the European ICOS (Integrated Carbon Observation System) research infrastructure. Data processing and quality control follow the SOCAT guidelines (Bakker et al.,
2016; Pfeil et al., 2013). Additional observation data used for comparison with model results are available from public data bases. Details are given in section *code and data availability*.

## 3    Results

### 3.1    How the non-Redfieldish approach works

In this section, we demonstrate how a non-Redfield elemental ratio in organic matter (OM) develops due to the
above described model extensions. OM involves all forms of model DOM and POM including model phytoplankton, zooplankton, and detritus. We show data averaged over the whole simulation period and seasonal climatologies. The elemental ratios are based on molar concentrations.



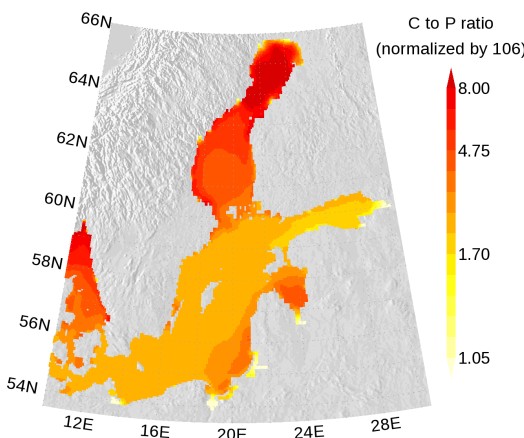

**Figure 7.** Mean elemental carbon to phosphorus ratio in surface organic matter. The ratio is normalized and a ratio of one refers to the classical Redfield ratio. The map was created using the software package GrADS 2.1.1.b0 (http://cola.gmu.edu/grads/, last access: 14 December 2021), using published topography data (Seifert et al., 2008).

In Figs. 6 and 7, the carbon (C) to nitrogen (N) and carbon to phosphorus (P) ratios in organic matter in surface water are shown. In both figures, the elemental ratios are normalized so that a ratio of one is for the classical Redfield ratio. The figures highlight the different nutrient limitation provinces in the Baltic Sea. The C:N ratio is high in the central Baltic Sea where N is a limiting nutrient and consequently the C:P ratio is low. The opposite is in the northern Baltic Sea where P is the limiting nutrient. We have to note that our model approach does not allow for C:N and C:P ratio below Redfield ratios in the DOM and POM fractions. Hence, the elemental ratios in OM are always above one. An exception are river mouths where almost no nutrient limitation keeps the C:N and C:P ratios close to one.

We show the N:P ratio in OM and its seasonality in Fig. 8. Again, the figure shows the separation between the nutrient limitation provinces. N limitation is denoted by a low N:P ratio in the central Baltic Sea and a high N:P ratio shows P limitation in the northern Baltic Sea. During the course of the year, the N:P ratio in the central Baltic Sea increases due to nitrogen fixation by cyanobacteria. The temporal development of the DOM fractions can be seen in Fig. 9. In the N-limited Gotland Sea (Fig. 9a), surplus phosphate is transferred into DOP after depleted N starts limiting phytoplankton growth. With intensified nutrient limitation also DON and DOC will be produced by phytoplankton. In summer, with a higher demand of phosphorus by cyanobacteria, the DOP pool is depleted. In contrast, in the northern part of the Baltic Sea, the Bothnian Bay, surplus nitrogen is transferred into DON (Fig. 9b). Almost no DOP develops. In Fig. 10, we show the surface climatology of simulated $DOC_{obs}$ (eq. 7) at station BY15 together with observations. Observed DOC concentrations constitute to a large extent of refractory fractions. In contrast, in the model we only consider the labile, autochthonous part of DOC. Therefore, we subtracted





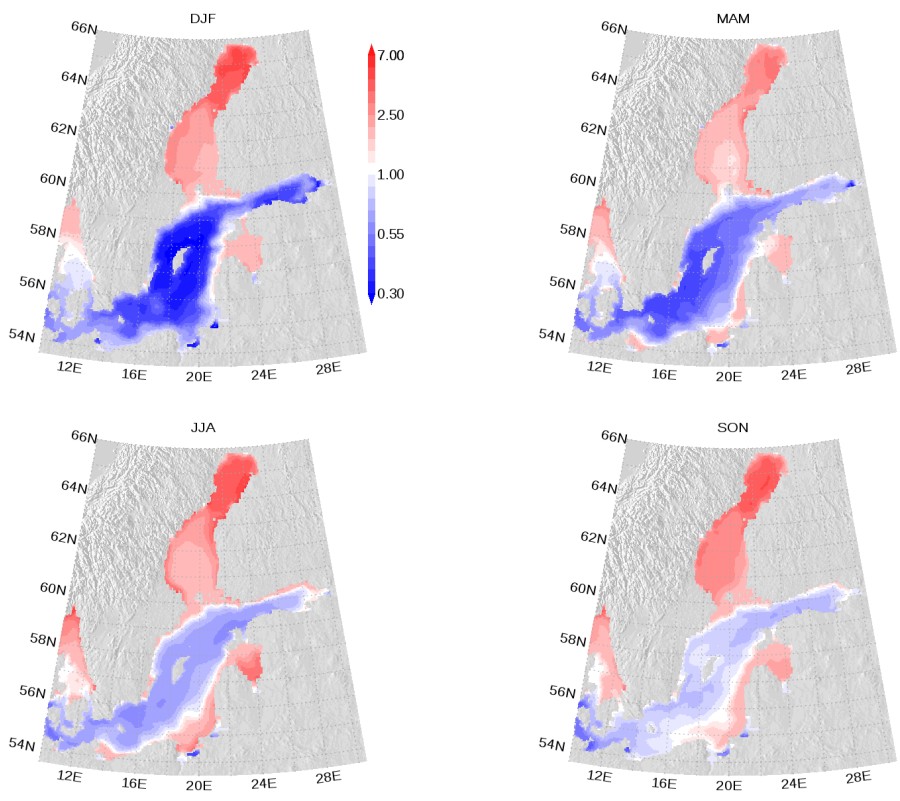

**Figure 8.** Mean elemental nitrogen to phosphorus ratio climatology in surface organic matter. The ratio is normalized and a ratio of one refers to the classical Redfield ratio. The map was created using the software package GrADS 2.1.1.b0 (http://cola.gmu.edu/grads/, last access: 14 December 2021), using published topography data (Seifert et al., 2008).

305 µmol kg$^{-1}$ from the observations which is the mean winter concentration. The annual DOC cycle in the observed data appears less pronounced compared to the modeled DOC$_{obs}$ cycle.

## 3.2 Primary production and extracellular production

We consider primary production (PP) as carbon fixation contributing to phytoplankton biomass while extracellular production (EP) is the carbon fixation resulting in DOM (DOC, DON, and DOP state variables). Figure 11 shows time series and climatology of PP and EP as mean of the whole model domain. The carbon fixation is dominated by EP. With increasing nutrient availability beginning in the 1960s, the fraction of PP increases (Fig. 11a). The PP and EP climatology in Fig. 11b shows that PP dominates in spring and fall, and EP dominates in summer. Figure 11c 210 shows PP of the model phytoplankton groups. Most PP occurs in spring mediated by the large cell phytoplankton group LPP. In contrast, the most EP is mediated by the small cell phytoplankton group SPP in summer (Fig. 11d).





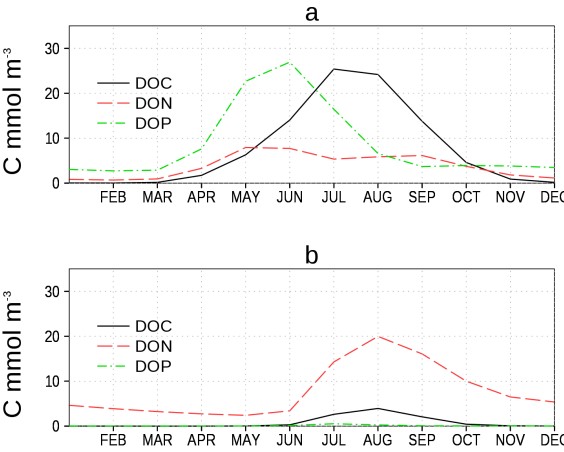

**Figure 9.** Climatology of surface model DOC, DON, and DOP (in carbon units, Tab. 1 and Eq. 7) at two stations. a) Central station in the Eastern Gotland Sea (BY15), and b) Central Station in the Bothnian Bay (BoB, Fig. 5). Model DON and DOP are converted into carbon units to show all variables in a comparable level.

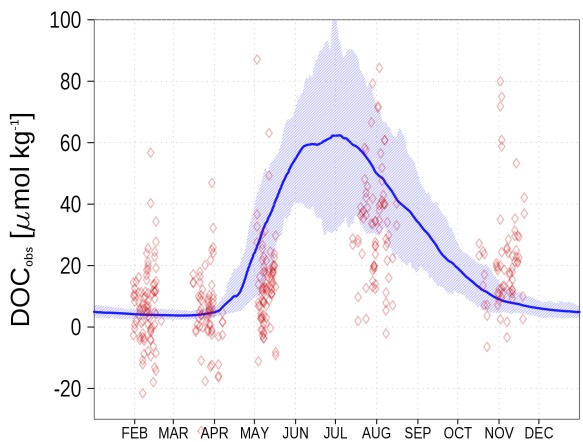

**Figure 10.** Climatology (1995-2019) of simulated surface $DOC_{obs}$ (eq. 7) at station BY15 (blue line) and observed DOC (red diamonds). The diamond's opacity reflects frequency of observations. The shaded area shows the range between 10th and 90th percentile. From observations, 305 µmol kg$^{-1}$ have been subtracted. Observed DOC data are available from IOW ODIN database (see *code and data availability*).



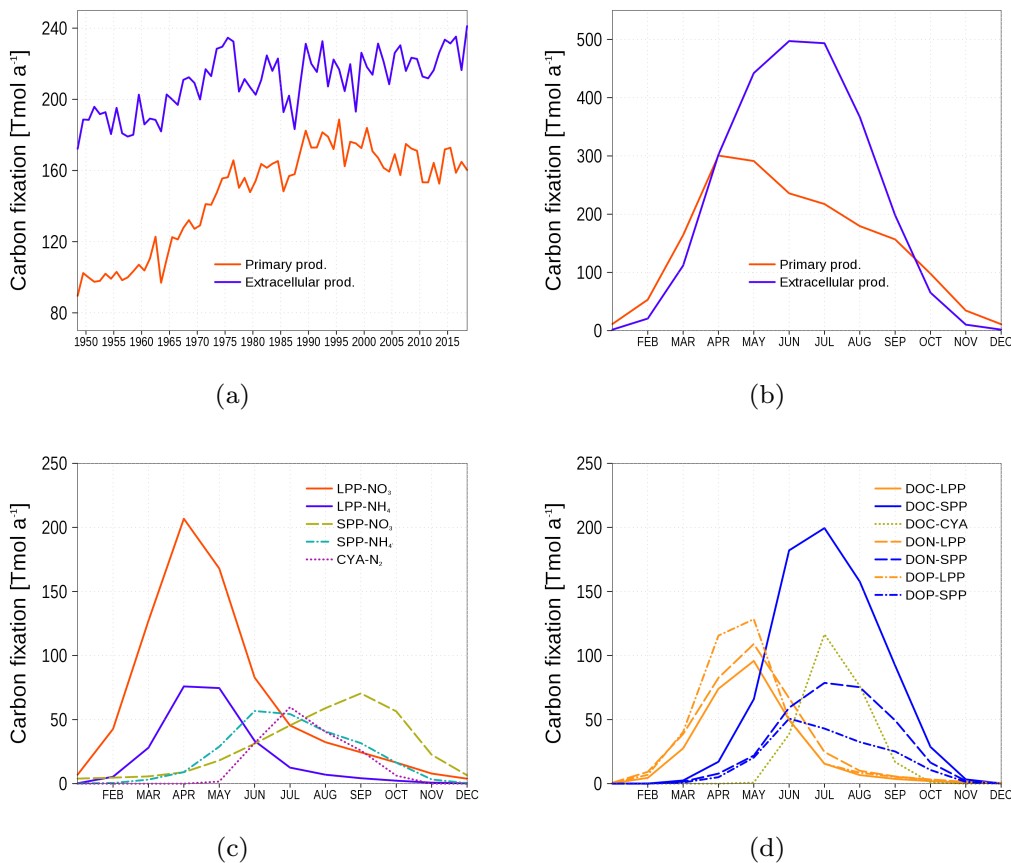

**Figure 11.** Temporal and spatial mean primary and extracelluar carbon fixation by model phytoplankton. a) Time series of annual carbon fixation. b) Climatology of carbon fixation. c) Climatology of primary production related to different uptake processes. LPP-NO3 and LPP-NH4: Carbon fixation by the large cell phytoplankton group related to NO3 and NH4 uptake, respectively. SPP-NO3 and SPP-NH4: The same as for LPP but for the small cell phytoplankton group. CYA-N: Carbon fixation by cyanobacteria related to nitrogen fixation. d) Climatology of extracellular production related to different phytoplankton groups: Red lines are uptake by LPP, blue lines by SPP, and green line by cyanobacteria. Different line styles refer DOC, DON, and DOP. All model variables have been converted into carbon units (Eq. 7).

## 3.3 Assessment of biogeochemical variables

We especially show model data and observations for sea surface carbon dioxide pressure and alkalinity. Other biogeochemical variables are shown in App. A.



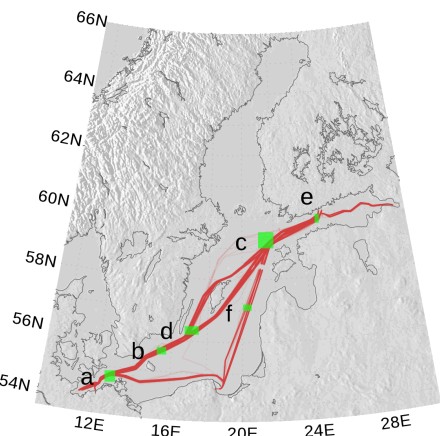

**Figure 12.** $sp\mathrm{CO_2}$ observations by VOS Finnmaid between 2003 and 2018 used for model analysis (red line). Opacity refers to frequency of observations. a–f denote regions selected (green rectangles) for comparison with model data. The map was created using the software package GrADS 2.1.1.b0 (http://cola.gmu.edu/grads/, last access: 14 December 2021), using published topography data (Seifert et al., 2008).

### 3.3.1 Sea surface pressure of carbon dioxide ($sp\mathbf{CO_2}$)

One motivation to introduce a non-Redfieldish carbon fixation into the ecosystem model ERGOM was the mismatch in observed and simulated $sp\mathrm{CO_2}$ (Kuznetsov et al., 2011, see also Fig. 1). Redfield models are not able to explain the low observed $sp\mathrm{CO_2}$ during summer. Temperature increase and ongoing mineralization in the surface layer increase the $sp\mathrm{CO_2}$ to unrealistic values in the simulations. One conclusion was that still after nutrient limitation, a substantial carbon fixation goes on. Consequently, the carbon fixation is not restricted to the classical Redfield ratio.

For the $sp\mathrm{CO_2}$ benchmark, we use data taken underway from the voluntary observing ship (VOS) Finnmaid regularly traveling between Lübeck-Travemünde and Helsinki. For more details see section 2.3 and Schneider and Müller (2018). Pathway and $sp\mathrm{CO_2}$ observations taken by VOS Finnmaid and used in this study are shown in Fig. 12. From the regions denoted by green rectangles, we have selected data to compare with our model simulation. As can be seen from the pathway's opacity, region f was crossed less frequently than the other regions. The $sp\mathrm{CO_2}$ climatology is shown in Fig. 13. The non-Redfieldish carbon fixation keeps the $sp\mathrm{CO_2}$ low during summer as seen in the observations. In the northern regions c and e, the spring bloom seems to be delayed in the model. However, the general picture is a strongly improved $sp\mathrm{CO_2}$ in the model compared to earlier model versions (e.g. Kuznetsov et al., 2011), as can be seen by comparing to Fig. 1.



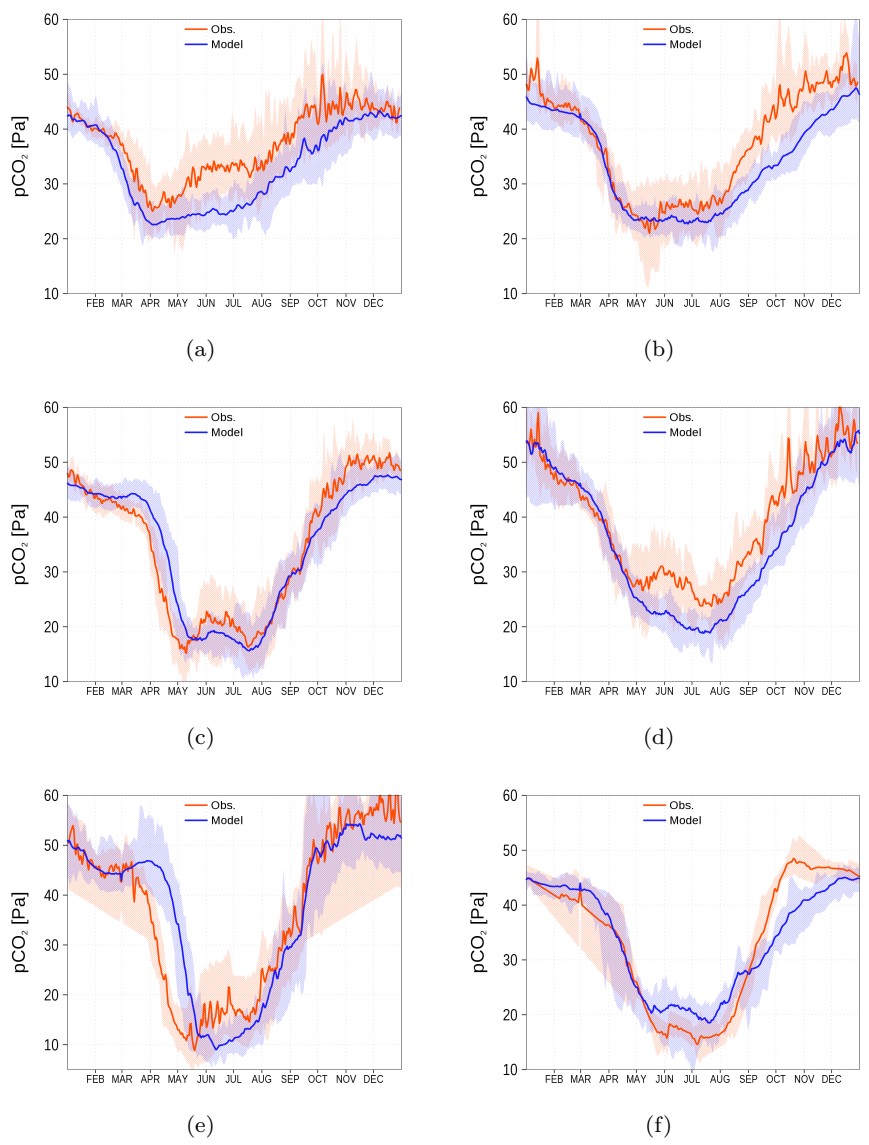

**Figure 13.** $spCO_2$ climatology (2003-2018) from observations (red) and from model simulation (blue). Shaded areas show the range between 10th and 90th percentile. The sub-figures a–f refer to the corresponding regions shown in Fig. 12 by green rectangles a–f. Observations are available from SOCAT (see *code and data availability*).

### 3.3.2 Alkalinity

Alkalinity in the model is estimated after the equation for `t_alk` in appendix B4. Figure 14 shows the alkalinity climatology from observations (red diamonds) and from the model simulation (blue). We show the climatology for





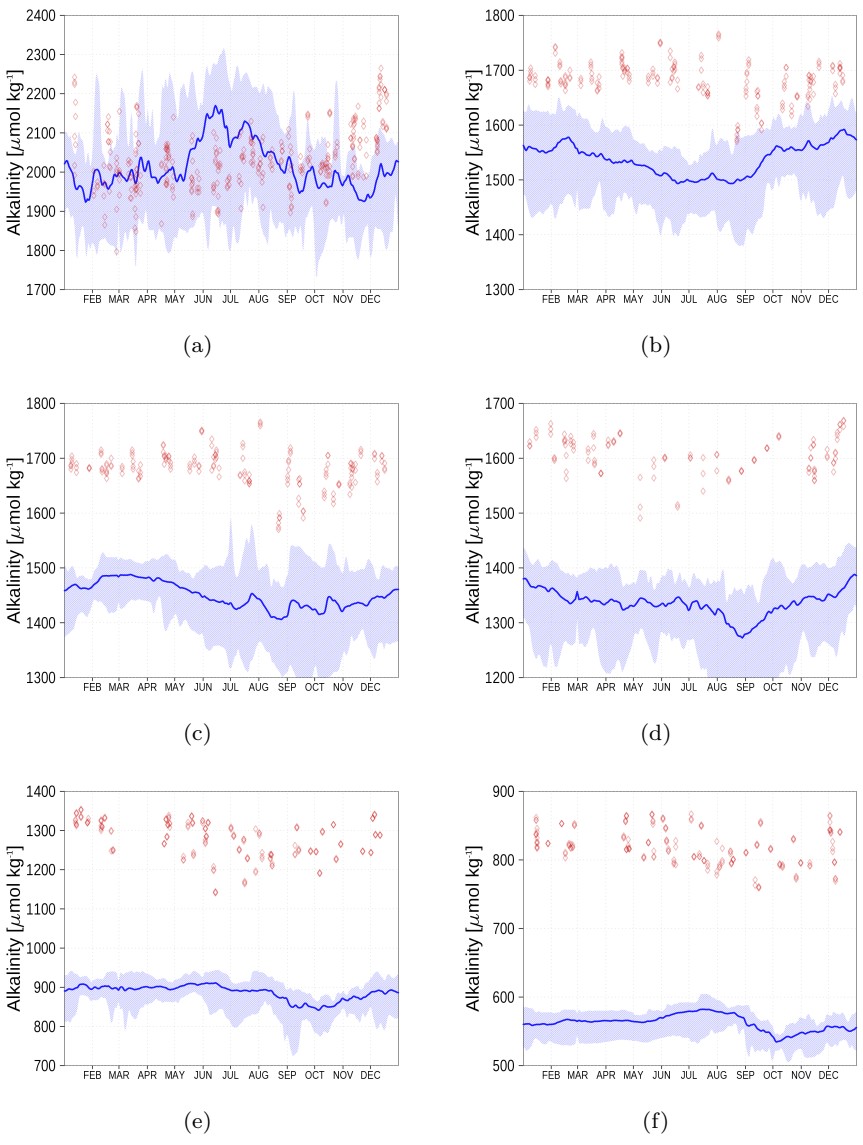

**Figure 14.** Surface alkalinity climatology (2013-2018) from Observations (red) and from model simulatuion (blue). Shaded areas show the range between 10th and 90th percentile. The subfigures represent stations AH (a), BY1 (b), BY15 (c), BY31 (d), C3 (e), and F9 (f) (Fig. 5). Observed alkalinity data are available from SHARK database (see *code and data availability*).

six stations from the Kattegat (a) to the Bothnian Bay (f). While in the Kattegat, the simulated alkalinity reflects 235 observations reasonably well, the model's underestimation amounts to roughly 20% in the central Baltic Sea and increases further towards the northern Baltic Sea. This will also have an effect on the total inorganic carbon (DIC)





content. However, once in a quasi equilibrium with the atmosphere, the air-sea fluxes will be affected marginally only.

### 3.3.3 Nutrients

Nutrient surface concentrations are shown in App. A1. We have chosen 6 stations and regions to cover the whole Baltic Sea. Figures A1–A6 show climatology and time series of simulated nitrate and phosphate together with observations. We find a good model performance for the western Baltic Sea, the central Baltic Sea, and the Gulf of Finland. In the northern Baltic Sea, the Gulf of Bothnia, the model overestimates slightly the nutrient concentrations. Nevertheless, the strong phosphate limitation in this region is well covered.

### 245 3.3.4 Oxygen

Oxygen concentrations of the near bottom water are shown in App. A2. Especially in the northern Baltic Sea, simulated concentrations are lower compared to observations.

### 3.4 Budgets

In this section, we show selected budgets as estimated from the model simulation and demonstrate that the model
closes the budget.

### 3.4.1 Carbon budget

The carbon budget is shown in Fig. 15. The budget considers the inventory change of all carbon containing state variables in the water column and in the sediment. Changes are the result of the boundary fluxes riverine load, air-sea fluxes, transport from and to the North Sea, and burial of carbon in the sediment. The closed budget, which we show
with the yellow line, should be zero, a deviation reflects cumulated numerical inaccuracies that are obviously small compared to the simulated signals. In Fig. 15a, annual fluxes and inventory changes are shown. Highest fluxes are the carbon export towards the North Sea and riverine carbon loads followed by air-sea flux and burial. Figures 15b and c show cumulated fluxes and inventory changes. Inventory changes are very small compared to the boundary fluxes. Therefore, we show in Fig. 15c the inventory changes separately. The sediment inventory stays relatively constant.
In the water column, carbon inventory increases in response to higher nutrient loads in the 1960s and 70s.

### 3.4.2 Alkalinity budget

The alkalinity budget is shown in Fig. 16. The budget considers the inventory change of the alkalinity state variable in the water column. Changes are the result of the boundary fluxes riverine load, and transport from and to the North Sea. In contrast to the carbon budget, the alkalinity budget is not closed (yellow line). The increasing sum
of fluxes, including the inventory change, suggests an internal alkalinity source. According to the implemented



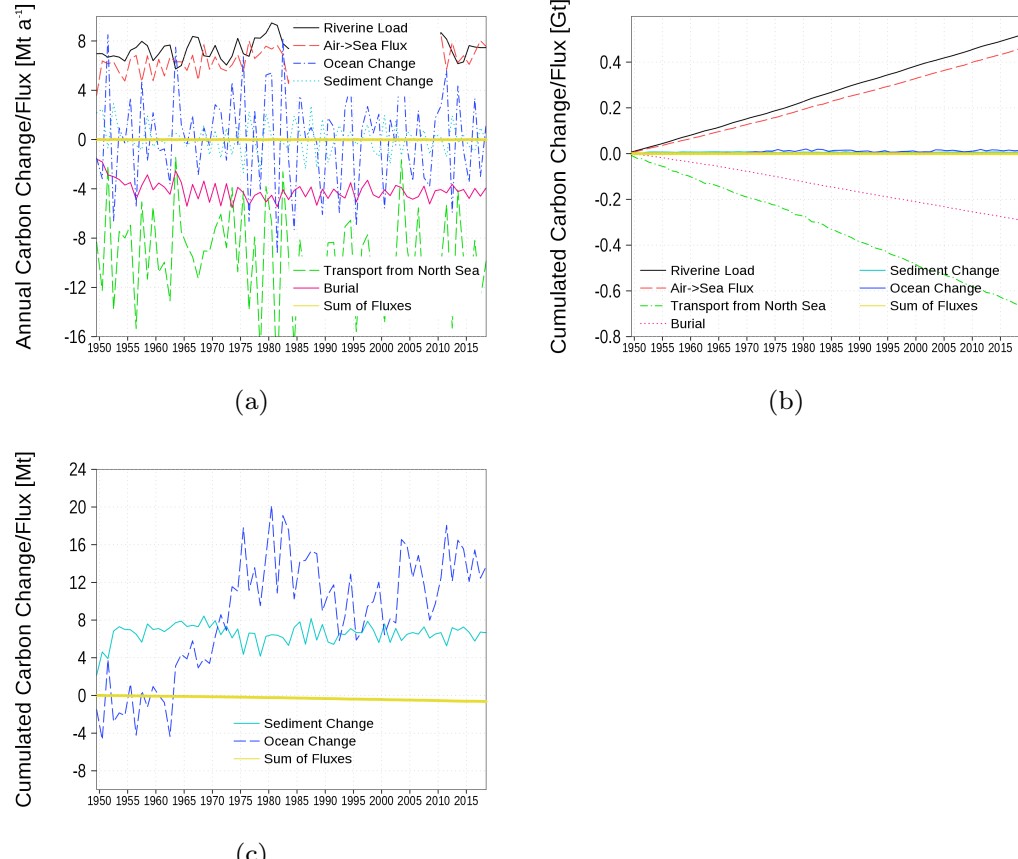

**Figure 15.** Model-domain integrated carbon budget. Shown are riverine loads, air-sea flux, burial, transport from the North Sea, and changes in the inventory of the ocean (water column) and the sediment. a) Fluxes and inventory change, b) cumulated fluxes, and c) detailed view on the cumulated inventory changes in the ocean and sediment. The yellow line is the sum of all fluxes and should be zero in a closed budget. Note: We use negative sign for sinks (burial and export towards North Sea).

processes affecting alkalinity (Eq. for `t_alk` in **??**), we attribute the alkalinity generation mainly to denitrification. The alkalinity generation estimates roughly to 7% of the loads.

### 3.4.3 Nitrogen budget

The nitrogen budget is shown in Fig. 17 with inventory changes, boundary fluxes, loads, transport from and to
the North Sea, burial in the sediment, and the internal sinks and sources denitrification and nitrogen fixation by cyanobacteria. The nitrogen load involves riverine, atmospheric, and point source loads. Strongest fluxes are due to loads as nitrogen source and sediment denitrification as sink. A detailed view on cumulated fluxes in Fig. 17c



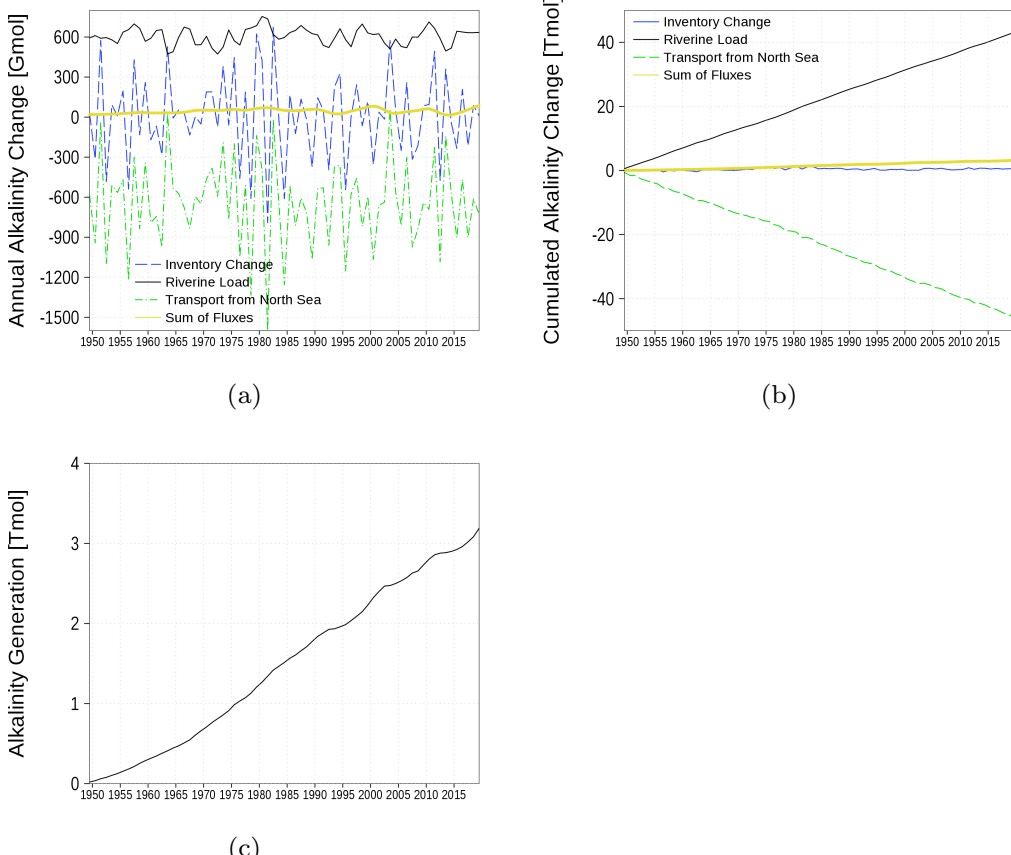

**Figure 16.** Model-domain integrated alkalinity budget. Shown are riverine loads, transport from the North Sea, and changes in the inventory of the ocean. a) Fluxes and inventory change, b) cumulated fluxes, and c) residual of the budget which can be attributed to alkalinity generation. The yellow line is the sum of all fluxes and should be zero in a closed budget. Note: We use a negative sign for sinks (export towards North Sea).

demonstrates that nitrogen fixation is nearly balanced by denitrification in the water column and only a small amount nitrogen is exported towards the North Sea.

### 3.4.4 Phosphorus budget

The phosphorus budget in Fig. 18 shows inventory changes, boundary fluxes, loads, transport from and to the North Sea, and burial. The phosphorus load involves riverine, atmospheric, and point source loads. In contrast to nitrogen, no internal sinks and sources exist. The most important sink for phosphorus loads is the burial in the sediment. Similar to nitrogen, a small amount of phosphorus is exported towards the North Sea.



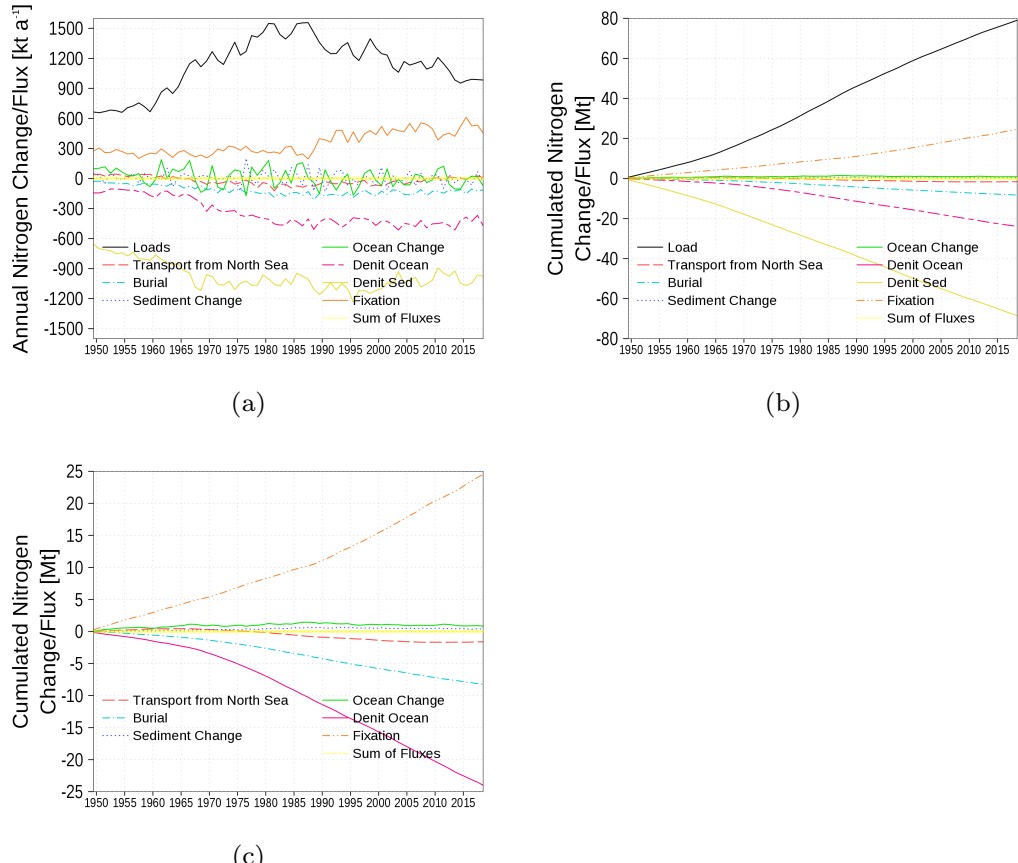

**Figure 17.** Model-domain integrated nitrogen budget. Shown are loads (riverine, atmospheric, point sources), transport from the North Sea, burial, changes in the inventory of ocean and sediment, denitrification in sediment and ocean, and nitrogen fixation. a) Fluxes and inventory change, b) cumulated fluxes, and c) detailed view without loads and sediment denitrification. The light yellow line is the sum of all fluxes and should be zero in a closed budget. Note: We use negative sign for sinks (burial, denitrification, and export towards North Sea).

## 4 Discussion and conclusion

We present a biogeochemical model for the Baltic Sea which is able to reproduce observed $sp$CO$_2$ data. This could be achieved solely by implementing a non-Redfield stoichiometry in carbon fixation. We realize this by introducing ER due to primary production. ER results in DOM with a flexible elemental ratio and eventually flocculates into POM which sinks down. This approach reproduces observed $sp$CO$_2$, nutrients, and oxygen concentrations reasonably well for the whole Baltic Sea. A different approach is used by Fransner et al. (2018). In their model, in addition to a release of DOC, phytoplankton is formulated as a quota model, that is, within the phytoplankton cells, a certain flexibility

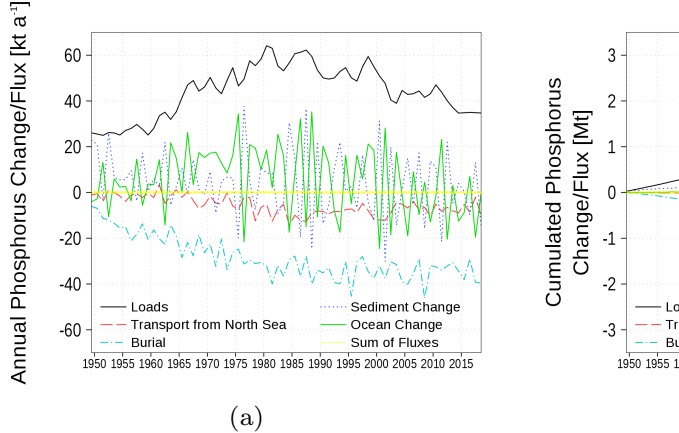
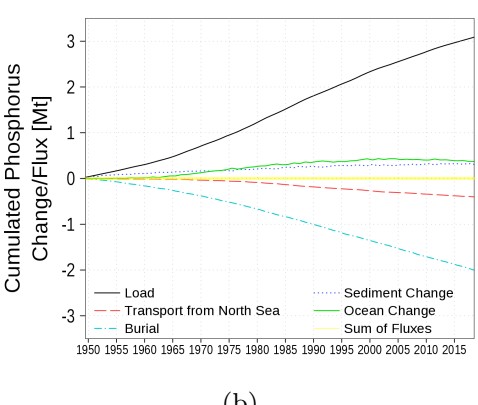

(a)  (b)

**Figure 18.** Model-domain integrated phosphorus budget. Shown are loads (riverine, atmospheric, point sources), transport from the North Sea, burial in the sediment, and changes in the inventory of ocean and sediment. a) Fluxes and inventory change, and b) cumulated fluxes. The light yellow line is the sum of all fluxes and should be zero in a closed budget. Note: We use negative sign for sinks (burial and export towards North Sea).

of the elemental ratio is allowed. This model is applied for the northern part of the Baltic Sea and reproduces well $sp$CO$_2$ and surface nutrient concentrations. The main difference besides the quota approach is that DOM in our model shows a flexible elemental ratio. This might correspond to the quota flexibility in phytoplankton cells.
However, we have chosen the fixed ratio (Redfield ratio) in healthy phytoplankton cells because of some evidence from literature (Sec: 1). We are also convinced that our approach is simpler to handle with respect to higher trophic levels which can rely on a fixed stoichiometry.

A similar model was introduced by Gustafsson et al. (2014a) also using the ER process to increase carbon fixation beyond the Redfield ratio. However, the authors do not show the model's performance with respect to $sp$CO$_2$ which
might be due to missing or rare observations during this time. Macias et al. (2019) implemented a non-Redfield nutrient uptake in an ecosystem model for the Mediterranean Sea which results in a fairly flexible elemental ratio in phytoplankton. This model gives good results for nutrients N and P but does not consider carbon. A cell quota model for global Earth system models is proposed by Pahlow et al. (2020); Chien et al. (2020). Also this model shows an advantage over fixed elemental ratio models with respect to nutrient concentration. However, a proof
against variables of the carbon cycle is unfortunately missing.

First evaluations of the simulation show an alkalinity generation of about 50 Gmol a$^{-1}$ (Fig. 16). Gustafsson et al. (2014b, 2019) estimated an alkalinity generation of 84 Gmol a$^{-1}$ and 120 Gmol a$^{-1}$, respectively. Alkalinity river loads in our model are 600 Gmol a$^{-1}$ and higher compared to loads in Gustafsson et al. (2014b, 2019) (470 Gmol a$^{-1}$, Tab. 2). Altogether, both models underestimate the alkalinity concentration (Fig. 14) and consequently, sources of
alkalinity are missing or underrepresented. Gustafsson et al. (2014b) investigate the contribution of a final pyrite



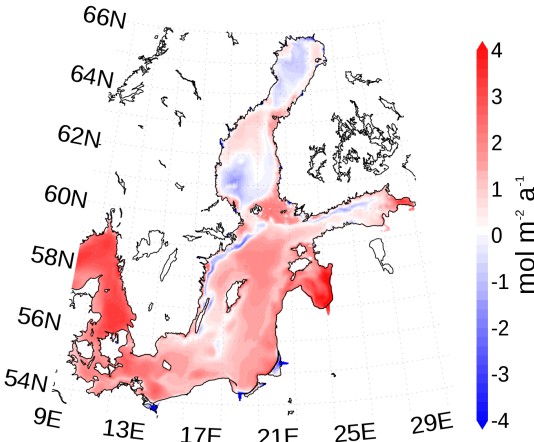

**Figure 19.** Mean atmosphere-ocean carbon dioxide flux. A positive flux is into the Baltic Sea. The map was created using the software package GrADS 2.1.1.b0 (http://cola.gmu.edu/grads/, last access: 14 December 2021).

burial in sediments to the missing alkalinity source with an advanced sediment model. However, pyrite burial can explain the missing source only partly. It remains still an open question whether riverine alkalinity loads are underestimated or an unknown source exists, e.g. groundwater discharge.

The Baltic Sea acts as a sink for carbon due to uptake of atmospheric carbon dioxide. The additional carbon is
310 partly buried and the remaining fraction is exported towards the North Sea (Fig. 15). However, the northern Baltic Sea emits carbon dioxide into the atmosphere. Figure 19 shows the horizontal pattern of the mean atmosphere-ocean flux. Sources of carbon dioxide for the atmosphere are the northern Baltic Sea and upwelling regions. The latter are caused by prevailing westerly winds with upwelling near the Swedish coast and in the Gulf of Finland. The upwelled, carbon dioxide rich deep water eventually comes in contact with the atmosphere and equilibriates by outgassing
of carbon dioxide. For the northern Baltic Sea, we hypothesize that low primary production due to low phosphate concentrations (Fig. A5) favors outgassing of carbon dioxide, which may be imported in subsurface waters.

We compare our carbon budget with estimates from Gustafsson et al. (2017) in Table 3. The most pronounced difference is the 4-fold burial of carbon in our estimates. It corresponds to a rate of 9 g m$^{-2}$ a$^{-1}$. Leipe et al. (2010, Fig. 7) estimate an observation based carbon burial rate which is similar to our rate. However, uncertainties in such
rates are large, specifically due to a strong spatial heterogeneity of the carbon burial.

Observations of the marine carbon cycle and especially the $sp$CO$_2$ provide an additional, independent state variable constraining ecosystem models. Therefore, models able to reproduce the carbon cycle in addition to e.g. nitrogen and phosphorus cycle should be more robust against changes in the forcing conditions (higher predictive capacity). This is especially important if the models will be used for projections or scenario simulations with changing forcing.





**Table 3.** Total carbon budget for the whole model domain (NM) compared with estimates from Gustafsson et al. (2017, Tab. 6) (GS).

|  | GS | NM |
|---|---|---|
| Riverine loads | 10646 | 7391 |
| Air-sea flux | 3878 | 6525 |
| Export | 13416 | 9614 |
| Burial | 909 | 4077 |

All carbon fluxes in kt a$^{-1}$ .

As a lot of observational effort in the past focused on N- and P-cycling, proper implementation of the carbon system requires additional observational and experimental data addressing the carbon cycle. For instance, the reason for the mismatch between observational and experimental alkalinity inventories needs to be addressed by re-addressing the alkalinity flux from the riverine input. Clear evidence has been provided for trends of increasing alkalinity in the major basins of the Baltic Sea (Müller et al., 2016) particularly pronounced in the Northern Basins, but a concerted effort to better constrain the alkalinity fluxes from the major riverine sources is currently lacking. Additional contributions from groundwater seepage can contribute to the alkalinity flux from land and have been shown to locally enhance alkalinity, but the importance on a basin-wide scale is unclear (e.g. Szymczycha et al., 2014).

The initial observational finding that the carbon loss during the spring bloom continues after nitrogen depletion had originally let to the hypothesis of N-fixation already in late-April (Schneider et al., 2009), an interpretation which has been revoked by the authors due to a lack of evidence of any known N-fixing organisms during that time of the year (Schneider and Müller, 2018). However, statistical analysis of observational data clearly revealed an increase in total N in the surface waters of the central Baltic Sea during this period (Eggert and Schneider, 2015), which would not be reproduced by our model. The authors speculated on a potential vertical shuttling of nitrate by the mixotroph mesodynium rubrum, a theory later supported by observations in the Gulf of Finland (Lips and Lips, 2017). Recently, anomalous high carbon fixation in the surface layer under extreme sunny and calm spring conditions in 2018 have been also linked to potential vertical nutrient shuttling (Rehder et al., 2020). However, studies on a process level are needed to explore the mechanism and quantity of a potential nutrient shuttle.

Finally, we present a biogeochemical model for the Baltic Sea reproducing parts of the nutrients and carbon cycle reasonable well. This progress allows now for numerical quantitative studies especially with focus on carbon dynamics in the Baltic Sea under different forcing conditions.



*Code and data availability.* *sp*$CO_2$ data used are available from https://www.socat.info (last access: 14 January 2022). Oceanographic data nutrients and oxygen used for model validation are available from https://www.ices.dk/data/data-portals/Pages/default.aspx (last access: 14 January 2022). DOC data used are available from IOW database ODIN https://odin2.
io-warnemuende.de/ (last access:18 February 2022). Alkalinity data used are available from SHARK database https://sharkweb.smhi.se/hamta-data/ (last access: 28 February 2022). The meteorological forcing is archived at https://doi.org/10.1594/WDCC/coastDat-2_COSMO-CLM (last access: 14 January 2022, Geyer and Rockel (2013)).

The code of the biogeochemical model is available at https://ergom.net/ (last access: 14 January 2022). The ocean model "Modular Ocean Model MOM 5-1", used in this study, is available from the developers respository https://github.com/
mom-ocean/MOM5 (last access: 14 January 2022).

Model data can be accessed via https://thredds-iow.io-warnemuende.de/thredds/catalogs/projects/integral/catalog_pocNP_V04R25_3nm_agg_time.html (last access: 14 January 2022, Neumann (2021)). All data used in this study for analysis and figures are archived on Zenodo at https://doi.org/10.5281/zenodo.6560174 (last access: 10 March 2022, Neumann (2022)).

The version of the model code used to produce the results in this study is archived on Zenodo at https://doi.org/10.5281/
zenodo.6560174 (last access: 10 March 2022, Neumann (2022)). In addition to the source code, the archive includes initial fields and boundary conditions except the meteorological forcing.





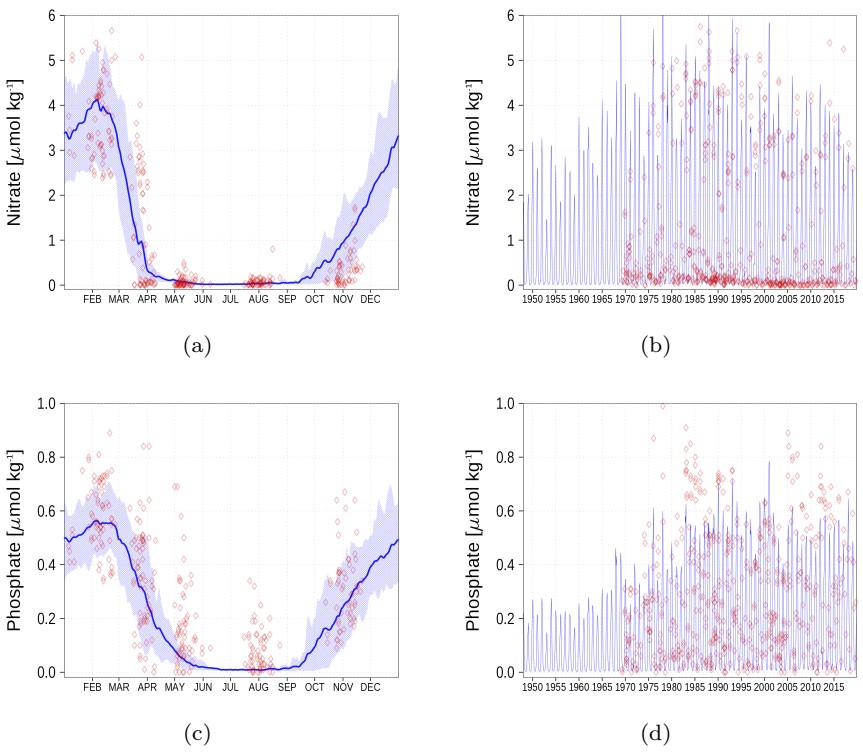

**Figure A1.** Surface nutrients concentrations at station BY1 (Fig. 5). Blue color are model simulations and observations are shown as red diamonds. The blue shaded area is the range between 10th and 90th percentile. Opacity of the red diamonds reflects the frequency of observations. a: Nitrate climatology, b: Nitrate time series, c: Phosphate climatology, d: Phosphate time series.

## Appendix A: Model performance

In this section, we compare model results with observations in order to verify the model performance for biogeochemical variables.

### A1 Surface nutrients concentrations

We demonstrate the model performance for surface nutrients at 6 stations and regions, respectively in Figs. A1–A6. For the climatology, we have chosen the time range 1990 until 2018 since observations for some stations are sparse for the period before 1990. Data for nutrients and oxygen have been extracted from the ICES database (see *code and data availability*).



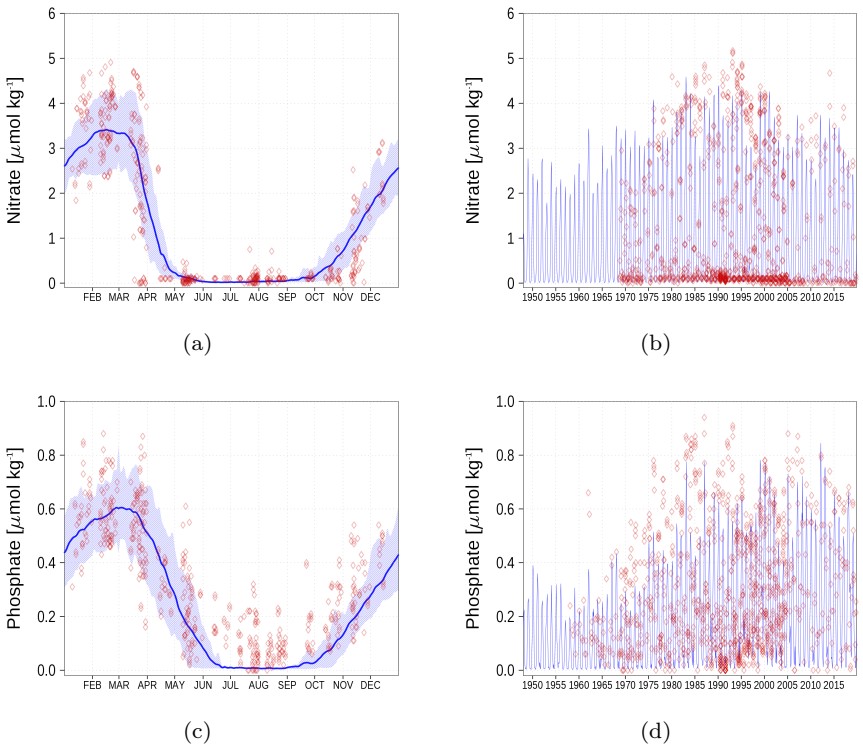

**Figure A2.** Surface nutrients concentrations at station BY5 (Fig. 5). Blue color are model simulations and observations are shown as red diamonds. The blue shaded area is the range between 10th and 90th percentile. Opacity of the red diamonds reflects the frequency of observations. a: Nitrate climatology, b: Nitrate time series, c: Phosphate climatology, d: Phosphate time series.

## A2 Oxygen

In Fig. A7, we show oxygen concentration close to the sea floor at six different stations together with observations. Hydrogen sulfide is represented as negative oxygen equivalents. The simulated oxygen concentration follows reasonable the observations. An exception is the underestimation in the Gulf of Bothnia (Fig. A7d and e). Beginning in 1970, the simulated values start to deviate from the field data.



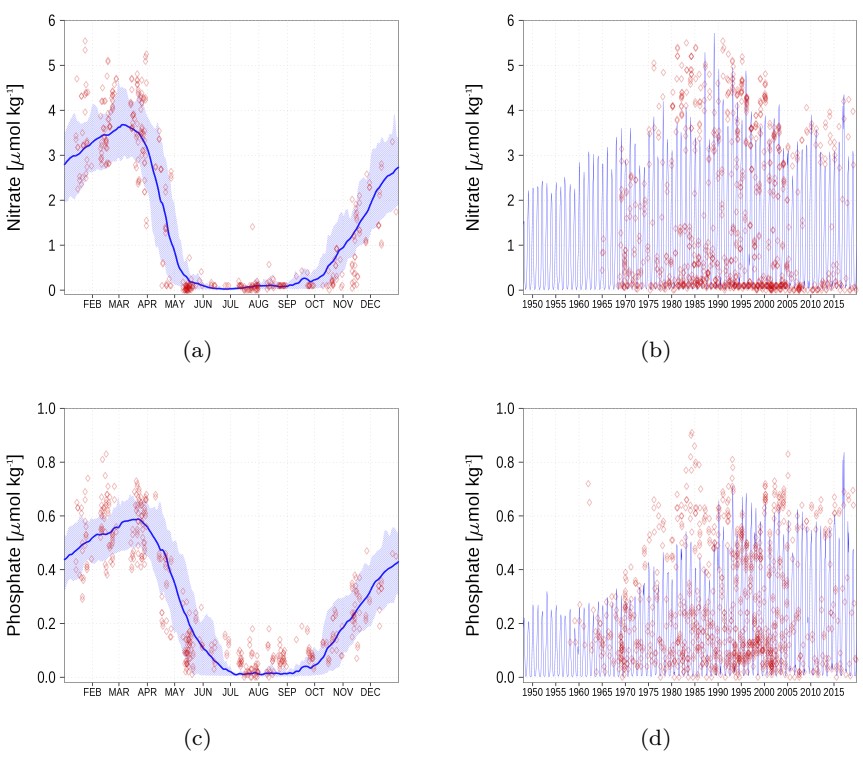

**Figure A3.** Surface nutrients concentrations at station BY15 (Fig. 5). Blue color are model simulations and observations are shown as red diamonds. The blue shaded area is the range between 10th and 90th percentile. Opacity of the red diamonds reflects the frequency of observations. a: Nitrate climatology, b: Nitrate time series, c: Phosphate climatology, d: Phosphate time series.



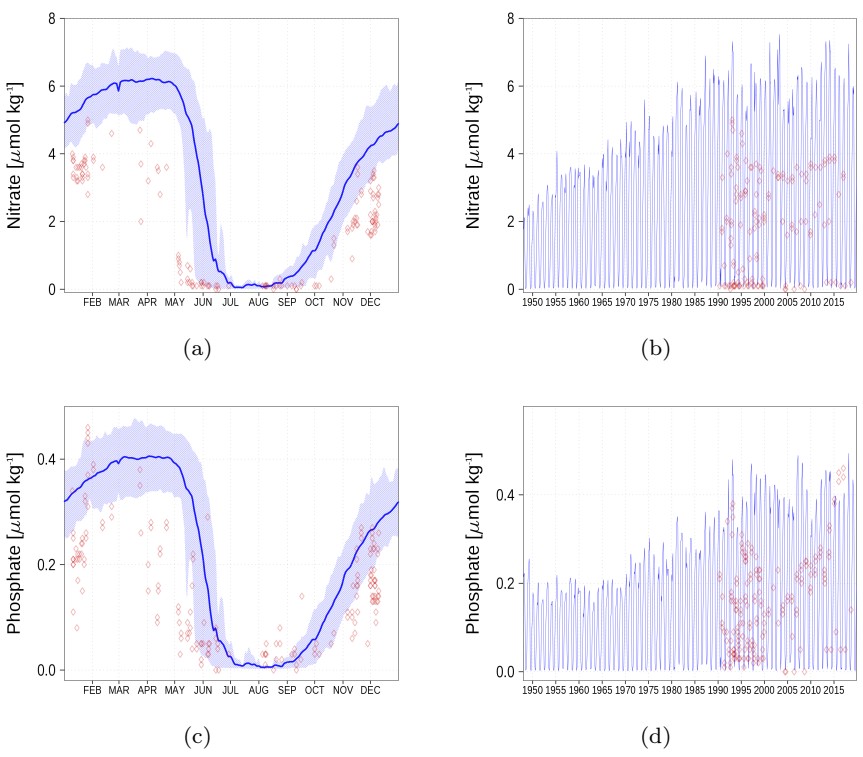

**Figure A4.** Surface nutrients concentrations at station F26 (Fig. 5). Blue color are model simulations and observations are shown as red diamonds. The blue shaded area is the range between 10th and 90th percentile. Opacity of the red diamonds reflects the frequency of observations. a: Nitrate climatology, b: Nitrate time series, c: Phosphate climatology, d: Phosphate time series.





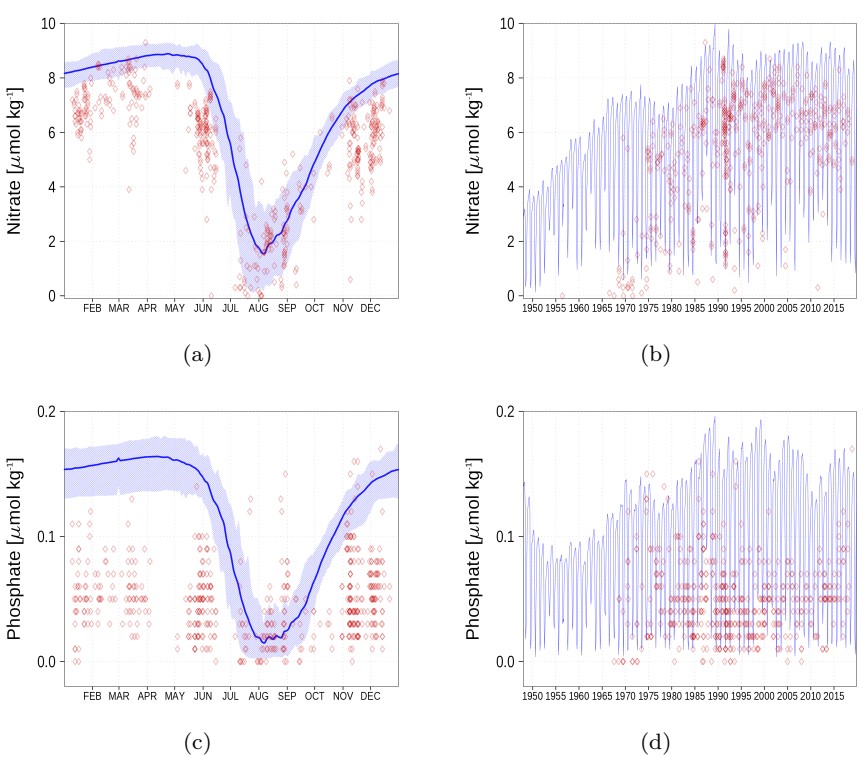

**Figure A5.** Surface nutrients concentrations in the Bothnian Bay (BoB, Fig. 5). Blue color are model simulations and observations are shown as red diamonds. The blue shaded area is the range between 10th and 90th percentile. Opacity of the red diamonds reflects the frequency of observations. a: Nitrate climatology, b: Nitrate time series, c: Phosphate climatology, d: Phosphate time series.





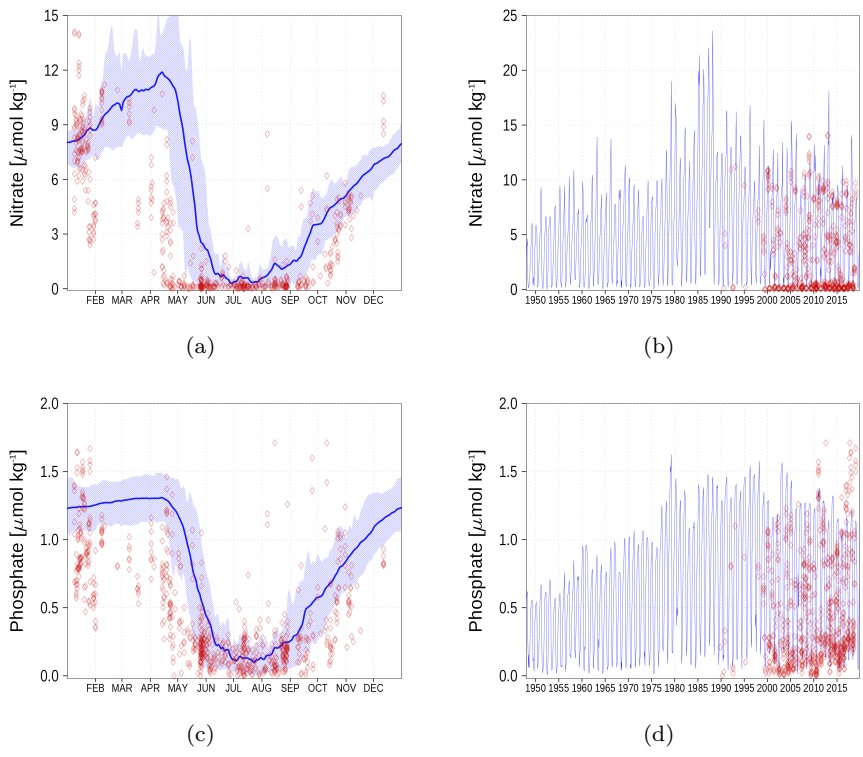

**Figure A6.** Surface nutrients concentrations at station Gulf of Finland (GoF, Fig. 5). Blue color are model simulations and observations are shown as red diamonds. The blue shaded area is the range between 10th and 90th percentile. Opacity of the red diamonds reflects the frequency of observations. a: Nitrate climatology, b: Nitrate time series, c: Phosphate climatology, d: Phosphate time series.



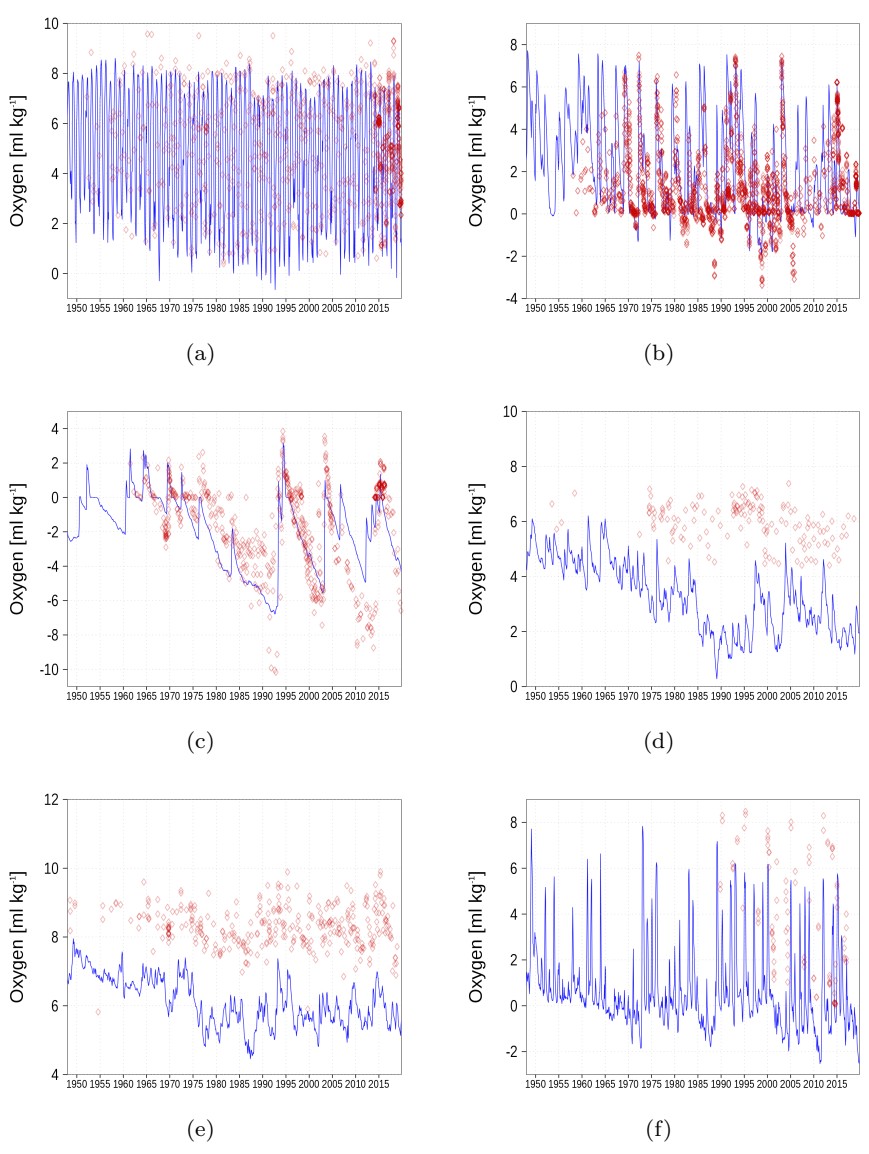

**Figure A7.** Bottom oxygen concentration at six stations in the Baltic Sea. Negative values denote the presence of hydrogen sulfide. Blue color are model simulations and observations are shown as red diamonds. Opacity of the red diamonds reflects the frequency of observations. a: BY1,b: BY5, c: BY15, d: F26, e: BoB, f: GoF (Fig. 5).



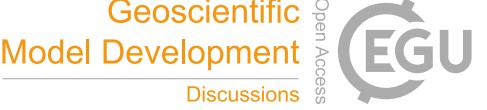

**Appendix B: ERGOM Documentation**

# Contents





## B1  Introduction

This is an automatically generated description of the ecosystem model ERGOM  version CDOM 1.2 . Model formulation is provided by text files in compliance with the rules of the Code Generation Tool (CGT) by Hagen Radtke (see www.ergom.net).

The ecosystem state variables are concentrations of several substances and are called tracers. In the host ocean model they undergo physical advection, turbulent diffusion or vertical motion as sinking or rising. The ecosystem model component defines their sources or sinks from element turnover through the ecosystem. They are defined and described in Sec. B2.

The following Sec. B3 is the main part of this model description document. It describes the processes changing
the tracer concentrations over time. Analogously to chemical processes, two components describe a process:

- A process equation which describes the transformation from precursors (on the left-hand side) to products (on the right-hand side), and

- a turnover rate, describing how fast the process runs.

The time tendency of a tracer can then easily be determined by multiplying the process turnover rate with the
stoichiometric ratio in which it consumes or produces the tracer according to the reaction equation.

The document structure reflects the different process types. All processes of one type (e.g. phytoplankton assimilation) are listed together with all their constants and auxiliary variables they depend on. For readability, some constants, such as stoichiometric ratios, will occur repeatedly. We take this compromise for the sake of readability, keeping all information required to understand a specific process in its own section.

For completeness, the tracer equations are given in Sec B4. However, we consider this as a supplementary chapter and suggest to study the model details from Sec. B3 instead.

## B2  Description of model state variables (tracers)





| Tracers in the water column only | |
|---|---|
| t_n2 | **dissolved molecular nitrogen (mol/kg)** |
| t_o2 | **dissolved oxygen (mol/kg)** |
| t_dic | **dissolved inorganic carbon, treated as carbon dioxide (mol/kg)** |
| t_nh4 | **ammonium (mol/kg)** |
| t_no3 | **nitrate (mol/kg)** |
| t_po4 | **phosphate (mol/kg)** |
| t_spp<br>opacity = | **small-cell phytoplankton (mol/kg)**<br>58.0 m$^2$/mol |
| t_zoo | **zooplankton (mol/kg)** |
| t_h2s | **hydrogen sulfide (mol/kg)** |
| t_sul | **sulfur (mol/kg)** |
| t_alk | **total alkalinity (mol/kg)** |
| t_lip<br>opacity = | **limnic phytoplankton (mol/kg)**<br>58.0 m$^2$/mol |
| t_doc | **dissolved organic carbon (mol/kg)** |
| t_dop | **phosphorus in dissolved organic carbon in Redfield ratio (mol/kg)** |
| t_don<br>opacity = | **nitrogen in dissolved organic carbon in Redfield ratio (mol/kg)**<br>12.6 m$^2$/mol |





---

Tracers in the water column only, continued from previous page

---

**t_cdom**      **colored dissolved organic carbon (mol/kg)**

**t_lpp**      **large-cell phytoplankton (mol/kg)**
vertical speed =      -0.5 m/day
opacity =      58.0 m$^2$/mol

**t_ipw**      **suspended iron phosphate (mol/kg)**
vertical speed =      -1.0 m/day

**t_cya**      **diazotroph cyanobacteria (mol/kg)**
vertical speed =      1.0 m/day
opacity =      58.0 m$^2$/mol

**t_det**      **detritus (mol/kg)**
vertical speed =      -4.5 m/day
opacity =      53.2 m$^2$/mol

**t_poc**      **particulate organic carbon (mol/kg)**
vertical speed =      w_poc_var m/day

**t_pocp**      **phosphorus in particulate organic carbon in Redfield ratio (mol/kg)**
vertical speed =      -0.1 m/day

**t_pocn**      **nitrogen in particulate organic carbon in Redfield ratio (mol/kg)**
vertical speed =      -0.1 m/day

---

end of table **Tracers in the water column only**

---

**Tracers in water and pore water**

---

end of table **Tracers in water and pore water**

---





| Tracers in fluff and sediment |
|---|
| t_sed          sediment detritus (mol/m$^2$) |
| t_ips          iron phosphate in sediment (mol/m$^2$) |
| t_sed_poc       sediment particular carbon (mol/m$^2$) |
| t_sed_pocn     sediment particular organic N+C (mol/m$^2$) |
| t_sed_pocp     sediment particular organic P+C (mol/m$^2$) |
| end of table **Tracers in fluff and sediment** |

## B3    Description of model processes, ordered by process type

### B3.1    Process type BGC/benthic/bioresuspension

| Processes |
|---|
| **bio resuspension of sedimentary detritus (sediment only)** [mol/m$^2$/day] |
| t_sed -> t_det |
| p_sed_biores_det =    (r_biores*exp(-0.02*cgt_bottomdepth)*sed_active)*lim_t_o2_6* <br>                   lim_t_sed_21 |
| **bio resuspension of iron PO4 (sediment only)** [mol/m$^2$/day] |
| t_ips -> t_ipw |
| p_ips_biores_ipw =    (r_biores*exp(-0.02*cgt_bottomdepth)*t_ips)*lim_t_o2_6* <br>                   lim_t_ips_23 |
| **bio resuspension of sedimentary poc (sediment only)** [mol/m$^2$/day] |
| t_sed_poc -> t_poc |
| continued on next page... |





---

Processes, continued from previous page

---

```
p_sed_biores_poc =    (r_biores*exp(-0.02*cgt_bottomdepth)*poc_active)*lim_t_o2_6*
                      lim_t_sed_poc_22
```

**bio resuspension of sedimentary pocn (sediment only) [mol/m$^2$/day]**

```
t_sed_pocn -> t_pocn
p_sed_biores_pocn =   (r_biores*exp(-0.02*cgt_bottomdepth)*pocn_active)*lim_t_o2_6*
                      lim_t_sed_pocn_27
```

**bio resuspension of sedimentary pocp (sediment only) [mol/m$^2$/day]**

```
t_sed_pocp -> t_pocp
p_sed_biores_pocp =   (r_biores*exp(-0.02*cgt_bottomdepth)*pocp_active)*lim_t_o2_6*
                      lim_t_sed_pocp_28
```

---

end of table **Processes**

---

---

**Auxiliary variables**

---

**total carbon in sediment layer [mol/m\*\*2]**

```
sed_tot =             t_sed*rfr_c + t_sed_poc + t_sed_pocn*rfr_c + t_sed_pocp*rfr_cp
```

**total carbon in active sediment layer [mol/m\*\*2]**

```
sed_tot_active =      max(0.0,min(sed_tot,sed_max*rfr_c))
```

**detritus in active sediment layer [mol/m\*\*2]**

```
sed_active =          sed_tot_active * t_sed/sed_tot
```

**poc in active sediment layer [mol/m\*\*2]**

```
poc_active =          sed_tot_active * t_sed_poc/sed_tot
```

**pocn in active sediment layer [mol/m\*\*2]**

```
pocn_active =         sed_tot_active * t_sed_pocn/sed_tot
```

---

---





| Auxiliary variables, continued from previous page |
|---|

**pocp in active sediment layer [mol/m\*\*2]**

```
pocp_active =        sed_tot_active * t_sed_pocp/sed_tot
```

| end of table **Auxiliary variables** |
|---|

| **Constants** |
|---|

**oxygen half-saturation constant for recycling of sediment detritus using oxygen [mol/kg]**

```
o2_min_sed_resp =    0.000064952
```

**bio-resuspension rate [1/day]**

```
r_biores =           0.015
```

**redfield ratio C/N**

```
rfr_c =              6.625
```

**redfield ratio C/P**

```
rfr_cp =             106.0
```

**maximum sediment detritus concentration that feels erosion [mol/m\*\*2]**

```
sed_max =            1.0
```

| end of table **Constants** |
|---|

| **Process limitation factors** |
|---|

```
lim_t_o2_6 =         t_o2*t_o2/(t_o2*t_o2+o2_min_sed_resp*o2_min_sed_resp)

lim_t_sed_21 =       theta(t_sed-0.0)

lim_t_ips_23 =       theta(t_ips-0.0)
```

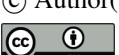



| Process limitation factors, continued from previous page |
|---|
| `lim_t_sed_poc_22 = theta(t_sed_poc-0.0)` |
| `lim_t_sed_pocn_27 = theta(t_sed_pocn-0.0)` |
| `lim_t_sed_pocp_28 = theta(t_sed_pocp-0.0)` |
| end of table **Process limitation factors** |

## B3.2 Process type BGC/benthic/mineralisation

| **Processes** |
|---|

**recycling of sedimentary detritus to ammonium using oxygen (respiration) (sediment only) [mol/m$^2$/day]**

`t_sed + 6.625*t_o2 + 0.8125*h3oplus -> t_nh4 + rfr_p*t_po4 + rfr_c*t_dic + 7.4375*h2o`

`p_sed_resp_nh4 =    (lr_sed_rec*sed_active)*lim_t_sed_21*lim_t_o2_2`

**coupled nitrification and denitrification after mineralization of detritus in oxic sediments (sediment only) [mol/m$^2$/day]**

`t_nh4 + 0.75*t_o2 -> 0.5*h2o + h3oplus + 0.5*t_n2`

`p_nh4_nitdenit_n2 =  (frac_denit_sed*(p_sed_resp_nh4+p_sed_pocn_resp)*theta(t_o2-5.0e-`
`                     6))*lim_t_nh4_11*lim_t_o2_2`

**recycling of sedimentary detritus to ammonium using nitrate (denitrification) (sediment only) [mol/m$^2$/day]**

`t_sed + 6.1125*h3oplus + 5.3*t_no3 -> rfr_c*t_dic + rfr_p*t_po4 + t_nh4 + 2.65*t_n2 +`
`15.3875*h2o`

`p_sed_denit_nh4 =    (lr_sed_rec*sed_active)*(1.0-lim_t_o2_2)*lim_t_no3_3*lim_t_sed_21`

**recycling of sedimentary detritus to ammonium using sulfate (sulfate reduction) (sediment only) [mol/m$^2$/day]**

| continued on next page... |
|---|





---

Processes, continued from previous page

---

```
t_sed + 3.3125*so4 + 7.4375*h3oplus -> t_nh4 + rfr_p*t_po4 + rfr_c*t_dic + 3.3125*
t_h2s + 14.0625*h2o
p_sed_sulf_nh4 =     (lr_sed_rec*sed_active)*(1.0-lim_t_o2_2)*(1.0-lim_t_no3_3)*
                     lim_t_sed_21
```

**recycling of sedimentary poc to dic using oxygen (respiration) (sediment only)**
$[\mathbf{mol/m^2/day}]$

```
t_sed_poc + t_o2 -> t_dic + h2o
p_sed_poc_resp =     (lr_sed_poc_rec*poc_active)*lim_t_sed_poc_22*lim_t_o2_2
```

**recycling of sedimentary poc to dic using nitrate (denitrification) (sediment only)**
$[\mathbf{mol/m^2/day}]$

```
0.8*t_no3 + 0.8*h3oplus + t_sed_poc -> 2.2*h2o + 0.4*t_n2 + t_dic
p_sed_poc_denit =     (lr_sed_poc_rec*poc_active)*(1.0-lim_t_o2_2)*lim_t_no3_3*
                      lim_t_sed_poc_22
```

**recycling of sedimentary poc to dic using sulfate (sulfate reduction) (sediment only)**
$[\mathbf{mol/m^2/day}]$

```
h3oplus + 0.5*so4 + t_sed_poc -> 2.0*h2o + 0.5*t_h2s + t_dic
p_sed_poc_sulf =     (lr_sed_poc_rec*poc_active)*(1.0-lim_t_o2_2)*(1.0-lim_t_no3_3)*
                     lim_t_sed_poc_22
```

**recycling of sedimentary pocn to dic and NH4 using oxygen (respiration) (sediment only)**
$[\mathbf{mol/m^2/day}]$

```
6.625*t_o2 + t_sed_pocn + 0.5*h3oplus -> 6.625*h2o + 6.625*t_dic + t_nh4 + 0.5*ohminus
p_sed_pocn_resp =     (lr_sed_rec*pocn_active)*lim_t_o2_2*lim_t_sed_pocn_27
```

**recycling of sedimentary pocp to dic and PO4 using oxygen (respiration) (sediment only)**
$[\mathbf{mol/m^2/day}]$

```
3*h2o + t_sed_pocp + 106*t_o2 -> 106*h2o + t_po4 + 106*t_dic + 3*h3oplus
p_sed_pocp_resp =     (lr_sed_rec*pocp_active)*lim_t_sed_pocp_28*lim_t_o2_2
```

---

---





---

Processes, continued from previous page

---

**recycling of sedimentary pocn to dic and NH4 using nitrate (denitrification) (sediment only) [mol/m$^2$/day]**

```
t_sed_pocn + 5.3*t_no3 + 5.8*h3oplus -> 6.625*t_dic + t_nh4 + 2.65*t_n2 + 14.575*h2o +
0.5*ohminus
p_sed_pocn_denit =    (lr_sed_rec*pocn_active)*(1.0-lim_t_o2_2)*lim_t_no3_3*
                      lim_t_sed_pocn_27
```

**recycling of sedimentary pocp to dic and PO4 using nitrate (denitrification) (sediment only) [mol/m$^2$/day]**

```
t_sed_pocp + 3*ohminus + 84.8*h3oplus + 84.8*t_no3 -> 106*t_dic + t_po4 + 42.4*t_n2 +
236.2*h2o
p_sed_pocp_denit =    (lr_sed_rec*pocp_active)*(1.0-lim_t_o2_2)*lim_t_no3_3*
                      lim_t_sed_pocp_28
```

**recycling of sedimentary pocn to dic and NH4 using sulfate (sulfate reduction) (sediment only) [mol/m$^2$/day]**

```
7.125*h3oplus + 3.3125*SO4 + t_pocn -> 0.5*ohminus + 13.25*H2O + 3.3125*t_h2s + t_nh4
+ 6.625*t_dic
p_sed_pocn_sulf =    (lr_sed_rec*pocn_active)*(1.0-lim_t_o2_2)*(1.0-lim_t_no3_3)*
                     lim_t_pocn_14
```

**recycling of sedimentary pocp to dic and PO4 using sulfate (sulfate reduction) (sediment only) [mol/m$^2$/day]**

```
t_pocp + 53*so4 + 106*h3oplus + 3*ohminus -> 106*t_dic + 215*h2o + 53*t_h2s + t_po4
p_sed_pocp_sulf =    (lr_sed_rec*pocp_active)*(1.0-lim_t_o2_2)*(1.0-lim_t_no3_3)*
                     lim_t_pocp_13
```

**coupled nitrification and denitrification after mineralization of pocn-detritus in oxic sediments (sediment only) [mol/m$^2$/day]**

```
t_nh4 + 0.75*t_o2 -> 0.5*h2o + h3oplus + 0.5*t_n2
```

---

---





| Processes, continued from previous page |
| --- |
| `(frac_denit_sed*p_sed_pocn_resp*theta(t_o2-5.0e-6))*lim_t_nh4_11*` |
| `p_nh4_nitdenit_pocn_n2lim_t_o2_2` |
| `=` |

| end of table **Processes** |
| --- |

| **Auxiliary variables** |
| --- |

**fraction of ammonium that is immediately nitrified and denitrified after remineralization in oxic sediments**

`frac_denit_sed =`    `frac_denit_scal*(0.5+0.5*exp(-0.01*cgt_bottomdepth))`

**total carbon in sediment layer [mol/m**2]**

`sed_tot =`    `t_sed*rfr_c + t_sed_poc + t_sed_pocn*rfr_c + t_sed_pocp*rfr_cp`

**total carbon in active sediment layer [mol/m**2]**

`sed_tot_active =`    `max(0.0,min(sed_tot,sed_max*rfr_c))`

**detritus in active sediment layer [mol/m**2]**

`sed_active =`    `sed_tot_active * t_sed/sed_tot`

**recycling rate of sediment detritus, limited by oxygen [1/d]**

`lr_sed_rec =`    `r_sed_rec*exp(q10_sed_rec*cgt_temp)*(1.0-reduced_rec*theta(2*`
    `t_h2s-t_o2))`

**recycling rate of sediment POC, limited by oxygen [1/d]**

`lr_sed_poc_rec =`    `r_sed_poc_rec*exp(q10_sed_rec*cgt_temp)*(1.0-reduced_rec*theta(2*`
    `t_h2s-t_o2))`

**poc in active sediment layer [mol/m**2]**

`poc_active =`    `sed_tot_active * t_sed_poc/sed_tot`





| Auxiliary variables, continued from previous page |
|---|

**pocn in active sediment layer [mol/m\*\*2]**

pocn_active =            sed_tot_active * t_sed_pocn/sed_tot

**pocp in active sediment layer [mol/m\*\*2]**

pocp_active =            sed_tot_active * t_sed_pocp/sed_tot

| end of table **Auxiliary variables** |
|---|

| **Constants** |
|---|

**nitrate half-saturation concentration for denitrification in the water column [mol/kg]**

no3_min_sed_denit =   1.423E-7

**q10 rule factor for detritus recycling in the sediment [1/K]**

q10_sed_rec =         0.175

**maximum recycling rate for sedimentary detritus [1/d]**

r_sed_rec =           0.003

**maximum recycling rate for sedimentary POC [1/d]**

r_sed_poc_rec =       0.0005

**redfield ratio C/N**

rfr_c =               6.625

**redfield ratio P/N**

rfr_p =               0.0625

**redfield ratio C/P**

rfr_cp =              106.0

**maximum sediment detritus concentration that feels erosion [mol/m\*\*2]**





| Constants, continued from previous page | |
|---|---|
| sed_max = | 1.0 |

**scaling frac_denit_sed**

| frac_denit_scal = | 1.0 |
|---|---|

**decrease recycling in sed under anoxia by reduce_rec**

| reduced_rec = | 0.8 |
|---|---|

| end of table **Constants** | |
|---|---|

| Process limitation factors | |
|---|---|
| lim_t_o2_2 = | theta(t_o2-0.0) |
| lim_t_nh4_11 = | theta(t_nh4-0.0) |
| lim_t_no3_3 = | t_no3*t_no3/(t_no3*t_no3+no3_min_sed_denit*no3_min_sed_denit) |
| lim_t_sed_21 = | theta(t_sed-0.0) |
| lim_t_sed_poc_22 = | theta(t_sed_poc-0.0) |
| lim_t_sed_pocn_27 = | theta(t_sed_pocn-0.0) |
| lim_t_sed_pocp_28 = | theta(t_sed_pocp-0.0) |
| lim_t_pocp_13 = | theta(t_pocp-0.0) |
| lim_t_pocn_14 = | theta(t_pocn-0.0) |

| end of table **Process limitation factors** | |
|---|---|





### B3.3    Process type BGC/benthic/P_retention

---

**Processes**

---

**retention of phosphate in the sediment under oxic conditions (sediment only)**
$[\mathrm{mol/m^2/day}]$

```
rfr_p*t_po4 + rfr_p*fe3plus -> rfr_p*t_ips
p_po4_retent_ips =    (p_sed_resp_nh4*frac_po4retent)*lim_t_o2_4*lim_t_po4_10
```

**liberation of phosphate from the sediment under anoxic conditions (sediment only)**
$[\mathrm{mol/m^2/day}]$

```
t_ips -> fe3plus + t_po4
p_ips_liber_po4 =    (t_ips*r_ips_liber)*lim_t_h2s_5*lim_t_ips_23
```

---

end of table **Processes**

---

---

**Auxiliary variables**

---

**fraction of phosphate which is retained as iron-bound phosphate instead of being released after mineralization in the sediment [1]**

```
frac_po4retent =     ret_po4_1 + ret_po4_2*theta(cgt_latitude-60.75) + ret_po4_3*
                     theta(cgt_latitude-63.75)
```

---

end of table **Auxiliary variables**

---

---

**Constants**

---

**minimum h2s concentration for liberation of iron phosphate from the sediment [mol/kg]**

```
h2s_min_po4_liber =   1.0E-6
```

**oxygen half-saturation concentration for retension of phosphate during sediment denitrification [mol/kg]**

```
o2_min_po4_retent =   0.0000375
```

---

---





---

Constants, continued from previous page

---

**PO4 liberation rate under anoxic conditions [1/day]**

r_ips_liber =          0.1

**redfield ratio P/N**

rfr_p =          0.0625

**PO4 retension in oxic sediments**

ret_po4_1 =          0.1

**additional PO4 retension in oxic sediments of the Bothnian Sea**

ret_po4_2 =          0.5

**additional PO4 retension in oxic sediments of the Bothnian Sea**

ret_po4_3 =          0.13

---

end of table <b>Constants</b>

---

---

<b>Process limitation factors</b>

---

lim_t_o2_4 =          t_o2*t_o2/(t_o2*t_o2+o2_min_po4_retent*o2_min_po4_retent)

lim_t_po4_10 =          theta(t_po4-0.0)

lim_t_h2s_5 =          theta(t_h2s-h2s_min_po4_liber)

lim_t_ips_23 =          theta(t_ips-0.0)

---

end of table <b>Process limitation factors</b>

---

5

### B3.4   Process type BGC/pelagic/mineralisation





| Processes |
| --- |

**recycling of POC using nitrate (denitrification) [mol/kg/day]**

```
t_poc + 0.8*t_no3 + 0.8*h3oplus -> t_dic + 2.2*h2o + 0.4*t_n2
p_poc_denit =        (t_poc*r_poc_rec*exp(q10_det_rec*cgt_temp))*(1.0-lim_t_o2_0)*
                     lim_t_no3_1*lim_t_poc_12
```

**Mineralization of POC, e-acceptor sulfate (sulfate reduction) [mol/kg/day]**

```
t_poc + 0.5*so4 + h3oplus -> t_dic + 0.5*t_h2s + 2*h2o
p_poc_sulf =         (t_poc*r_poc_rec*exp(q10_det_rec*cgt_temp))*(1.0-lim_t_o2_0)*
                     (1.0-lim_t_no3_1)*lim_t_poc_12
```

**respiration of POCP [mol/kg/day]**

```
106*t_o2 + t_pocp + 3*H2O -> 106*t_dic + t_po4 + 106*H2O + 3*h3oplus
p_pocp_resp =        (t_pocp * lr_pocp * exp(q10_det_rec * cgt_temp))*lim_t_o2_0*
                     lim_t_pocp_13
```

**recycling of POC using nitrate (denitrification) [mol/kg/day]**

```
3*ohminus + 84.8*h3oplus + 84.8*t_no3 + t_pocp -> t_po4 + 42.4*t_n2 + 236.2*H2O + 106*
t_dic
p_pocp_denit =       (t_pocp*r_pocp_rec*exp(q10_det_rec*cgt_temp))*(1.0-lim_t_o2_0)*
                     lim_t_no3_1*lim_t_pocp_13
```

**Mineralization of POC, e-acceptor sulfate (sulfate reduction) [mol/kg/day]**

```
t_pocp + 53*so4 + 106*h3oplus + 3*ohminus -> 106*t_dic + 215*h2o + 53*t_h2s + t_po4
p_pocp_sulf =        (t_pocp*r_pocp_rec*exp(q10_det_rec*cgt_temp))*(1.0-lim_t_o2_0)*
                     (1.0-lim_t_no3_1)*lim_t_pocp_13
```

**respiration of POCN [mol/kg/day]**

```
0.5*h3oplus + 6.625*t_o2 + t_pocn -> 0.5*ohminus + 6.625*H2O + t_nh4 + 6.625*t_dic
p_pocn_resp =        (t_pocn * lr_pocn * exp(q10_det_rec * cgt_temp))*lim_t_o2_0*
                     lim_t_pocn_14
```

**recycling of POCN using nitrate (denitrification) [mol/kg/day]**





---

Processes, continued from previous page

---

```
5.8*h3oplus + 5.3*t_no3 + t_pocn -> 0.5*ohminus + 14.575*H2O + 2.65*t_n2 + t_nh4 +
6.625*t_dic
p_pocn_denit =         (t_pocn*r_pocn_rec*exp(q10_det_rec*cgt_temp))*(1.0-lim_t_o2_0)*
                       lim_t_no3_1*lim_t_pocn_14
```

**Mineralization of POCN, e-acceptor sulfate (sulfate reduction) [mol/kg/day]**

```
t_pocn + 3.3125*SO4 + 7.125*h3oplus -> 6.625*t_dic + t_nh4 + 3.3125*t_h2s + 13.25*H2O
+ 0.5*ohminus
p_pocn_sulf =          (t_pocn*r_pocn_rec*exp(q10_det_rec*cgt_temp))*(1.0-lim_t_o2_0)*
                       (1.0-lim_t_no3_1)*lim_t_pocn_14
```

**recycling of detritus using oxygen (respiration) [mol/kg/day]**

```
t_det + 6.625*t_o2 + 0.8125*h3oplus -> t_nh4 + rfr_p*t_po4 + rfr_c*t_dic + 7.4375*h2o
p_det_resp_nh4 =       (t_det*r_det_rec*exp(q10_det_rec*cgt_temp))*lim_t_o2_0*
                       lim_t_det_20
```

**recycling of detritus using nitrate (denitrification) [mol/kg/day]**

```
t_det + 5.3*t_no3 + 6.1125*h3oplus -> 2.65*t_n2 + 15.3875*h2o + t_nh4 + rfr_p*t_po4 +
rfr_c*t_dic
p_det_denit_nh4 =      (t_det*r_det_rec*exp(q10_det_rec*cgt_temp))*(1.0-lim_t_o2_0)*
                       lim_t_no3_1*lim_t_det_20
```

**recycling of detritus using sulfate (sulfate reduction) [mol/kg/day]**

```
7.4375*h3oplus + 3.3125*so4 + t_det -> 14.0625*h2o + 3.3125*t_h2s + rfr_c*t_dic +
rfr_p*t_po4 + t_nh4
p_det_sulf_nh4 =       (t_det*r_det_rec*exp(q10_det_rec*cgt_temp))*(1.0-lim_t_o2_0)*
                       (1.0-lim_t_no3_1)*lim_t_det_20
```

**recycling of DOC using nitrate (denitrification) [mol/kg/day]**

```
t_doc + 0.8*t_no3 + 0.8*h3oplus -> t_dic + 2.2*h2o + 0.4*t_n2
p_doc_denit =          (t_doc*r_doc_rec*exp(q10_det_rec*cgt_temp))*(1.0-lim_t_o2_0)*
                       lim_t_no3_1*lim_t_doc_29
```

---

continued on next page...

---





---

Processes, continued from previous page

---

**Mineralization of DOC, e-acceptor sulfate (sulfate reduction) [mol/kg/day]**

```
t_doc + 0.5*so4 + h3oplus -> t_dic + 0.5*t_h2s + 2*h2o

p_doc_sulf =        (t_doc*r_doc_rec*exp(q10_det_rec*cgt_temp))*(1.0-lim_t_o2_0)*
                    (1.0-lim_t_no3_1)*lim_t_doc_29
```

**respiration of DOP [mol/kg/day]**

```
3*H2O + t_dop + 106*t_o2 -> 3*h3oplus + 106*H2O + t_po4 + 106*t_dic

p_dop_resp =        (t_dop * lr_dop * exp(q10_det_rec * cgt_temp))*lim_t_o2_0*
                    lim_t_dop_30
```

**recycling of DOP using nitrate (denitrification) [mol/kg/day]**

```
t_dop + 84.8*t_no3 + 84.8*h3oplus + 3*ohminus -> 106*t_dic + 236.2*H2O + 42.4*t_n2 +
t_po4

p_dop_denit =       (t_dop*r_dop_rec*exp(q10_det_rec*cgt_temp))*(1.0-lim_t_o2_0)*
                    lim_t_no3_1*lim_t_dop_30
```

**Mineralization of DOP, e-acceptor sulfate (sulfate reduction) [mol/kg/day]**

```
3*ohminus + 106*h3oplus + 53*so4 + t_dop -> t_po4 + 53*t_h2s + 215*h2o + 106*t_dic

p_dop_sulf =        (t_dop*r_dop_rec*exp(q10_det_rec*cgt_temp))*(1.0-lim_t_o2_0)*
                    (1.0-lim_t_no3_1)*lim_t_dop_30
```

**respiration of DON [mol/kg/day]**

```
0.5*h3oplus + 6.625*t_o2 + t_don -> 0.5*ohminus + 6.625*H2O + t_nh4 + 6.625*t_dic

p_don_resp =        (t_don * lr_don * exp(q10_det_rec * cgt_temp))*lim_t_o2_0*
                    lim_t_don_31
```

**recycling of DON using nitrate (denitrification) [mol/kg/day]**

```
5.8*h3oplus + 5.3*t_no3 + t_don -> 0.5*ohminus + 14.575*H2O + 2.65*t_n2 + t_nh4 +
6.625*t_dic

p_don_denit =       (t_don*r_don_rec*exp(q10_det_rec*cgt_temp))*(1.0-lim_t_o2_0)*
                    lim_t_no3_1*lim_t_don_31
```

---

---





---

Processes, continued from previous page

---

**Mineralization of DON, e-acceptor sulfate (sulfate reduction) [mol/kg/day]**

`7.125*h3oplus + 3.3125*SO4 + t_don -> 0.5*ohminus + 13.25*H2O + 3.3125*t_h2s + t_nh4 +`
`6.625*t_dic`

```
p_don_sulf =           (t_don*r_don_rec*exp(q10_det_rec*cgt_temp))*(1.0-lim_t_o2_0)*
                       (1.0-lim_t_no3_1)*lim_t_don_31
```

**decay of cdom due to light [mol/kg/day]**

`t_cdom ->`

```
p_cdom_decay =         (t_cdom*r_cdom_decay*cgt_light/r_cdom_light)*lim_t_cdom_32
```

---

end of table **Processes**

---

---

**Auxiliary variables**

**dissolved inorganic nitrogen [mol/kg]**

```
din =                  t_no3+t_nh4
```

**squared DIN [mol2/kg2]**

```
din_sq =               din*din
```

**squared phosphate [mol\*\*2/kg\*\*2]**

```
po4_sq =               t_po4*t_po4
```

**modifies pocp recycling towards Refield ratio if PO4 is depleted**

```
ref_p_sw =             (1 - (po4_sq/(rfr_p*din_min_lpp*rfr_p*din_min_lpp+po4_sq)))/(1+
                       exp(6.0*(1-din/(t_po4/rfr_p+epsilon))))
```

**modifies pocn recycling towards Refield ratio if DIN is depleted**

```
ref_n_sw =             (1 - (din_sq/(din_min_lpp*din_min_lpp+din_sq)))/(1+exp(6.0*(1-
                       t_po4/rfr_p/(din+epsilon))))
```

---

---





---

Auxiliary variables, continued from previous page

---

**add an additional POCP recycling if PO4 below Redfield but sufficient DIN**

```
lr_pocp =          r_pocp_rec*(1 + fac_enh_rec*ref_p_sw)
```

**add an additional DOP recycling if PO4 is below Redfield but sufficient DIN**

```
lr_dop =           r_dop_rec*(1 + fac_enh_rec*ref_p_sw)
```

**add an additional POCN recycling if DIN below Redfield but sufficient PO4**

```
lr_pocn =          r_pocn_rec*(1 + fac_enh_rec*ref_n_sw)
```

**add an additional DON recycling if DIN below Redfield but sufficient PO4**

```
lr_don =           r_don_rec*(1 + fac_enh_rec*ref_n_sw)
```

---

end of table **Auxiliary variables**

---

---

**Constants**

---

**DIN half saturation constant for large-cell phytoplankton growth [mol/kg]**

```
din_min_lpp =      1.0E-6
```

**no division by 0**

```
epsilon =          4.5E-17
```

**minimum no3 concentration for recycling of detritus using nitrate (denitrification)**

```
no3_min_det_denit =   1.0E-9
```

**oxygen half-saturation constant for detritus recycling [mol/kg]**

```
o2_min_det_resp =    1.0E-6
```

**q10 rule factor for recycling [1/K]**

```
q10_det_rec =        0.15
```

**recycling rate (detritus to ammonium) at 0℃ [1/day]**

---

---





| Constants, continued from previous page |
| --- |

`r_det_rec =`          0.003

**redfield ratio C/N**

`rfr_c =`          6.625

**redfield ratio P/N**

`rfr_p =`          0.0625

**recycling rate (poc to dic) at 0°C [1/day]**

`r_poc_rec =`          0.003

**recycling rate (pocp to dic and po4) at 0°C [1/day]**

`r_pocp_rec =`          0.002

**recycling rate (pocn to dic and nh4) at 0°C [1/day]**

`r_pocn_rec =`          0.002

**enhance recyclig of DON,POCN/DOP,POCP in case of limiting DIN/DIP**

`fac_enh_rec =`          10.0

**recycling rate (doc to dic) at 0°C [1/day]**

`r_doc_rec =`          0.001

**recycling rate (don to dic and NH4) at 0°C [1/day]**

`r_don_rec =`          0.001

**recycling rate (dop to dic and PO4) at 0°C [1/day]**

`r_dop_rec =`          0.001

**decay rate of cdom**

`r_cdom_decay =`          0.0035





| Constants, continued from previous page |
|---|

**PAR intensity controling CDOM decay**

`r_cdom_light =`      40.0

| end of table **Constants** |
|---|

| Process limitation factors |
|---|

`lim_t_o2_0 =`      `1.0-exp(-t_o2/o2_min_det_resp)`

`lim_t_no3_1 =`      `1.0-exp(-t_no3/no3_min_det_denit)`

`lim_t_doc_29 =`      `theta(t_doc-0.0)`

`lim_t_dop_30 =`      `theta(t_dop-0.0)`

`lim_t_don_31 =`      `theta(t_don-0.0)`

`lim_t_cdom_32 =`      `theta(t_cdom-0.0)`

`lim_t_det_20 =`      `theta(t_det-0.0)`

`lim_t_poc_12 =`      `theta(t_poc-0.0)`

`lim_t_pocp_13 =`      `theta(t_pocp-0.0)`

`lim_t_pocn_14 =`      `theta(t_pocn-0.0)`

| end of table **Process limitation factors** |
|---|

### B3.5 Process type BGC/pelagic/phytoplankton





---

### Processes

---

**assimilation of nitrate by large-cell phytoplankton [mol/kg/day]**

```
t_no3 + rfr_p*t_po4 + rfr_c*t_dic + 6.4375*h2o + 1.1875*h3oplus -> t_lpp + 8.625*t_o2

p_no3_assim_lpp =    (lpp_plus_lpp0*lr_assim_lpp*t_no3/(din+epsilon))*lim_t_no3_9*
                     lim_t_po4_10*lim_t_dic_8
```

**assimilation of ammonium by large-cell phytoplankton [mol/kg/day]**

```
7.4375*h2o + rfr_c*t_dic + rfr_p*t_po4 + t_nh4 -> 0.8125*h3oplus + 6.625*t_o2 + t_lpp

p_nh4_assim_lpp =    (lpp_plus_lpp0*lr_assim_lpp*t_nh4/(din+epsilon))*lim_t_dic_8*
                     lim_t_po4_10*lim_t_nh4_11
```

**assimilation of nitrate by small-cell phytoplankton [mol/kg/day]**

```
t_no3 + rfr_p*t_po4 + rfr_c*t_dic + 6.4375*h2o + 1.1875*h3oplus -> t_spp + 8.625*t_o2

p_no3_assim_spp =    (spp_plus_spp0*lr_assim_spp*t_no3/(din+epsilon))*lim_t_no3_9*
                     lim_t_po4_10*lim_t_dic_8
```

**assimilation of ammonium by small-cell phytoplankton [mol/kg/day]**

```
7.4375*h2o + rfr_c*t_dic + rfr_p*t_po4 + t_nh4 -> 0.8125*h3oplus + 6.625*t_o2 + t_spp

p_nh4_assim_spp =    (spp_plus_spp0*lr_assim_spp*t_nh4/(din+epsilon))*lim_t_dic_8*
                     lim_t_po4_10*lim_t_nh4_11
```

**assimilation of ammonium by limnic phytoplankton [mol/kg/day]**

```
t_nh4 + rfr_p*t_po4 + rfr_c*t_dic + 7.4375*h2o -> t_lip + 6.625*t_o2 + 0.8125*h3oplus

p_nh4_assim_lip =    (lip_plus_lip0*lr_assim_lip*t_nh4/(din+epsilon))*lim_t_nh4_11*
                     lim_t_po4_10*lim_t_dic_8
```

**assimilation of nitrate by limnic phytoplankton [mol/kg/day]**

```
1.1875*h3oplus + 6.4375*h2o + rfr_c*t_dic + rfr_p*t_po4 + t_no3 -> 8.625*t_o2 + t_lip

p_no3_assim_lip =    (lip_plus_lip0*lr_assim_lip*t_no3/(din+epsilon))*lim_t_dic_8*
                     lim_t_po4_10*lim_t_no3_9
```

**fixation of dinitrogen by diazotroph cyanobacteria [mol/kg/day]**

---

---



---

Processes, continued from previous page

---

```
7.9375*h2o + rfr_c*t_dic + rfr_p*t_po4 + 0.5*t_n2 + 0.1875*h3oplus -> 7.375*t_o2 +
t_cya
p_n2_assim_cya =     (cya_plus_cya0*lr_assim_cya)*lim_t_dic_8*lim_t_po4_10*lim_t_n2_7
```

**Production of DOC by LPP [mol/kg/day]**
```
h2o + t_dic -> t_o2 + t_doc
p_assim_lpp_doc =     (rfr_c * t_lpp * lr_assim_lpp_doc)*lim_t_dic_8
```

**Production of DOC by SPP [mol/kg/day]**
```
h2o + t_dic -> t_o2 + t_doc
p_assim_spp_doc =     (rfr_c * t_spp * lr_assim_spp_doc)*lim_t_dic_8
```

**Production of DOC by LPP [mol/kg/day]**
```
t_dic + h2o -> t_doc + t_o2
p_assim_lip_doc =     (rfr_c * t_lip * lr_assim_lip_doc)*lim_t_dic_8
```

**Production of DOC by CYA [mol/kg/day]**
```
t_dic + h2o -> t_doc + t_o2
p_assim_cya_doc =     (rfr_c * t_cya * lr_assim_cya_doc)*lim_t_dic_8
```

**Production of DOP by LPP [mol/kg/day]**
```
3*h3oplus + 106*h2o + t_po4 + 106*t_dic -> 3*h2o + 106*t_o2 + t_dop
p_assim_lpp_dop =     (rfr_p * t_lpp * lr_assim_lpp_dop)*lim_t_po4_10*lim_t_dic_8
```

**Production of DOP by SPP [mol/kg/day]**
```
106*t_dic + t_po4 + 106*h2o + 3*h3oplus -> t_dop + 106*t_o2 + 3*h2o
p_assim_spp_dop =     (rfr_p * t_spp * lr_assim_spp_dop)*lim_t_dic_8*lim_t_po4_10
```

**Production of DOP by LIP [mol/kg/day]**
```
3*h3oplus + 106*h2o + t_po4 + 106*t_dic -> 3*h2o + 106*t_o2 + t_dop
p_assim_lip_dop =     (rfr_p * t_lip * lr_assim_lip_dop)*lim_t_po4_10*lim_t_dic_8
```

---

---





---

Processes, continued from previous page

---

**Production of DON by LPP [mol/kg/day]**

```
rfr_c*t_dic + t_nh4 + 6.625*H2O + ohminus -> t_don + 6.625*t_o2 + H2O

p_nh4_assim_lpp_don = (t_lpp * lr_assim_lpp_don*t_nh4/(din+epsilon))*lim_t_dic_8*
                      lim_t_nh4_11
```

**Production of DON by LPP [mol/kg/day]**

```
h3oplus + 6.625*H2O + t_no3 + rfr_c*t_dic -> 8.625*t_o2 + t_don

p_no3_assim_lpp_don = (t_lpp * lr_assim_lpp_don*t_no3/(din+epsilon))*lim_t_no3_9*
                      lim_t_dic_8
```

**Production of DON by SPP [mol/kg/day]**

```
ohminus + 6.625*H2O + t_nh4 + rfr_c*t_dic -> H2O + 6.625*t_o2 + t_don

p_nh4_assim_spp_don = (t_spp * lr_assim_spp_don*t_nh4/(din+epsilon))*lim_t_nh4_11*
                      lim_t_dic_8
```

**Production of DON by SPP [mol/kg/day]**

```
rfr_c*t_dic + t_no3 + 6.625*H2O + h3oplus -> t_don + 8.625*t_o2

p_no3_assim_spp_don = (t_spp * lr_assim_spp_don*t_no3/(din+epsilon))*lim_t_dic_8*
                      lim_t_no3_9
```

**Production of DON by LIP [mol/kg/day]**

```
ohminus + 6.625*H2O + t_nh4 + rfr_c*t_dic -> H2O + 6.625*t_o2 + t_don

p_nh4_assim_lip_don = (t_lip * lr_assim_lip_don*t_nh4/(din+epsilon))*lim_t_nh4_11*
                      lim_t_dic_8
```

**Production of DON by LIP [mol/kg/day]**

```
rfr_c*t_dic + t_no3 + 6.625*H2O + h3oplus -> t_don + 8.625*t_o2

p_no3_assim_lip_don = (t_lip * lr_assim_lip_don*t_no3/(din+epsilon))*lim_t_dic_8*
                      lim_t_no3_9
```

**respiration of POC [mol/kg/day]**

```
t_poc + t_o2 -> t_dic + h2o
```

---

---





---

Processes, continued from previous page

---

p_poc_resp =            (t_poc * r_poc_rec * exp(q10_det_rec * cgt_temp))*lim_t_o2_0*
                        lim_t_poc_12

**respiration of large-cell phytoplankton [mol/kg/day]**

t_lpp + 6.625*t_o2 + 0.8125*h3oplus -> don_fraction*t_don + (1-don_fraction)*t_nh4 +
rfr_p*t_po4 + rfr_c*t_dic + 7.4375*h2o

p_lpp_resp_nh4 =        (t_lpp*r_lpp_resp)*lim_t_lpp_15*lim_t_o2_2

**respiration of small-cell phytoplankton [mol/kg/day]**

0.8125*h3oplus + 6.625*t_o2 + t_spp -> 7.4375*h2o + rfr_c*t_dic + rfr_p*t_po4 + (1-
don_fraction)*t_nh4 + don_fraction*t_don

p_spp_resp_nh4 =        (t_spp*r_spp_resp)*lim_t_o2_2*lim_t_spp_16

**respiration of limnic phytoplankton [mol/kg/day]**

0.8125*h3oplus + 6.625*t_o2 + t_lip -> 7.4375*h2o + rfr_c*t_dic + rfr_p*t_po4 + (1-
don_fraction)*t_nh4 + don_fraction*t_don

p_lip_resp_nh4 =        (t_lip*r_lip_resp)*lim_t_o2_2*lim_t_lip_18

**respiration of diazotroph cyanobacteria [mol/kg/day]**

0.8125*h3oplus + 6.625*t_o2 + t_cya -> 7.4375*h2o + rfr_c*t_dic + rfr_p*t_po4 +
don_fraction*t_don + (1-don_fraction)*t_nh4

p_cya_resp_nh4 =        (t_cya*r_cya_resp)*lim_t_o2_2*lim_t_cya_17

**mortality of large-cell phytoplankton [mol/kg/day]**

t_lpp -> t_det

p_lpp_mort_det =        (t_lpp*r_pp_mort*(1+9*theta(5.0e-6-t_o2)))*lim_t_lpp_15

**mortality of small-scale phytoplankton [mol/kg/day]**

t_spp -> t_det

p_spp_mort_det =        (t_spp*r_pp_mort*(1+9*theta(5.0e-6-t_o2)))*lim_t_spp_16

**mortality of limnic phytoplankton [mol/kg/day]**

---

---





---

Processes, continued from previous page

---

```
t_lip -> t_det
p_lip_mort_det =      (t_lip*r_pp_mort*(1+9*theta(5.0e-6-t_o2)))*lim_t_lip_18
```

**mortality of diazotroph cyanobacteria [mol/kg/day]**
```
t_cya -> t_det
p_cya_mort_det =      (t_cya*r_pp_mort*(1+9*theta(5.0e-6-t_o2)))*lim_t_cya_17
```

**mortality of diazotroph cyanobacteria due to strong turbulence [mol/kg/day]**
```
t_cya -> t_det
p_cya_mort_det_diff = (t_cya*r_pp_mort*(r_cya_mort_diff*theta(cgt_diffusivity-
                      r_cya_mort_thresh)))*lim_t_cya_17
```

**respiration of DOC [mol/kg/day]**
```
t_o2 + t_doc -> h2o + t_dic
p_doc_resp =          (t_doc * r_doc_rec * exp(q10_doc_rec * cgt_temp))*lim_t_o2_0*
                      lim_t_doc_29
```

---

end of table Processes

---

---

Auxiliary variables

---

**square of positive temperature [°C * °C]**
```
temp_sq =             max(0.0,cgt_temp)*max(0.0,cgt_temp)
```

**dissolved inorganic nitrogen [mol/kg]**
```
din =                 t_no3+t_nh4
```

**squared DIN [mol2/kg2]**
```
din_sq =              din*din
```

**squared phosphate [mol**2/kg**2]**
```
po4_sq =              t_po4*t_po4
```

---

---





---

Auxiliary variables, continued from previous page

---

**large-cell phytoplankton plus seed concentration [mol/kg]**

```
lpp_plus_lpp0 =        t_lpp+lpp0
```

**small-cell phytoplankton plus seed concentration [mol/kg]**

```
spp_plus_spp0 =        t_spp+spp0
```

**limnic phytoplankton plus seed concentration [mol/kg]**

```
lip_plus_lip0 =        t_lip+lip0
```

**diazotroph cyanobacteria plus seed concentration [mol/kg]**

```
cya_plus_cya0 =        t_cya+cya0
```

**light limitation factor for large-cell phytoplankton growth [1]**

```
temp1 =                max(cgt_light/2.0,light_opt_lpp)
lim_light_lpp =        cgt_light/temp1*exp(1-cgt_light/temp1)
```

**light limitation factor for small-cell phytoplankton growth [1]**

```
temp1 =                max(cgt_light/2.0,light_opt_spp)
lim_light_spp =        cgt_light/temp1*exp(1-cgt_light/temp1)
```

**light limitation factor for limnic phytoplankton growth [1]**

```
temp1 =                max(cgt_light/2.0,light_opt_lip)
lim_light_lip =        cgt_light/temp1*exp(1-cgt_light/temp1)
```

**light limitation factor for diazotroph cyanobacteria growth [1]**

```
temp1 =                max(cgt_light/2.0,light_opt_cya)
lim_light_cya =        cgt_light/temp1*exp(1-cgt_light/temp1)
```

**growth rate of large-cell phytoplankton, limited by DIN, DIP, light and oxygen [1/day]**

---

---





---

Auxiliary variables, continued from previous page

---

lr_assim_lpp =      r_lpp_assim*theta(t_o2-2*t_h2s)*min(din_sq/(din_sq+din_min_lpp*
                    din_min_lpp),min(po4_sq/(po4_sq+din_min_lpp*din_min_lpp*rfr_p*
                    rfr_p),lim_light_lpp))

**growth rate of small-cell phytoplankton, limited by DIN, DIP, light, oxygen and
temperature [1/day]**

lr_assim_spp =      r_spp_assim*theta(t_o2-2*t_h2s)*min(din_sq/(din_sq+din_min_spp*
                    din_min_spp),min(po4_sq/(po4_sq+din_min_spp*din_min_spp*rfr_p*
                    rfr_p),lim_light_spp))*(1+temp_sq/(temp_sq+temp_min_spp*
                    temp_min_spp))

**growth rate of limnic phytoplankton, limited by DIN, DIP, light, salt and oxygen [1/day]**

lr_assim_lip =      r_lip_assim*theta(t_o2-2*t_h2s)*min(din_sq/(din_sq+din_min_lip*
                    din_min_lip),min(po4_sq/(po4_sq+din_min_lip*din_min_lip*rfr_p*
                    rfr_p),lim_light_lip))*(1/(1+exp(cgt_sali*cgt_sali-sali_max_lip*
                    sali_max_lip)))

**growth rate of diazotroph cyanobacteria, limited by DIP, light, oxygen, temperature and
salinity [1/day]**

lr_assim_cya =      r_cya_assim*theta(t_o2-2*t_h2s)*min(po4_sq/(po4_sq+dip_min_cya*
                    dip_min_cya),lim_light_cya)*(1/(1+exp(temp_switch_cya*
                    (temp_min_cya-cgt_temp))))*(1/(1+exp(cgt_sali-sali_max_cya)))*
                    (1/(1+exp(sali_min_cya-cgt_sali)))*(1/(1+exp(nit_switch_cya*(din-
                    nit_max_cya))))

**production rate of DOC by LPP**

lr_assim_lpp_doc =  fac_doc_assim_lpp * r_lpp_assim * theta(t_o2-2*t_h2s) * min(max(1
                    - din_sq/(din_sq+din_min_lpp*din_min_lpp),1 -
                    po4_sq/(din_min_lpp*din_min_lpp*rfr_p*rfr_p + po4_sq)),
                    lim_light_lpp)

**production rate of DOC by SPP**

---

---





---

Auxiliary variables, continued from previous page

---

lr_assim_spp_doc =     fac_doc_assim_spp * r_spp_assim * theta(t_o2-2*t_h2s) * min(max(1
                       - din_sq/(din_sq+din_min_spp*din_min_spp),1 -
                       po4_sq/(din_min_spp*din_min_spp*rfr_p*rfr_p + po4_sq)),
                       lim_light_spp)*(1+temp_sq/(temp_sq+temp_min_spp*temp_min_spp))

**production rate of DOC by CYA**

lr_assim_cya_doc =     fac_doc_assim_cya * r_cya_assim*theta(t_o2-2*t_h2s)*min(1 -
                       po4_sq/(po4_sq+dip_min_cya*dip_min_cya),lim_light_cya)*(1/(1+
                       exp(temp_switch_cya*(temp_min_cya-cgt_temp))))*(1/(1+
                       exp(cgt_sali-sali_max_cya)))*(1/(1+exp(sali_min_cya-cgt_sali)))

**production rate of DOC by LPP**

lr_assim_lip_doc =     fac_doc_assim_lip * r_lip_assim * theta(t_o2-2*t_h2s) * min(max(1
                       - din_sq/(din_sq+din_min_lip*din_min_lip),1 -
                       po4_sq/(din_min_lip*din_min_lip*rfr_p*rfr_p + po4_sq)),
                       lim_light_lip)*(1/(1+exp(cgt_sali-sali_max_lip)))

**production rate of DOP by LPP**

lr_assim_lpp_dop =     fac_dop_assim * r_lpp_assim * theta(t_o2-2*t_h2s) * min(min(1 -
                       din_sq/(din_sq+din_min_lpp*din_min_lpp),po4_sq/(din_min_lpp*
                       din_min_lpp*rfr_p*rfr_p + po4_sq)), lim_light_lpp)

**production rate of DOP by SPP**

lr_assim_spp_dop =     fac_dop_assim * r_spp_assim * theta(t_o2-2*t_h2s) * min(min(1 -
                       din_sq/(din_sq+din_min_spp*din_min_spp),po4_sq/(din_min_spp*
                       din_min_spp*rfr_p*rfr_p + po4_sq)), lim_light_spp)*(1+
                       temp_sq/(temp_sq+temp_min_spp*temp_min_spp))

**production rate of DOP by LPP**

---

continued on next page...

---

lr_assim_lip_dop = fac_dop_assim * r_lip_assim * theta(t_o2-2*t_h2s) * min(min(1 - din_sq/(din_sq+din_min_lip*din_min_lip),po4_sq/(din_min_lip* din_min_lip*rfr_p*rfr_p + po4_sq)), lim_light_lip)*(1/(1+ exp(cgt_sali-sali_max_lip)))

**production rate of DON by LPP**

lr_assim_lpp_don = fac_don_assim * r_lpp_assim * theta(t_o2-2*t_h2s) * min(min(din_sq/(din_sq+din_min_lpp*din_min_lpp),1 - po4_sq/(din_min_lpp*din_min_lpp*rfr_p*rfr_p + po4_sq)), lim_light_lpp)

**production rate of DON by SPP**

lr_assim_spp_don = fac_don_assim * r_spp_assim * theta(t_o2-2*t_h2s) * min(min(din_sq/(din_sq+din_min_spp*din_min_spp),1 - po4_sq/(din_min_spp*din_min_spp*rfr_p*rfr_p + po4_sq)), lim_light_spp)*(1+temp_sq/(temp_sq+temp_min_spp*temp_min_spp))

**production rate of DON by limnic phytoplankton**

lr_assim_lip_don = fac_don_assim * r_lip_assim * theta(t_o2-2*t_h2s) * min(min(din_sq/(din_sq+din_min_lip*din_min_lip),1 - po4_sq/(din_min_lip*din_min_lip*rfr_p*rfr_p + po4_sq)), lim_light_lip)*(1/(1+exp(cgt_sali-sali_max_lip)))

end of table **Auxiliary variables**

**Constants**

**seed concentration for diazotroph cyanobacteria [mol/kg]**

cya0 = 9.0E-8

**DIN half saturation constant for large-cell phytoplankton growth [mol/kg]**

din_min_lpp = 1.0E-6





---

Constants, continued from previous page

---

**DIN half saturation constant for small-cell phytoplankton growth [mol/kg]**

```
din_min_spp =        1.6E-7
```

**DIP half saturation constant for diazotroph cyanobacteria growth [mol/kg]**

```
dip_min_cya =        1.0E-8
```

**DIN half saturation constant for limnic phytoplankton growth [mol/kg]**

```
din_min_lip =        1.0E-6
```

**no division by 0**

```
epsilon =            4.5E-17
```

**optimal light for diazotroph cyanobacteria growth [W/m**2]**

```
light_opt_cya =      50.0
```

**optimal light for large-cell phytoplankton growth [W/m**2]**

```
light_opt_lpp =      35.0
```

**optimal light for small-cell phytoplankton growth [W/m**2]**

```
light_opt_spp =      50.0
```

**optimal light for limnic phytoplankton growth [W/m**2]**

```
light_opt_lip =      30.0
```

**seed concentration for limnic phytoplankton [mol/kg]**

```
lip0 =               4.5E-9
```

**seed concentration for large-cell phytoplankton [mol/kg]**

```
lpp0 =               4.5E-9
```

**oxygen half-saturation constant for detritus recycling [mol/kg]**

---

---





---

Constants, continued from previous page

---

o2_min_det_resp =     1.0E-6

**q10 rule factor for recycling [1/K]**

q10_det_rec =         0.15

**q10 rule factor for DOC recycling [1/K]**

q10_doc_rec =         0.069

**maximum rate for nutrient uptake of diazotroph cyanobacteria [1/day]**

r_cya_assim =         0.75

**respiration rate of cyanobacteria to ammonium [1/day]**

r_cya_resp =          0.01

**maximum rate for nutrient uptake of large-cell phytoplankton [1/day]**

r_lpp_assim =         1.38

**respiration rate of large phytoplankton to ammonium [1/day]**

r_lpp_resp =          0.075

**maximum rate for nutrient uptake of limnic phytoplankton [1/day]**

r_lip_assim =         1.38

**respiration rate of limnic phytoplankton to ammonium [1/day]**

r_lip_resp =          0.075

**mortality rate of phytoplankton [1/day]**

r_pp_mort =           0.03

**enhanced cya mortality due to strong turbulence**

r_cya_mort_diff =     40.0

---

continued on next page...

---





| Constants, continued from previous page |
| --- |

**diffusivity threshold for enhanced cyano mortality**

`r_cya_mort_thresh =   0.02`

**maximum rate for nutrient uptake of small-cell phytoplankton [1/day]**

`r_spp_assim =        0.4`

**respiration rate of small phytoplankton to ammonium [1/day]**

`r_spp_resp =         0.0175`

**redfield ratio C/N**

`rfr_c =              6.625`

**redfield ratio P/N**

`rfr_p =              0.0625`

**upper salinity limit - diazotroph cyanobacteria [psu]**

`sali_max_cya =       8.0`

**lower salinity limit - diazotroph cyanobacteria [psu]**

`sali_min_cya =       4.0`

**limits cyano growth in DIN reach environment**

`nit_max_cya =        5.0E-7`

**strengs of DIN control for cyano growth**

`nit_switch_cya =     8.0`

**lower salinity limit - limnic phytoplankton [psu]**

`sali_max_lip =       2.0`

**seed concentration for small-cell phytoplankton [mol/kg]**

`spp0 =               4.5E-9`





---

Constants, continued from previous page

---

**lower temperature limit - diazotroph cyanobacteria [℃]**

`temp_min_cya =        13.5`

**strengs of temperature control for cyano growth**

`temp_switch_cya =     4.0`

**lower temperature limit - small-cell phytoplankton [℃]**

`temp_min_spp =        10.0`

**fraction of DON in respiration products**

`don_fraction =        0.0`

**recycling rate (poc to dic) at 0℃ [1/day]**

`r_poc_rec =           0.003`

**factor modifying DOC assimilation rate of large phytoplankton LPP**

`fac_doc_assim_lpp =   1.0`

**factor modifying DOC assimilation rate of cyanobacteria**

`fac_doc_assim_cya =   1.0`

**factor modifying DOC assimilation rate of small phytoplankton SPP**

`fac_doc_assim_spp =   1.0`

**factor modifying DOC assimilation rate of limnic phytoplankton LIP**

`fac_doc_assim_lip =   1.0`

**factor modifying assimilation rate for POCP production**

`fac_dop_assim =       0.5`

**factor modifying assimilation rate for POCN production**

---

---





---

Constants, continued from previous page

---

```
fac_don_assim =        1.0
```

**recycling rate (doc to dic) at 0°C [1/day]**
```
r_doc_rec =            0.001
```

---

end of table **Constants**

---

---

**Process limitation factors**

---

```
lim_t_n2_7 =          theta(t_n2-0.0)

lim_t_o2_0 =          1.0-exp(-t_o2/o2_min_det_resp)

lim_t_o2_2 =          theta(t_o2-0.0)

lim_t_dic_8 =         theta(t_dic-0.0)

lim_t_nh4_11 =        theta(t_nh4-0.0)

lim_t_no3_9 =         theta(t_no3-0.0)

lim_t_po4_10 =        theta(t_po4-0.0)

lim_t_spp_16 =        theta(t_spp-0.0)

lim_t_lip_18 =        theta(t_lip-0.0)

lim_t_doc_29 =        theta(t_doc-0.0)

lim_t_lpp_15 =        theta(t_lpp-0.0)

lim_t_cya_17 =        theta(t_cya-0.0)
```

---

---





| Process limitation factors, continued from previous page |
|---|
| lim_t_poc_12 =        theta(t_poc-0.0) |
| end of table **Process limitation factors** |

## B3.6   Process type BGC/pelagic/reoxidation

| **Processes** |
|---|
| **nitrification [mol/kg/day]** |
| t_nh4 + 2*t_o2 + h2o -> t_no3 + 2*h3oplus |
| p_nh4_nit_no3 =        (t_nh4*r_nh4_nitrif*exp(q10_nit*cgt_temp))*lim_t_nh4_11* lim_t_o2_2 |
| |
| **oxidation of hydrogen sulfide with oxygen [mol/kg/day]** |
| 0.5*t_o2 + t_h2s -> h2o + t_sul |
| p_h2s_oxo2_sul =        (t_h2s*t_o2*k_h2s_o2*exp(q10_h2s*cgt_temp))*lim_t_o2_2* lim_t_h2s_24 |
| |
| **oxidation of hydrogen sulfide with nitrate [mol/kg/day]** |
| t_h2s + 0.4*t_no3 + 0.4*h3oplus -> t_sul + 1.6*h2o + 0.2*t_n2 |
| p_h2s_oxno3_sul =        (t_h2s*t_no3*k_h2s_no3*exp(q10_h2s*cgt_temp))*lim_t_h2s_24* lim_t_no3_9 |
| |
| **oxidation of elemental sulfur with oxygen [mol/kg/day]** |
| t_sul + 1.5*t_o2 + 3*h2o -> so4 + 2*h3oplus |
| p_sul_oxo2_so4 =        (t_sul*t_o2*k_sul_o2*exp(q10_h2s*cgt_temp))*lim_t_sul_25* lim_t_o2_2 |
| |
| **oxidation of elemental sulfur with nitrate [mol/kg/day]** |
| t_sul + 1.2*t_no3 + 1.2*h2o -> so4 + 0.8*h3oplus + 0.6*t_n2 |
| continued on next page... |





| Processes, continued from previous page |
| --- |
| `p_sul_oxno3_so4 =`      `(t_sul*t_no3*k_sul_no3*exp(q10_h2s*cgt_temp))*lim_t_sul_25*` `lim_t_no3_9` |

| end of table **Processes** |
| --- |

| **Auxiliary variables** |
| --- |
| end of table **Auxiliary variables** |

| **Constants** |
| --- |

**reaction constant h2s oxidation with no3 [kg/mol/day]**

`k_h2s_no3 =`      `800000.0`

**reaction constant h2s oxidation with o2 [kg/mol/day]**

`k_h2s_o2 =`      `800000.0`

**reaction constant sul oxidation with no3 [kg/mol/day]**

`k_sul_no3 =`      `20000.0`

**reaction constant sul oxidation with o2 [kg/mol/day]**

`k_sul_o2 =`      `20000.0`

**q10 rule factor for oxidation of h2s and sul [1/K]**

`q10_h2s =`      `0.0693`

**q10 rule factor for nitrification [1/K]**

`q10_nit =`      `0.11`

**nitrification rate at 0℃ [1/day]**

`r_nh4_nitrif =`      `0.05`

| end of table **Constants** |
| --- |





| Process limitation factors | |
| --- | --- |
| lim_t_o2_2 = | theta(t_o2-0.0) |
| lim_t_nh4_11 = | theta(t_nh4-0.0) |
| lim_t_no3_9 = | theta(t_no3-0.0) |
| lim_t_h2s_24 = | theta(t_h2s-0.0) |
| lim_t_sul_25 = | theta(t_sul-0.0) |
| end of table **Process limitation factors** | |

## B3.7   Process type BGC/pelagic/zooplankton

| Processes |
| --- |
| **grazing of zooplankton eating large-cell phytoplankton [mol/kg/day]** |
| t_lpp -> t_zoo |
| p_lpp_graz_zoo =     ((t_zoo+zoo0)*lr_graz_zoo*t_lpp/max(food_zoo,epsilon))*<br>lim_t_lpp_15 |
| **grazing of zooplankton eating small-cell phytoplankton [mol/kg/day]** |
| t_spp -> t_zoo |
| p_spp_graz_zoo =     ((t_zoo+zoo0)*lr_graz_zoo*t_spp/max(food_zoo,epsilon))*<br>lim_t_spp_16 |
| **grazing of zooplankton eating diazotroph cyanobacteria [mol/kg/day]** |
| t_cya -> t_zoo |
| p_cya_graz_zoo =     ((t_zoo+zoo0)*lr_graz_zoo*(0.5*t_cya)/max(food_zoo,epsilon))*<br>lim_t_cya_17 |
| continued on next page... |





---

Processes, continued from previous page

---

**grazing of zooplankton eating limnic phytoplankton [mol/kg/day]**

```
t_lip -> t_zoo
p_lip_graz_zoo =    ((t_zoo+zoo0)*lr_graz_zoo*t_lip/max(food_zoo,epsilon))*
                    lim_t_lip_18
```

**respiration of zooplankton [mol/kg/day]**

```
0.8125*h3oplus + 6.625*t_o2 + t_zoo -> 7.4375*h2o + rfr_c*t_dic + rfr_p*t_po4 + (1-
don_fraction)*t_nh4 + don_fraction*t_don
p_zoo_resp_nh4 =    (zoo_eff*r_zoo_resp)*lim_t_o2_2*lim_t_zoo_19
```

**mortality of zooplankton [mol/kg/day]**

```
t_zoo -> t_det
p_zoo_mort_det =    (zoo_eff*r_zoo_mort*(1+9*theta(5.0e-6-t_o2)))*lim_t_zoo_19
```

---

end of table **Processes**

---

---

**Auxiliary variables**

---

**square of positive temperature [°C * °C]**

```
temp_sq =           max(0.0,cgt_temp)*max(0.0,cgt_temp)
```

**effectice zooplankton concentration assumed for mortality and respiration process [mol/kg]**

```
zoo_eff =           t_zoo*t_zoo/zoo_cl
```

**suitable food for zooplankton (weighted with food preferences) [mol/kg]**

```
food_zoo =          t_lpp+t_spp+t_lip+0.5*t_cya
```

**growth rate of zooplankton, limited by food, oxygen and temperature [1/day]**

```
lr_graz_zoo =       r_zoo_graz*(1-exp(-food_zoo*food_zoo/(food_min_zoo*food_min_zoo))
                    )*theta(t_o2-2*t_h2s)*(1.0+temp_sq/(temp_opt_zoo*temp_opt_zoo)*
                    exp(2.0-cgt_temp*2.0/temp_opt_zoo))
```

---

end of table **Auxiliary variables**

---





| Constants |
|---|
| **no division by 0** |
| epsilon =            4.5E-17 |
| |
| **Ivlev phytoplankton concentration for zooplankton grazing [mol/kg]** |
| food_min_zoo =       4.108E-6 |
| |
| **maximum zooplankton grazing rate [1/day]** |
| r_zoo_graz =         0.5 |
| |
| **mortality rate of zooplankton [1/day]** |
| r_zoo_mort =         0.03 |
| |
| **respiration rate of zooplankton [1/day]** |
| r_zoo_resp =         0.01 |
| |
| **redfield ratio C/N** |
| rfr_c =              6.625 |
| |
| **redfield ratio P/N** |
| rfr_p =              0.0625 |
| |
| **optimal temperature for zooplankton grazing [°C]** |
| temp_opt_zoo =       20.0 |
| |
| **seed concentration for zooplankton [mol/kg]** |
| zoo0 =               4.5E-9 |
| |
| **zooplankton closure parameter [mol/kg]** |
| zoo_cl =             9.0E-8 |
| |
| **fraction of DON in respiration products** |
| don_fraction =       0.0 |





| Constants, continued from previous page |
| --- |

| end of table **Constants** |
| --- |

| **Process limitation factors** |
| --- |
| `lim_t_o2_2 =`    `theta(t_o2-0.0)` |
| `lim_t_spp_16 =`    `theta(t_spp-0.0)` |
| `lim_t_zoo_19 =`    `theta(t_zoo-0.0)` |
| `lim_t_lip_18 =`    `theta(t_lip-0.0)` |
| `lim_t_lpp_15 =`    `theta(t_lpp-0.0)` |
| `lim_t_cya_17 =`    `theta(t_cya-0.0)` |

| end of table **Process limitation factors** |
| --- |

## B3.8   Process type gas_exchange

| **Processes** |
| --- |
| end of table **Processes** |

| **Auxiliary variables** |
| --- |
| **absolute temperature [K]** |
| `temp_k =`    `cgt_temp + 273.15` |
| **temporary value assumed for pH [1]** |





---

Auxiliary variables, continued from previous page

---

`ph_temp =`            `0.0-log(h3o)/log(10.0)`

calculated iteratively, 10 iterations, initial value = 0.0

**self-ionization constant of Water [mol2/kg2]**

`k_water =`            `exp( -13847.26 / temp_k + 148.96502 - 23.6521 * log(temp_k) +`

`(118.67/temp_k - 5.977 + 1.0495 * log(temp_k)) * sqrt(cgt_sali) -`

`0.01615 * cgt_sali)`

**Solubility of CO2 [mol/kg/Pa]**

`k0_co2 =`            `exp(9345.17 / temp_k - 60.2409 + 23.3585 * (log(temp_k) -`

`4.605170186) + cgt_sali*(0.023517 - 0.00023656 * temp_k +`

`0.00000047036 *temp_k*temp_k))/101325.0`

**Acid dissociation constant CO2 + 2 H2O <-> HCO3- + H3O+ [mol/kg]**

`k1_co2 =`            `power(10.0,( -3633.86 / temp_k + 61.2172 - 9.6777 * log(temp_k) +`

`0.011555 * cgt_sali - 0.0001152 * cgt_sali * cgt_sali))`

**Acid dissociation constant HCO3- + H2O <-> [CO3 2-] + H3O+ [mol/kg]**

`k2_co2 =`            `power(10.0,( -471.78 / temp_k - 25.929 + 3.16967 * log(temp_k) +`

`0.01781 * cgt_sali - 0.0001122 * cgt_sali * cgt_sali))`

**Acid dissociation constant of boric acid [mol/kg]**

`k_boron =`            `exp(( -8966.9 - 2890.53*sqrt(cgt_sali) - 77.942*cgt_sali + 1.728*`

`cgt_sali*sqrt(cgt_sali) - 0.0996*cgt_sali*cgt_sali) / temp_k +`

`148.0248 + 137.1942*sqrt(cgt_sali) + 1.62142*cgt_sali + (-24.4344`

`- 25.085*sqrt(cgt_sali) - 0.2474*cgt_sali)*log(temp_k) +`

`0.053105*sqrt(cgt_sali)*temp_k )`

**Acid dissociation constant H3PO4 + H2O <-> [H2PO4 -] + H3O+ [mol/kg]**

`k1_po4 =`            `exp( -4576.752/temp_k + 115.525 - 18.453*log(temp_k) + (0.69171 -`

`106.736/temp_k)*sqrt(cgt_sali) - (0.01844 + 0.65643/temp_k)*`

`cgt_sali )`

---

---





---

Auxiliary variables, continued from previous page

---

**Acid dissociation constant [H2PO4 -] + H2O+ <-> [HPO4 2-] + H3O+ [mol/kg]**

```
k2_po4 =            exp( -8814.715/temp_k + 172.0883 - 27.927*log(temp_k) + (1.35660
                    - 160.340/temp_k)*sqrt(cgt_sali) - (0.05778 - 0.37335/temp_k)*
                    cgt_sali )
```

**Acid dissociation constant [HPO4 2-] + H2O <-> [PO4 3-] + H3O+ [mol/kg]**

```
k3_po4 =            exp( -3070.75/temp_k - 18.141 + (2.81197 + 17.27039/temp_k)*
                    sqrt(cgt_sali) - (0.09984 + 44.99486/temp_k)*cgt_sali )
```

**Acid dissociation constant H2S + H2O <-> HS- + H3O+ [mol/kg]**

```
k1_h2s =            exp( -3131.42/temp_k + 5.818 + 0.368*(power(max(0.0,cgt_sali)
                    ,(1.0/3.0))))
```

**total concentration of boron [mol/kg]**

```
boron_total =       0.000416 * cgt_sali/35.0
```

**boron alkalinity [mol/kg]**

```
alk_boron =         boron_total * k_boron / (k_boron + h3o)
```
calculated iteratively, 10 iterations, initial value = 0.0

**hydrogen sulfide alkalinity [mol/kg]**

```
alk_h2s =           t_h2s * k1_h2s / (k1_h2s + h3o)
```
calculated iteratively, 10 iterations, initial value = 0.0

**water alkalinity [mol/kg]**

```
alk_water =         k_water / h3o - h3o
```
calculated iteratively, 10 iterations, initial value = 0.0

**denominator in phosphate alkalinity formula [mol3/kg3]**

```
alk_po4_denominator = (h3o*h3o*h3o + k1_po4*h3o*h3o + k1_po4*k2_po4*h3o + k1_po4*
                    k2_po4*k3_po4)
```

---

---





---

Auxiliary variables, continued from previous page

---

calculated iteratively, 10 iterations, initial value = 0.0

**phosphate alkalinity [mol/kg]**

alk_po4 =               t_po4*(k1_po4*k2_po4*h3o + 2.0*k1_po4*k2_po4*k3_po4 - h3o*h3o*
                        h3o) / alk_po4_denominator

calculated iteratively, 10 iterations, initial value = 0.0

**denominator in carbonate alkalinity formula [mol2/kg2]**

alk_co2_denominator = (h3o*h3o + k1_co2*h3o + k1_co2*k2_co2)

calculated iteratively, 10 iterations, initial value = 0.0

**carbonate alkalinity [mol/kg]**

alk_co2 =               t_dic*k1_co2*(h3o+2*k2_co2)/alk_co2_denominator

calculated iteratively, 10 iterations, initial value = 0.0

**error in total alkalinity calculation at the assumed pH [mol/kg]**

alk_residual =       t_alk - alk_co2 - alk_po4 - alk_boron - alk_h2s - alk_water

calculated iteratively, 10 iterations, initial value = 0.0

**derivative of phosphate alkalinity with respect to h3o [1]**

dalkp_dh3o =            t_po4*(0.0-k1_po4*h3o*h3o*h3o*h3o-4*k1_po4*k2_po4*h3o*h3o*h3o-
                        (k1_po4*k1_po4*k2_po4+9*k1_po4*k2_po4*k3_po4)*h3o*h3o-4*k1_po4*
                        k1_po4*k2_po4*k3_po4*h3o-k1_po4*k1_po4*k2_po4*k2_po4*k3_po4)
                        /(alk_po4_denominator*alk_po4_denominator)

calculated iteratively, 10 iterations, initial value = 0.0

**derivative of carbonate alkalinity with respect to h3o [1]**

dalkc_dh3o =            t_dic*(0.0-k1_co2*h3o*h3o-k1_co2*k1_co2*k2_co2-4*k1_co2*k2_co2*
                        h3o)/(alk_co2_denominator*alk_co2_denominator)

calculated iteratively, 10 iterations, initial value = 0.0

**derivative of residual_alk with respect to pH [mol/kg]**

---

continued on next page...

---



---

Auxiliary variables, continued from previous page

---

```
dalkresidual_dpH =    0.0-log(10.0)*h3o*(alk_boron/(k_boron+h3o)+alk_h2s/(k1_h2s+h3o)+
                      k_water/(h3o*h3o)+1-dalkp_dh3o-dalkc_dh3o)
```

calculated iteratively, 10 iterations, initial value = 0.0

**newly determined pH value [1]**

```
temp1 =               alk_residual/dalkresidual_dpH
ph =                  ph_temp - temp1 + theta(abs(temp1) - 1)*0.5*temp1
```

calculated iteratively, 10 iterations, initial value = 0.0

**h3o ion concentration [mol/kg]**

```
h3o =                 power(10.0,0.0-max(1.0,min(13.0,ph)))
```

calculated iteratively, 10 iterations, initial value = 1.0e-8

**co2 partial pressure [Pa]**

```
pco2 =                t_dic / k0_co2 / (1 + k1_co2/h3o + k1_co2*k2_co2/h3o/h3o)
```

**oxygen saturation concentration [mol/kg]**

```
o2_sat =              (10.18e0+((5.306e-3-4.8725e-5*cgt_temp)*cgt_temp-0.2785e0)*
                      cgt_temp+cgt_sali*((2.2258e-3+(4.39e-7*cgt_temp-4.645e-5)*
                      cgt_temp)*cgt_temp-6.33e-2))*44.66e0*1e-6
```

**dissolved molecular nitrogen saturation concentration [mol/kg]**

```
temp1 =               log((298.15-cgt_temp)/(273.15+cgt_temp))
temp2 =               temp1*temp1
temp3 =               temp2*temp1
n2_sat =              1e-6*exp(6.42931 + 2.92704*temp1 + 4.32531*temp2 + 4.69149*temp3
                      + cgt_sali*(0.0 -7.44129e-3 - 8.02566e-3*temp1 - 1.46775e-2*
                      temp2))
```

---

end of table **Auxiliary variables**

---





| Constants |
| --- |
| **atmospheric partial pressure of CO2 [Pa]** |
| patm_co2 = 38.0 |
| **piston velocity for co2 surface flux [m/d]** |
| w_co2_stf = 4.0 |
| **piston velocity for n2 surface flux [m/d]** |
| w_n2_stf = 5.0 |
| **piston velocity for oxygen surface flux [m/d]** |
| w_o2_stf = 5.0 |
| end of table **Constants** |

| Process limitation factors |
| --- |
| lim_t_n2_7 =        theta(t_n2-0.0) |
| lim_t_o2_2 =        theta(t_o2-0.0) |
| lim_t_dic_8 =       theta(t_dic-0.0) |
| end of table **Process limitation factors** |

5

## B3.9   Process type physics/erosion

| Processes |
| --- |
| **sedimentary detritus erosion (sediment only) [mol/m$^2$/day]** |
| t_sed -> t_det |
| p_sed_ero_det =        (erosion_is_active*r_sed_ero*sed_active)*lim_t_sed_21 |
| continued on next page... |





---

Processes, continued from previous page

---

**erosion of iron PO4 (sediment only) [mol/m$^2$/day]**

```
t_ips -> t_ipw
```

```
p_ips_ero_ipw =        (erosion_is_active*r_ips_ero*t_ips)*lim_t_ips_23
```

**sedimentary poc erosion (sediment only) [mol/m$^2$/day]**

```
t_sed_poc -> t_poc
```

```
p_sed_ero_poc =        (erosion_is_active*r_sed_ero*poc_active)*lim_t_sed_poc_22
```

**sedimentary pocn erosion (sediment only) [mol/m$^2$/day]**

```
t_sed_pocn -> t_pocn
```

```
p_sed_ero_pocn =        (erosion_is_active*r_sed_ero*pocn_active)*lim_t_sed_pocn_27
```

**sedimentary pocp erosion (sediment only) [mol/m$^2$/day]**

```
t_sed_pocp -> t_pocp
```

```
p_sed_ero_pocp =        (erosion_is_active*r_sed_ero*pocp_active)*lim_t_sed_pocp_28
```

---

end of table **Processes**

---

---

**Auxiliary variables**

---

**total carbon in sediment layer [mol/m\*\*2]**

```
sed_tot =              t_sed*rfr_c + t_sed_poc + t_sed_pocn*rfr_c + t_sed_pocp*rfr_cp
```

**total carbon in active sediment layer [mol/m\*\*2]**

```
sed_tot_active =       max(0.0,min(sed_tot,sed_max*rfr_c))
```

**detritus in active sediment layer [mol/m\*\*2]**

```
sed_active =           sed_tot_active * t_sed/sed_tot
```

**switch (1=erosion, 0=no erosion) which depends on the combined bottom stress of currents and waves**

---

---





---

Auxiliary variables, continued from previous page

---

```
erosion_is_active =   theta(cgt_current_wave_stress - critical_stress)
```

**poc in active sediment layer [mol/m\*\*2]**

```
poc_active =          sed_tot_active * t_sed_poc/sed_tot
```

**pocn in active sediment layer [mol/m\*\*2]**

```
pocn_active =         sed_tot_active * t_sed_pocn/sed_tot
```

**pocp in active sediment layer [mol/m\*\*2]**

```
pocp_active =         sed_tot_active * t_sed_pocp/sed_tot
```

---

end of table Auxiliary variables

---

---

Constants

---

**critical shear stress for sediment erosion [N/m2]**

```
critical_stress =   0.016
```

**erosion rate for iron PO4 [1/day]**

```
r_ips_ero =         6.0
```

**maximum sediment detritus erosion rate [1/day]**

```
r_sed_ero =         6.0
```

**redfield ratio C/N**

```
rfr_c =             6.625
```

**redfield ratio C/P**

```
rfr_cp =            106.0
```

**maximum sediment detritus concentration that feels erosion [mol/m\*\*2]**

```
sed_max =           1.0
```

---

---





---

Constants, continued from previous page

---

end of table **Constants**

---

**Process limitation factors**

---

`lim_t_sed_21 =`           `theta(t_sed-0.0)`

`lim_t_ips_23 =`           `theta(t_ips-0.0)`

`lim_t_sed_poc_22 =`     `theta(t_sed_poc-0.0)`

`lim_t_sed_pocn_27 =`    `theta(t_sed_pocn-0.0)`

`lim_t_sed_pocp_28 =`    `theta(t_sed_pocp-0.0)`

---

end of table **Process limitation factors**

---

## B3.10   Process type physics/parametrization_deep_burial

---

**Processes**

---

**burial of detritus deeper than max_sed (sediment only) [mol/m$^2$/day]**

`t_sed ->`

`p_sed_burial =`           `((sed_tot-sed_tot_burial)/cgt_timestep*t_sed/sed_tot)*`
                           `lim_t_sed_21`

**burial of iron PO4 (sediment only) [mol/m$^2$/day]**

`t_ips ->`

`p_ips_burial =`           `(fac_ips_burial*(sed_tot-sed_tot_burial)/cgt_timestep*`
                           `t_ips/sed_tot)*lim_t_ips_23`

---

continued on next page...

---





---

Processes, continued from previous page

---

**burial of poc deeper than max_sed (sediment only) $[\text{mol/m}^2/\text{day}]$**

`t_sed_poc ->`

`p_poc_burial =`      `((sed_tot-sed_tot_burial)/cgt_timestep*t_sed_poc/sed_tot)*`

                  `lim_t_sed_poc_22`

**burial of pocn deeper than max_sed (sediment only) $[\text{mol/m}^2/\text{day}]$**

`t_sed_pocn ->`

`p_pocn_burial =`      `((sed_tot-sed_tot_burial)/cgt_timestep*t_sed_pocn/sed_tot)*`

                  `lim_t_sed_pocn_27`

**burial of pocp deeper than max_sed (sediment only) $[\text{mol/m}^2/\text{day}]$**

`t_sed_pocp ->`

`p_pocp_burial =`      `((sed_tot-sed_tot_burial)/cgt_timestep*t_sed_pocp/sed_tot)*`

                  `lim_t_sed_pocp_28`

---

end of table **Processes**

---

---

**Auxiliary variables**

---

**total carbon in sediment layer [mol/m\*\*2]**

`sed_tot =`      `t_sed*rfr_c + t_sed_poc + t_sed_pocn*rfr_c + t_sed_pocp*rfr_cp`

**total carbon in sediment layer before burial [mol/m\*\*2]**

`sed_tot_burial =`      `max(0.0,min(sed_tot,sed_burial*rfr_c))`

---

end of table **Auxiliary variables**

---

---

**Constants**

---

**redfield ratio C/N**

`rfr_c =`      `6.625`

---

---





| Constants, continued from previous page |
|---|

**redfield ratio C/P**

`rfr_cp =`             `106.0`

**maximum sediment load before burial**

`sed_burial =`        `1.0`

**reduced burial of t_ips, mimicing resolving iron-P complexes in deeper sediment and subsequent upward PO4 flux**

`fac_ips_burial =`    `0.5`

| end of table **Constants** |
|---|

| Process limitation factors |
|---|

`lim_t_sed_21 =`       `theta(t_sed-0.0)`

`lim_t_ips_23 =`       `theta(t_ips-0.0)`

`lim_t_sed_poc_22 =`   `theta(t_sed_poc-0.0)`

`lim_t_sed_pocn_27 =`  `theta(t_sed_pocn-0.0)`

`lim_t_sed_pocp_28 =`  `theta(t_sed_pocp-0.0)`

| end of table **Process limitation factors** |
|---|

**B3.11   Process type physics/sedimentation**

| Processes |
|---|

**detritus sedimentation (sediment only) [mol/m$^2$/day]**





---

Processes, continued from previous page

---

```
t_det -> t_sed
p_det_sedi_sed =    ((1.0-erosion_is_active)*(0.0-w_det_sedi)*t_det*cgt_density)*
                    lim_t_det_20
```

**sedimentation of iron PO4 (sediment only) [mol/m²/day]**

```
t_ipw -> t_ips
p_ipw_sedi_ips =    ((1.0-erosion_is_active)*(0.0-w_ipw_sedi)*t_ipw*cgt_density)*
                    lim_t_ipw_26
```

**poc sedimentation (sediment only) [mol/m²/day]**

```
t_poc -> t_sed_poc
p_poc_sedi_sed =    ((1.0-erosion_is_active)*(0.0-w_poc_var)*t_poc*cgt_density)*
                    lim_t_poc_12
```

**pocn sedimentation (sediment only) [mol/m²/day]**

```
t_pocn -> t_sed_pocn
p_pocn_sedi_sed =    ((1.0-erosion_is_active)*(0.0-w_pocn_sedi)*t_pocn*cgt_density)*
                     lim_t_pocn_14
```

**pocp sedimentation (sediment only) [mol/m²/day]**

```
t_pocp -> t_sed_pocp
p_pocp_sedi_sed =    ((1.0-erosion_is_active)*(0.0-w_pocp_sedi)*t_pocp*cgt_density)*
                     lim_t_pocp_13
```

---

end of table <b>Processes</b>

---

<b>Auxiliary variables</b>

---

**switch (1=erosion, 0=no erosion) which depends on the combined bottom stress of currents and waves**

```
erosion_is_active =   theta(cgt_current_wave_stress - critical_stress)
```

---

continued on next page...

---





| Auxiliary variables, continued from previous page |
| --- |

**depth dependent POC sinking speed**

`w_poc_var =`        `martin_fac_poc * cgt_bottomdepth * (-1.0)`

| end of table **Auxiliary variables** |
| --- |

| **Constants** |
| --- |

**critical shear stress for sediment erosion [N/m2]**

`critical_stress =`      `0.016`

**sedimentation velocity (negative for downward) [m/day]**

`w_det_sedi =`      `-2.25`

**sedimentation velocity for iron PO4 [m/day]**

`w_ipw_sedi =`      `-0.5`

**sedimentation velocity (negative for downward) [m/day]**

`w_pocp_sedi =`      `-0.05`

**sedimentation velocity (negative for downward) [m/day]**

`w_pocn_sedi =`      `-0.05`

**[1/d], depth dependence of POC sinking speed**

`martin_fac_poc =`      `0.01`

| end of table **Constants** |
| --- |

| **Process limitation factors** |
| --- |

`lim_t_ipw_26 =`      `theta(t_ipw-0.0)`

`lim_t_det_20 =`      `theta(t_det-0.0)`





| Process limitation factors, continued from previous page |
|---|

| | |
|---|---|
| lim_t_poc_12 = | theta(t_poc-0.0) |
| lim_t_pocp_13 = | theta(t_pocp-0.0) |
| lim_t_pocn_14 = | theta(t_pocn-0.0) |

| end of table **Process limitation factors** |
|---|

## B3.12  Process type standard

| **Processes** |
|---|

**particle formation from DOC [mol/kg/day]**

t_doc -> t_poc

| | |
|---|---|
| p_doc2pco = | (t_doc * r_doc2poc)*lim_t_doc_29 |

**particle formation from DOP [mol/kg/day]**

t_dop -> t_pocp

| | |
|---|---|
| p_dop2pocp = | (t_dop * r_dop2pocp)*lim_t_dop_30 |

**particle formation from DON [mol/kg/day]**

t_don -> t_pocn

| | |
|---|---|
| p_don2pocn = | (t_don * r_don2pocn)*lim_t_don_31 |

| end of table **Processes** |
|---|

| **Auxiliary variables** |
|---|
| end of table **Auxiliary variables** |





| Constants | |
|---|---|
| **POC formation rate** | |
| r_doc2poc = | 0.01 |
| **POCN formation rate** | |
| r_don2pocn = | 0.01 |
| **POCP formation rate** | |
| r_dop2pocp = | 0.01 |
| end of table **Constants** | |

| Process limitation factors | |
|---|---|
| lim_t_doc_29 = | theta(t_doc-0.0) |
| lim_t_dop_30 = | theta(t_dop-0.0) |
| lim_t_don_31 = | theta(t_don-0.0) |
| end of table **Process limitation factors** | |

## B4 Tracer equations

| Tracer equations | |
|---|---|
| **Change of: dissolved molecular nitrogen** | |
| $\frac{d}{dt}$ t_n2 = | |
| + (p_poc_denit)*(0.4) | recycling of POC using nitrate (denitrification) |
| + (p_pocp_denit)*(42.4) | recycling of POC using nitrate (denitrification) |
| + (p_pocn_denit)*(2.65) | recycling of POCN using nitrate (denitrification) |
| continued on next page... | |





---

Tracer equations, continued from previous page

---

+ (p_det_denit_nh4)*(2.65)          recycling of detritus using nitrate (denitrification)

+ (p_nh4_nitdenit_n2)*(0.5)         coupled nitrification and denitrification after
/(cgt_cellheight*cgt_density)       mineralization of detritus in oxic sediments

+ (p_sed_denit_nh4)*(2.65)          recycling of sedimentary detritus to ammonium
/(cgt_cellheight*cgt_density)       using nitrate (denitrification)

+ (p_sed_poc_denit)*(0.4)           recycling of sedimentary poc to dic using nitrate
/(cgt_cellheight*cgt_density)       (denitrification)

+ (p_h2s_oxno3_sul)*(0.2)           oxidation of hydrogen sulfide with nitrate

+ (p_sul_oxno3_so4)*(0.6)           oxidation of elemental sulfur with nitrate

+ (p_sed_pocn_denit)*(2.65)         recycling of sedimentary pocn to dic and NH4 using
/(cgt_cellheight*cgt_density)       nitrate (denitrification)

+ (p_sed_pocp_denit)*(42.4)         recycling of sedimentary pocp to dic and PO4 using
/(cgt_cellheight*cgt_density)       nitrate (denitrification)

+ (p_nh4_nitdenit_pocn_n2)*(0.5)    coupled nitrification and denitrification after
/(cgt_cellheight*cgt_density)       mineralization of pocn-detritus in oxic sediments

+ (p_doc_denit)*(0.4)               recycling of DOC using nitrate (denitrification)

+ (p_dop_denit)*(42.4)              recycling of DOP using nitrate (denitrification)

+ (p_don_denit)*(2.65)              recycling of DON using nitrate (denitrification)

− (p_n2_assim_cya)*(0.5)            fixation of dinitrogen by diazotroph cyanobacteria

---

---





---

Tracer equations, continued from previous page

---

**Change of: dissolved oxygen**

$\frac{d}{dt}$ t_o2 =

| | |
|---|---|
| + (p_no3_assim_lpp)*(8.625) | assimilation of nitrate by large-cell phytoplankton |
| + (p_nh4_assim_lpp)*(6.625) | assimilation of ammonium by large-cell phytoplankton |
| + (p_no3_assim_spp)*(8.625) | assimilation of nitrate by small-cell phytoplankton |
| + (p_nh4_assim_spp)*(6.625) | assimilation of ammonium by small-cell phytoplankton |
| + (p_nh4_assim_lip)*(6.625) | assimilation of ammonium by limnic phytoplankton |
| + (p_no3_assim_lip)*(8.625) | assimilation of nitrate by limnic phytoplankton |
| + (p_n2_assim_cya)*(7.375) | fixation of dinitrogen by diazotroph cyanobacteria |
| + p_assim_lpp_doc | Production of DOC by LPP |
| + p_assim_spp_doc | Production of DOC by SPP |
| + p_assim_lip_doc | Production of DOC by LPP |
| + p_assim_cya_doc | Production of DOC by CYA |
| + (p_assim_lpp_dop)*(106) | Production of DOP by LPP |
| + (p_assim_spp_dop)*(106) | Production of DOP by SPP |

---

continued on next page...

---



| Tracer equations, continued from previous page | |
| --- | --- |
| `+ (p_assim_lip_dop)*(106)` | Production of DOP by LIP |
| `+ (p_nh4_assim_lpp_don)*(6.625)` | Production of DON by LPP |
| `+ (p_no3_assim_lpp_don)*(8.625)` | Production of DON by LPP |
| `+ (p_nh4_assim_spp_don)*(6.625)` | Production of DON by SPP |
| `+ (p_no3_assim_spp_don)*(8.625)` | Production of DON by SPP |
| `+ (p_nh4_assim_lip_don)*(6.625)` | Production of DON by LIP |
| `+ (p_no3_assim_lip_don)*(8.625)` | Production of DON by LIP |
| `- p_poc_resp` | respiration of POC |
| `- (p_pocp_resp)*(106)` | respiration of POCP |
| `- (p_pocn_resp)*(6.625)` | respiration of POCN |
| `- (p_lpp_resp_nh4)*(6.625)` | respiration of large-cell phytoplankton |
| `- (p_spp_resp_nh4)*(6.625)` | respiration of small-cell phytoplankton |
| `- (p_lip_resp_nh4)*(6.625)` | respiration of limnic phytoplankton |
| `- (p_cya_resp_nh4)*(6.625)` | respiration of diazotroph cyanobacteria |
| `- (p_zoo_resp_nh4)*(6.625)` | respiration of zooplankton |
| `- (p_nh4_nit_no3)*(2)` | nitrification |





---

| Tracer equations, continued from previous page | |
|---|---|
| `- (p_det_resp_nh4)*(6.625)` | recycling of detritus using oxygen (respiration) |
| `- (p_sed_resp_nh4)*(6.625)` `/(cgt_cellheight*cgt_density)` | recycling of sedimentary detritus to ammonium using oxygen (respiration) |
| `- (p_nh4_nitdenit_n2)*(0.75)` `/(cgt_cellheight*cgt_density)` | coupled nitrification and denitrification after mineralization of detritus in oxic sediments |
| `-` `p_sed_poc_resp/(cgt_cellheight*` `cgt_density)` | recycling of sedimentary poc to dic using oxygen (respiration) |
| `- (p_h2s_oxo2_sul)*(0.5)` | oxidation of hydrogen sulfide with oxygen |
| `- (p_sul_oxo2_so4)*(1.5)` | oxidation of elemental sulfur with oxygen |
| `- (p_sed_pocn_resp)*(6.625)` `/(cgt_cellheight*cgt_density)` | recycling of sedimentary pocn to dic and NH4 using oxygen (respiration) |
| `- (p_sed_pocp_resp)*(106)` `/(cgt_cellheight*cgt_density)` | recycling of sedimentary pocp to dic and PO4 using oxygen (respiration) |
| `- (p_nh4_nitdenit_pocn_n2)*` `(0.75)/(cgt_cellheight*` `cgt_density)` | coupled nitrification and denitrification after mineralization of pocn-detritus in oxic sediments |
| `- p_doc_resp` | respiration of DOC |
| `- (p_dop_resp)*(106)` | respiration of DOP |
| `- (p_don_resp)*(6.625)` | respiration of DON |

---

continued on next page...





---

Tracer equations, continued from previous page

---

**Change of: dissolved inorganic carbon, treated as carbon dioxide**

$\frac{d}{dt}$ t_dic =

| | |
|---|---|
| + p_poc_resp | respiration of POC |
| + p_poc_denit | recycling of POC using nitrate (denitrification) |
| + p_poc_sulf | Mineralization of POC, e-acceptor sulfate (sulfate reduction) |
| + (p_pocp_resp)*(106) | respiration of POCP |
| + (p_pocp_denit)*(106) | recycling of POC using nitrate (denitrification) |
| + (p_pocp_sulf)*(106) | Mineralization of POC, e-acceptor sulfate (sulfate reduction) |
| + (p_pocn_resp)*(6.625) | respiration of POCN |
| + (p_pocn_denit)*(6.625) | recycling of POCN using nitrate (denitrification) |
| + (p_pocn_sulf)*(6.625) | Mineralization of POCN, e-acceptor sulfate (sulfate reduction) |
| + (p_lpp_resp_nh4)*(rfr_c) | respiration of large-cell phytoplankton |
| + (p_spp_resp_nh4)*(rfr_c) | respiration of small-cell phytoplankton |
| + (p_lip_resp_nh4)*(rfr_c) | respiration of limnic phytoplankton |
| + (p_cya_resp_nh4)*(rfr_c) | respiration of diazotroph cyanobacteria |

---





---

| Tracer equations, continued from previous page | |
|---|---|
| `+ (p_zoo_resp_nh4)*(rfr_c)` | respiration of zooplankton |
| `+ (p_det_resp_nh4)*(rfr_c)` | recycling of detritus using oxygen (respiration) |
| `+ (p_det_denit_nh4)*(rfr_c)` | recycling of detritus using nitrate (denitrification) |
| `+ (p_det_sulf_nh4)*(rfr_c)` | recycling of detritus using sulfate (sulfate reduction) |
| `+ (p_sed_resp_nh4)*(rfr_c) /(cgt_cellheight*cgt_density)` | recycling of sedimentary detritus to ammonium using oxygen (respiration) |
| `+ (p_sed_denit_nh4)*(rfr_c) /(cgt_cellheight*cgt_density)` | recycling of sedimentary detritus to ammonium using nitrate (denitrification) |
| `+ (p_sed_sulf_nh4)*(rfr_c) /(cgt_cellheight*cgt_density)` | recycling of sedimentary detritus to ammonium using sulfate (sulfate reduction) |
| `+ p_sed_poc_resp/(cgt_cellheight* cgt_density)` | recycling of sedimentary poc to dic using oxygen (respiration) |
| `+ p_sed_poc_denit/(cgt_cellheight* cgt_density)` | recycling of sedimentary poc to dic using nitrate (denitrification) |
| `+ p_sed_poc_sulf/(cgt_cellheight* cgt_density)` | recycling of sedimentary poc to dic using sulfate (sulfate reduction) |
| `+ (p_sed_pocn_resp)*(6.625) /(cgt_cellheight*cgt_density)` | recycling of sedimentary pocn to dic and NH4 using oxygen (respiration) |

---





| Tracer equations, continued from previous page | |
|---|---|
| `+ (p_sed_pocp_resp)*(106)` `/(cgt_cellheight*cgt_density)` | recycling of sedimentary pocp to dic and PO4 using oxygen (respiration) |
| `+ (p_sed_pocn_denit)*(6.625)` `/(cgt_cellheight*cgt_density)` | recycling of sedimentary pocn to dic and NH4 using nitrate (denitrification) |
| `+ (p_sed_pocp_denit)*(106)` `/(cgt_cellheight*cgt_density)` | recycling of sedimentary pocp to dic and PO4 using nitrate (denitrification) |
| `+ (p_sed_pocn_sulf)*(6.625)` `/(cgt_cellheight*cgt_density)` | recycling of sedimentary pocn to dic and NH4 using sulfate (sulfate reduction) |
| `+ (p_sed_pocp_sulf)*(106)` `/(cgt_cellheight*cgt_density)` | recycling of sedimentary pocp to dic and PO4 using sulfate (sulfate reduction) |
| `+ p_doc_resp` | respiration of DOC |
| `+ p_doc_denit` | recycling of DOC using nitrate (denitrification) |
| `+ p_doc_sulf` | Mineralization of DOC, e-acceptor sulfate (sulfate reduction) |
| `+ (p_dop_resp)*(106)` | respiration of DOP |
| `+ (p_dop_denit)*(106)` | recycling of DOP using nitrate (denitrification) |
| `+ (p_dop_sulf)*(106)` | Mineralization of DOP, e-acceptor sulfate (sulfate reduction) |
| `+ (p_don_resp)*(6.625)` | respiration of DON |
| `+ (p_don_denit)*(6.625)` | recycling of DON using nitrate (denitrification) |





---

Tracer equations, continued from previous page

---

| | |
|---|---|
| + (p_don_sulf)*(6.625) | Mineralization of DON, e-acceptor sulfate (sulfate reduction) |
| − (p_no3_assim_lpp)*(rfr_c) | assimilation of nitrate by large-cell phytoplankton |
| − (p_nh4_assim_lpp)*(rfr_c) | assimilation of ammonium by large-cell phytoplankton |
| − (p_no3_assim_spp)*(rfr_c) | assimilation of nitrate by small-cell phytoplankton |
| − (p_nh4_assim_spp)*(rfr_c) | assimilation of ammonium by small-cell phytoplankton |
| − (p_nh4_assim_lip)*(rfr_c) | assimilation of ammonium by limnic phytoplankton |
| − (p_no3_assim_lip)*(rfr_c) | assimilation of nitrate by limnic phytoplankton |
| − (p_n2_assim_cya)*(rfr_c) | fixation of dinitrogen by diazotroph cyanobacteria |
| − p_assim_lpp_doc | Production of DOC by LPP |
| − p_assim_spp_doc | Production of DOC by SPP |
| − p_assim_lip_doc | Production of DOC by LPP |
| − p_assim_cya_doc | Production of DOC by CYA |
| − (p_assim_lpp_dop)*(106) | Production of DOP by LPP |
| − (p_assim_spp_dop)*(106) | Production of DOP by SPP |

---

---





---

| Tracer equations, continued from previous page | |
|---|---|
| - (p_assim_lip_dop)*(106) | Production of DOP by LIP |
| - (p_nh4_assim_lpp_don)*(rfr_c) | Production of DON by LPP |
| - (p_no3_assim_lpp_don)*(rfr_c) | Production of DON by LPP |
| - (p_nh4_assim_spp_don)*(rfr_c) | Production of DON by SPP |
| - (p_no3_assim_spp_don)*(rfr_c) | Production of DON by SPP |
| - (p_nh4_assim_lip_don)*(rfr_c) | Production of DON by LIP |
| - (p_no3_assim_lip_don)*(rfr_c) | Production of DON by LIP |

**Change of: ammonium**

$\frac{d}{dt}$ t_nh4 =

| | |
|---|---|
| + p_pocn_resp | respiration of POCN |
| + p_pocn_denit | recycling of POCN using nitrate (denitrification) |
| + p_pocn_sulf | Mineralization of POCN, e-acceptor sulfate (sulfate reduction) |
| + (p_lpp_resp_nh4)*((1-don_fraction)) | respiration of large-cell phytoplankton |
| + (p_spp_resp_nh4)*((1-don_fraction)) | respiration of small-cell phytoplankton |
| + (p_lip_resp_nh4)*((1-don_fraction)) | respiration of limnic phytoplankton |





---

Tracer equations, continued from previous page

---

| | |
|---|---|
| `+ (p_cya_resp_nh4)*((1-don_fraction))` | respiration of diazotroph cyanobacteria |
| `+ (p_zoo_resp_nh4)*((1-don_fraction))` | respiration of zooplankton |
| `+ p_det_resp_nh4` | recycling of detritus using oxygen (respiration) |
| `+ p_det_denit_nh4` | recycling of detritus using nitrate (denitrification) |
| `+ p_det_sulf_nh4` | recycling of detritus using sulfate (sulfate reduction) |
| `+ p_sed_resp_nh4/(cgt_cellheight* cgt_density)` | recycling of sedimentary detritus to ammonium using oxygen (respiration) |
| `+ p_sed_denit_nh4/(cgt_cellheight* cgt_density)` | recycling of sedimentary detritus to ammonium using nitrate (denitrification) |
| `+ p_sed_sulf_nh4/(cgt_cellheight* cgt_density)` | recycling of sedimentary detritus to ammonium using sulfate (sulfate reduction) |
| `+ p_sed_pocn_resp/(cgt_cellheight* cgt_density)` | recycling of sedimentary pocn to dic and NH4 using oxygen (respiration) |
| `+ p_sed_pocn_denit/(cgt_cellheight* cgt_density)` | recycling of sedimentary pocn to dic and NH4 using nitrate (denitrification) |

---

---





---

Tracer equations, continued from previous page

---

| | |
|---|---|
| `+ p_sed_pocn_sulf/(cgt_cellheight* cgt_density)` | recycling of sedimentary pocn to dic and NH4 using sulfate (sulfate reduction) |
| `+ p_don_resp` | respiration of DON |
| `+ p_don_denit` | recycling of DON using nitrate (denitrification) |
| `+ p_don_sulf` | Mineralization of DON, e-acceptor sulfate (sulfate reduction) |
| `- p_nh4_assim_lpp` | assimilation of ammonium by large-cell phytoplankton |
| `- p_nh4_assim_spp` | assimilation of ammonium by small-cell phytoplankton |
| `- p_nh4_assim_lip` | assimilation of ammonium by limnic phytoplankton |
| `- p_nh4_assim_lpp_don` | Production of DON by LPP |
| `- p_nh4_assim_spp_don` | Production of DON by SPP |
| `- p_nh4_assim_lip_don` | Production of DON by LIP |
| `- p_nh4_nit_no3` | nitrification |
| `- p_nh4_nitdenit_n2/(cgt_cellheight* cgt_density)` | coupled nitrification and denitrification after mineralization of detritus in oxic sediments |

---

---



---

Tracer equations, continued from previous page

| | | |
|---|---|---|
| − | | coupled nitrification and denitrification after |
| p_nh4_nitdenit_pocn_n2/(cgt_cell* | | mineralization of pocn-detritus in oxic sediments |
| cgt_density) | | |

**Change of: nitrate**

$\frac{d}{dt}$ t_no3 =

| | |
|---|---|
| + p_nh4_nit_no3 | nitrification |
| − p_no3_assim_lpp | assimilation of nitrate by large-cell phytoplankton |
| − p_no3_assim_spp | assimilation of nitrate by small-cell phytoplankton |
| − p_no3_assim_lip | assimilation of nitrate by limnic phytoplankton |
| − p_no3_assim_lpp_don | Production of DON by LPP |
| − p_no3_assim_spp_don | Production of DON by SPP |
| − p_no3_assim_lip_don | Production of DON by LIP |
| − (p_poc_denit)*(0.8) | recycling of POC using nitrate (denitrification) |
| − (p_pocp_denit)*(84.8) | recycling of POC using nitrate (denitrification) |
| − (p_pocn_denit)*(5.3) | recycling of POCN using nitrate (denitrification) |
| − (p_det_denit_nh4)*(5.3) | recycling of detritus using nitrate (denitrification) |
| − (p_sed_denit_nh4)*(5.3) /(cgt_cellheight*cgt_density) | recycling of sedimentary detritus to ammonium using nitrate (denitrification) |

---

continued on next page...





---

Tracer equations, continued from previous page

| | |
|---|---|
| `- (p_sed_poc_denit)*(0.8)` `/(cgt_cellheight*cgt_density)` | recycling of sedimentary poc to dic using nitrate (denitrification) |
| `- (p_h2s_oxno3_sul)*(0.4)` | oxidation of hydrogen sulfide with nitrate |
| `- (p_sul_oxno3_so4)*(1.2)` | oxidation of elemental sulfur with nitrate |
| `- (p_sed_pocn_denit)*(5.3)` `/(cgt_cellheight*cgt_density)` | recycling of sedimentary pocn to dic and NH4 using nitrate (denitrification) |
| `- (p_sed_pocp_denit)*(84.8)` `/(cgt_cellheight*cgt_density)` | recycling of sedimentary pocp to dic and PO4 using nitrate (denitrification) |
| `- (p_doc_denit)*(0.8)` | recycling of DOC using nitrate (denitrification) |
| `- (p_dop_denit)*(84.8)` | recycling of DOP using nitrate (denitrification) |
| `- (p_don_denit)*(5.3)` | recycling of DON using nitrate (denitrification) |

**Change of: phosphate**

$\frac{d}{dt}$ `t_po4` =

| | |
|---|---|
| `+ p_pocp_resp` | respiration of POCP |
| `+ p_pocp_denit` | recycling of POC using nitrate (denitrification) |
| `+ p_pocp_sulf` | Mineralization of POC, e-acceptor sulfate (sulfate reduction) |
| `+ (p_lpp_resp_nh4)*(rfr_p)` | respiration of large-cell phytoplankton |
| `+ (p_spp_resp_nh4)*(rfr_p)` | respiration of small-cell phytoplankton |

continued on next page...

---





---

Tracer equations, continued from previous page

---

| | |
|---|---|
| `+ (p_lip_resp_nh4)*(rfr_p)` | respiration of limnic phytoplankton |
| `+ (p_cya_resp_nh4)*(rfr_p)` | respiration of diazotroph cyanobacteria |
| `+ (p_zoo_resp_nh4)*(rfr_p)` | respiration of zooplankton |
| `+ (p_det_resp_nh4)*(rfr_p)` | recycling of detritus using oxygen (respiration) |
| `+ (p_det_denit_nh4)*(rfr_p)` | recycling of detritus using nitrate (denitrification) |
| `+ (p_det_sulf_nh4)*(rfr_p)` | recycling of detritus using sulfate (sulfate reduction) |
| `+ (p_sed_resp_nh4)*(rfr_p)` `/(cgt_cellheight*cgt_density)` | recycling of sedimentary detritus to ammonium using oxygen (respiration) |
| `+ (p_sed_denit_nh4)*(rfr_p)` `/(cgt_cellheight*cgt_density)` | recycling of sedimentary detritus to ammonium using nitrate (denitrification) |
| `+ (p_sed_sulf_nh4)*(rfr_p)` `/(cgt_cellheight*cgt_density)` | recycling of sedimentary detritus to ammonium using sulfate (sulfate reduction) |
| `+` `p_ips_liber_po4/(cgt_cellheight*` `cgt_density)` | liberation of phosphate from the sediment under anoxic conditions |
| `+` `p_sed_pocp_resp/(cgt_cellheight*` `cgt_density)` | recycling of sedimentary pocp to dic and PO4 using oxygen (respiration) |

---

---





| Tracer equations, continued from previous page | |
|---|---|
| + p_sed_pocp_denit/(cgt_cellheight*cgt_density) | recycling of sedimentary pocp to dic and PO4 using nitrate (denitrification) |
| + p_sed_pocp_sulf/(cgt_cellheight*cgt_density) | recycling of sedimentary pocp to dic and PO4 using sulfate (sulfate reduction) |
| + p_dop_resp | respiration of DOP |
| + p_dop_denit | recycling of DOP using nitrate (denitrification) |
| + p_dop_sulf | Mineralization of DOP, e-acceptor sulfate (sulfate reduction) |
| - (p_no3_assim_lpp)*(rfr_p) | assimilation of nitrate by large-cell phytoplankton |
| - (p_nh4_assim_lpp)*(rfr_p) | assimilation of ammonium by large-cell phytoplankton |
| - (p_no3_assim_spp)*(rfr_p) | assimilation of nitrate by small-cell phytoplankton |
| - (p_nh4_assim_spp)*(rfr_p) | assimilation of ammonium by small-cell phytoplankton |
| - (p_nh4_assim_lip)*(rfr_p) | assimilation of ammonium by limnic phytoplankton |
| - (p_no3_assim_lip)*(rfr_p) | assimilation of nitrate by limnic phytoplankton |
| - (p_n2_assim_cya)*(rfr_p) | fixation of dinitrogen by diazotroph cyanobacteria |
| - p_assim_lpp_dop | Production of DOP by LPP |
|  | |





---

Tracer equations, continued from previous page

---

| | |
|---|---|
| – `p_assim_spp_dop` | Production of DOP by SPP |
| – `p_assim_lip_dop` | Production of DOP by LIP |
| – `(p_po4_retent_ips)*(rfr_p)` `/(cgt_cellheight*cgt_density)` | retention of phosphate in the sediment under oxic conditions |

**Change of: small-cell phytoplankton**

$\frac{d}{dt}$ `t_spp =`

| | |
|---|---|
| + `p_no3_assim_spp` | assimilation of nitrate by small-cell phytoplankton |
| + `p_nh4_assim_spp` | assimilation of ammonium by small-cell phytoplankton |
| – `p_spp_graz_zoo` | grazing of zooplankton eating small-cell phytoplankton |
| – `p_spp_resp_nh4` | respiration of small-cell phytoplankton |
| – `p_spp_mort_det` | mortality of small-scale phytoplankton |

**Change of: zooplankton**

$\frac{d}{dt}$ `t_zoo =`

| | |
|---|---|
| + `p_lpp_graz_zoo` | grazing of zooplankton eating large-cell phytoplankton |
| + `p_spp_graz_zoo` | grazing of zooplankton eating small-cell phytoplankton |

---

---





---

| Tracer equations, continued from previous page | |
|---|---|
| + p_cya_graz_zoo | grazing of zooplankton eating diazotroph cyanobacteria |
| + p_lip_graz_zoo | grazing of zooplankton eating limnic phytoplankton |
| − p_zoo_resp_nh4 | respiration of zooplankton |
| − p_zoo_mort_det | mortality of zooplankton |

**Change of: hydrogen sulfide**

$\frac{d}{dt}$ t_h2s =

| | |
|---|---|
| + (p_poc_sulf)*(0.5) | Mineralization of POC, e-acceptor sulfate (sulfate reduction) |
| + (p_pocp_sulf)*(53) | Mineralization of POC, e-acceptor sulfate (sulfate reduction) |
| + (p_pocn_sulf)*(3.3125) | Mineralization of POCN, e-acceptor sulfate (sulfate reduction) |
| + (p_det_sulf_nh4)*(3.3125) | recycling of detritus using sulfate (sulfate reduction) |
| + (p_sed_sulf_nh4)*(3.3125) /(cgt_cellheight*cgt_density) | recycling of sedimentary detritus to ammonium using sulfate (sulfate reduction) |
| + (p_sed_poc_sulf)*(0.5) /(cgt_cellheight*cgt_density) | recycling of sedimentary poc to dic using sulfate (sulfate reduction) |
| + (p_sed_pocn_sulf)*(3.3125) /(cgt_cellheight*cgt_density) | recycling of sedimentary pocn to dic and NH4 using sulfate (sulfate reduction) |

---





---

Tracer equations, continued from previous page

| | |
|---|---|
| + (p_sed_pocp_sulf)*(53) /(cgt_cellheight*cgt_density) | recycling of sedimentary pocp to dic and PO4 using sulfate (sulfate reduction) |
| + (p_doc_sulf)*(0.5) | Mineralization of DOC, e-acceptor sulfate (sulfate reduction) |
| + (p_dop_sulf)*(53) | Mineralization of DOP, e-acceptor sulfate (sulfate reduction) |
| + (p_don_sulf)*(3.3125) | Mineralization of DON, e-acceptor sulfate (sulfate reduction) |
| - p_h2s_oxo2_sul | oxidation of hydrogen sulfide with oxygen |
| - p_h2s_oxno3_sul | oxidation of hydrogen sulfide with nitrate |

**Change of: sulfur**

$\frac{d}{dt}$ t_sul =

| | |
|---|---|
| + p_h2s_oxo2_sul | oxidation of hydrogen sulfide with oxygen |
| + p_h2s_oxno3_sul | oxidation of hydrogen sulfide with nitrate |
| - p_sul_oxo2_so4 | oxidation of elemental sulfur with oxygen |
| - p_sul_oxno3_so4 | oxidation of elemental sulfur with nitrate |

**Change of: total alkalinity**

$\frac{d}{dt}$ t_alk =

| | |
|---|---|
| + (1)*(p_pocn_resp)*(0.5) | respiration of POCN (produces ohminus) |

---



| Tracer equations, continued from previous page | |
|---|---|
| `+ (1)*(p_pocn_denit)*(0.5)` | recycling of POCN using nitrate (denitrification) (produces ohminus) |
| `+ (1)*(p_pocn_sulf)*(0.5)` | Mineralization of POCN, e-acceptor sulfate (sulfate reduction) (produces ohminus) |
| `+ (1)*(p_sed_pocn_resp)*(0.5)` `/(cgt_cellheight*cgt_density)` | recycling of sedimentary pocn to dic and NH4 using oxygen (respiration) (produces ohminus) |
| `+ (1)*(p_sed_pocn_denit)*(0.5)` `/(cgt_cellheight*cgt_density)` | recycling of sedimentary pocn to dic and NH4 using nitrate (denitrification) (produces ohminus) |
| `+ (1)*(p_sed_pocn_sulf)*(0.5)` `/(cgt_cellheight*cgt_density)` | recycling of sedimentary pocn to dic and NH4 using sulfate (sulfate reduction) (produces ohminus) |
| `+ (1)*(p_don_resp)*(0.5)` | respiration of DON (produces ohminus) |
| `+ (1)*(p_don_denit)*(0.5)` | recycling of DON using nitrate (denitrification) (produces ohminus) |
| `+ (1)*(p_don_sulf)*(0.5)` | Mineralization of DON, e-acceptor sulfate (sulfate reduction) (produces ohminus) |
| `- (1)*(p_nh4_assim_lpp_don)` | Production of DON by LPP (consumes ohminus) |
| `- (1)*(p_nh4_assim_spp_don)` | Production of DON by SPP (consumes ohminus) |
| `- (1)*(p_nh4_assim_lip_don)` | Production of DON by LIP (consumes ohminus) |
| `- (1)*(p_pocp_denit)*(3)` | recycling of POC using nitrate (denitrification) (consumes ohminus) |





| Tracer equations, continued from previous page | |
|---|---|
| – (1)*(p_pocp_sulf)*(3) | Mineralization of POC, e-acceptor sulfate (sulfate reduction) (consumes ohminus) |
| – (1)*(p_sed_pocp_denit)*(3) /(cgt_cellheight*cgt_density) | recycling of sedimentary pocp to dic and PO4 using nitrate (denitrification) (consumes ohminus) |
| – (1)*(p_sed_pocp_sulf)*(3) /(cgt_cellheight*cgt_density) | recycling of sedimentary pocp to dic and PO4 using sulfate (sulfate reduction) (consumes ohminus) |
| – (1)*(p_dop_denit)*(3) | recycling of DOP using nitrate (denitrification) (consumes ohminus) |
| – (1)*(p_dop_sulf)*(3) | Mineralization of DOP, e-acceptor sulfate (sulfate reduction) (consumes ohminus) |
| + (-1)*(p_nh4_assim_lpp)* (0.8125) | assimilation of ammonium by large-cell phytoplankton (produces h3oplus) |
| + (-1)*(p_nh4_assim_spp)* (0.8125) | assimilation of ammonium by small-cell phytoplankton (produces h3oplus) |
| + (-1)*(p_nh4_assim_lip)* (0.8125) | assimilation of ammonium by limnic phytoplankton (produces h3oplus) |
| + (-1)*(p_pocp_resp)*(3) | respiration of POCP (produces h3oplus) |
| + (-1)*(p_nh4_nit_no3)*(2) | nitrification (produces h3oplus) |
| + (-1)*(p_nh4_nitdenit_n2) /(cgt_cellheight*cgt_density) | coupled nitrification and denitrification after mineralization of detritus in oxic sediments (produces h3oplus) |





| Tracer equations, continued from previous page | |
|---|---|
| + (-1)*(p_sul_oxo2_so4)*(2) | oxidation of elemental sulfur with oxygen (produces h3oplus) |
| + (-1)*(p_sul_oxno3_so4)*(0.8) | oxidation of elemental sulfur with nitrate (produces h3oplus) |
| + (-1)*(p_sed_pocp_resp)*(3) /(cgt_cellheight*cgt_density) | recycling of sedimentary pocp to dic and PO4 using oxygen (respiration) (produces h3oplus) |
| + (-1)*(p_nh4_nitdenit_pocn_n2) /(cgt_cellheight*cgt_density) | coupled nitrification and denitrification after mineralization of pocn-detritus in oxic sediments (produces h3oplus) |
| + (-1)*(p_dop_resp)*(3) | respiration of DOP (produces h3oplus) |
| - (-1)*(p_no3_assim_lpp)* (1.1875) | assimilation of nitrate by large-cell phytoplankton (consumes h3oplus) |
| - (-1)*(p_no3_assim_spp)* (1.1875) | assimilation of nitrate by small-cell phytoplankton (consumes h3oplus) |
| - (-1)*(p_no3_assim_lip)* (1.1875) | assimilation of nitrate by limnic phytoplankton (consumes h3oplus) |
| - (-1)*(p_n2_assim_cya)*(0.1875) | fixation of dinitrogen by diazotroph cyanobacteria (consumes h3oplus) |
| - (-1)*(p_assim_lpp_dop)*(3) | Production of DOP by LPP (consumes h3oplus) |
| - (-1)*(p_assim_spp_dop)*(3) | Production of DOP by SPP (consumes h3oplus) |
| - (-1)*(p_assim_lip_dop)*(3) | Production of DOP by LIP (consumes h3oplus) |
| continued on next page... | |





| Tracer equations, continued from previous page | |
|---|---|
| - (-1)*(p_no3_assim_lpp_don) | Production of DON by LPP (consumes h3oplus) |
| - (-1)*(p_no3_assim_spp_don) | Production of DON by SPP (consumes h3oplus) |
| - (-1)*(p_no3_assim_lip_don) | Production of DON by LIP (consumes h3oplus) |
| - (-1)*(p_poc_denit)*(0.8) | recycling of POC using nitrate (denitrification) (consumes h3oplus) |
| - (-1)*(p_poc_sulf) | Mineralization of POC, e-acceptor sulfate (sulfate reduction) (consumes h3oplus) |
| - (-1)*(p_pocp_denit)*(84.8) | recycling of POC using nitrate (denitrification) (consumes h3oplus) |
| - (-1)*(p_pocp_sulf)*(106) | Mineralization of POC, e-acceptor sulfate (sulfate reduction) (consumes h3oplus) |
| - (-1)*(p_pocn_resp)*(0.5) | respiration of POCN (consumes h3oplus) |
| - (-1)*(p_pocn_denit)*(5.8) | recycling of POCN using nitrate (denitrification) (consumes h3oplus) |
| - (-1)*(p_pocn_sulf)*(7.125) | Mineralization of POCN, e-acceptor sulfate (sulfate reduction) (consumes h3oplus) |
| - (-1)*(p_lpp_resp_nh4)*(0.8125) | respiration of large-cell phytoplankton (consumes h3oplus) |
| - (-1)*(p_spp_resp_nh4)*(0.8125) | respiration of small-cell phytoplankton (consumes h3oplus) |





---

Tracer equations, continued from previous page

---

| | |
|---|---|
| - (-1)*(p_lip_resp_nh4)*(0.8125) | respiration of limnic phytoplankton (consumes h3oplus) |
| - (-1)*(p_cya_resp_nh4)*(0.8125) | respiration of diazotroph cyanobacteria (consumes h3oplus) |
| - (-1)*(p_zoo_resp_nh4)*(0.8125) | respiration of zooplankton (consumes h3oplus) |
| - (-1)*(p_det_resp_nh4)*(0.8125) | recycling of detritus using oxygen (respiration) (consumes h3oplus) |
| - (-1)*(p_det_denit_nh4)* (6.1125) | recycling of detritus using nitrate (denitrification) (consumes h3oplus) |
| - (-1)*(p_det_sulf_nh4)*(7.4375) | recycling of detritus using sulfate (sulfate reduction) (consumes h3oplus) |
| - (-1)*(p_sed_resp_nh4)*(0.8125) /(cgt_cellheight*cgt_density) | recycling of sedimentary detritus to ammonium using oxygen (respiration) (consumes h3oplus) |
| - (-1)*(p_sed_denit_nh4)* (6.1125)/(cgt_cellheight* cgt_density) | recycling of sedimentary detritus to ammonium using nitrate (denitrification) (consumes h3oplus) |
| - (-1)*(p_sed_sulf_nh4)*(7.4375) /(cgt_cellheight*cgt_density) | recycling of sedimentary detritus to ammonium using sulfate (sulfate reduction) (consumes h3oplus) |
| - (-1)*(p_sed_poc_denit)*(0.8) /(cgt_cellheight*cgt_density) | recycling of sedimentary poc to dic using nitrate (denitrification) (consumes h3oplus) |

---

---



| Tracer equations, continued from previous page | |
|---|---|
| - (-1)*(p_sed_poc_sulf) /(cgt_cellheight*cgt_density) | recycling of sedimentary poc to dic using sulfate (sulfate reduction) (consumes h3oplus) |
| - (-1)*(p_h2s_oxno3_sul)*(0.4) | oxidation of hydrogen sulfide with nitrate (consumes h3oplus) |
| - (-1)*(p_sed_pocn_resp)*(0.5) /(cgt_cellheight*cgt_density) | recycling of sedimentary pocn to dic and NH4 using oxygen (respiration) (consumes h3oplus) |
| - (-1)*(p_sed_pocn_denit)*(5.8) /(cgt_cellheight*cgt_density) | recycling of sedimentary pocn to dic and NH4 using nitrate (denitrification) (consumes h3oplus) |
| - (-1)*(p_sed_pocp_denit)*(84.8) /(cgt_cellheight*cgt_density) | recycling of sedimentary pocp to dic and PO4 using nitrate (denitrification) (consumes h3oplus) |
| - (-1)*(p_sed_pocn_sulf)*(7.125) /(cgt_cellheight*cgt_density) | recycling of sedimentary pocn to dic and NH4 using sulfate (sulfate reduction) (consumes h3oplus) |
| - (-1)*(p_sed_pocp_sulf)*(106) /(cgt_cellheight*cgt_density) | recycling of sedimentary pocp to dic and PO4 using sulfate (sulfate reduction) (consumes h3oplus) |
| - (-1)*(p_doc_denit)*(0.8) | recycling of DOC using nitrate (denitrification) (consumes h3oplus) |
| - (-1)*(p_doc_sulf) | Mineralization of DOC, e-acceptor sulfate (sulfate reduction) (consumes h3oplus) |
| - (-1)*(p_dop_denit)*(84.8) | recycling of DOP using nitrate (denitrification) (consumes h3oplus) |
| - (-1)*(p_dop_sulf)*(106) | Mineralization of DOP, e-acceptor sulfate (sulfate reduction) (consumes h3oplus) |
| continued on next page... | |





---

Tracer equations, continued from previous page

---

| | |
|---|---|
| − (-1)*(p_don_resp)*(0.5) | respiration of DON (consumes h3oplus) |
| − (-1)*(p_don_denit)*(5.8) | recycling of DON using nitrate (denitrification) (consumes h3oplus) |
| − (-1)*(p_don_sulf)*(7.125) | Mineralization of DON, e-acceptor sulfate (sulfate reduction) (consumes h3oplus) |
| + (2)*(p_pocp_resp) | respiration of POCP (produces t_po4) |
| + (2)*(p_pocp_denit) | recycling of POC using nitrate (denitrification) (produces t_po4) |
| + (2)*(p_pocp_sulf) | Mineralization of POC, e-acceptor sulfate (sulfate reduction) (produces t_po4) |
| + (2)*(p_lpp_resp_nh4)*(rfr_p) | respiration of large-cell phytoplankton (produces t_po4) |
| + (2)*(p_spp_resp_nh4)*(rfr_p) | respiration of small-cell phytoplankton (produces t_po4) |
| + (2)*(p_lip_resp_nh4)*(rfr_p) | respiration of limnic phytoplankton (produces t_po4) |
| + (2)*(p_cya_resp_nh4)*(rfr_p) | respiration of diazotroph cyanobacteria (produces t_po4) |
| + (2)*(p_zoo_resp_nh4)*(rfr_p) | respiration of zooplankton (produces t_po4) |

---

---





| Tracer equations, continued from previous page | |
|---|---|
| `+ (2)*(p_det_resp_nh4)*(rfr_p)` | recycling of detritus using oxygen (respiration) (produces t_po4) |
| `+ (2)*(p_det_denit_nh4)*(rfr_p)` | recycling of detritus using nitrate (denitrification) (produces t_po4) |
| `+ (2)*(p_det_sulf_nh4)*(rfr_p)` | recycling of detritus using sulfate (sulfate reduction) (produces t_po4) |
| `+ (2)*(p_sed_resp_nh4)*(rfr_p)` `/(cgt_cellheight*cgt_density)` | recycling of sedimentary detritus to ammonium using oxygen (respiration) (produces t_po4) |
| `+ (2)*(p_sed_denit_nh4)*(rfr_p)` `/(cgt_cellheight*cgt_density)` | recycling of sedimentary detritus to ammonium using nitrate (denitrification) (produces t_po4) |
| `+ (2)*(p_sed_sulf_nh4)*(rfr_p)` `/(cgt_cellheight*cgt_density)` | recycling of sedimentary detritus to ammonium using sulfate (sulfate reduction) (produces t_po4) |
| `+ (2)*(p_ips_liber_po4)` `/(cgt_cellheight*cgt_density)` | liberation of phosphate from the sediment under anoxic conditions (produces t_po4) |
| `+ (2)*(p_sed_pocp_resp)` `/(cgt_cellheight*cgt_density)` | recycling of sedimentary pocp to dic and PO4 using oxygen (respiration) (produces t_po4) |
| `+ (2)*(p_sed_pocp_denit)` `/(cgt_cellheight*cgt_density)` | recycling of sedimentary pocp to dic and PO4 using nitrate (denitrification) (produces t_po4) |
| `+ (2)*(p_sed_pocp_sulf)` `/(cgt_cellheight*cgt_density)` | recycling of sedimentary pocp to dic and PO4 using sulfate (sulfate reduction) (produces t_po4) |
| `+ (2)*(p_dop_resp)` | respiration of DOP (produces t_po4) |





| Tracer equations, continued from previous page | |
|---|---|
| + (2)*(p_dop_denit) | recycling of DOP using nitrate (denitrification) (produces t_po4) |
| + (2)*(p_dop_sulf) | Mineralization of DOP, e-acceptor sulfate (sulfate reduction) (produces t_po4) |
| − (2)*(p_no3_assim_lpp)*(rfr_p) | assimilation of nitrate by large-cell phytoplankton (consumes t_po4) |
| − (2)*(p_nh4_assim_lpp)*(rfr_p) | assimilation of ammonium by large-cell phytoplankton (consumes t_po4) |
| − (2)*(p_no3_assim_spp)*(rfr_p) | assimilation of nitrate by small-cell phytoplankton (consumes t_po4) |
| − (2)*(p_nh4_assim_spp)*(rfr_p) | assimilation of ammonium by small-cell phytoplankton (consumes t_po4) |
| − (2)*(p_nh4_assim_lip)*(rfr_p) | assimilation of ammonium by limnic phytoplankton (consumes t_po4) |
| − (2)*(p_no3_assim_lip)*(rfr_p) | assimilation of nitrate by limnic phytoplankton (consumes t_po4) |
| − (2)*(p_n2_assim_cya)*(rfr_p) | fixation of dinitrogen by diazotroph cyanobacteria (consumes t_po4) |
| − (2)*(p_assim_lpp_dop) | Production of DOP by LPP (consumes t_po4) |
| − (2)*(p_assim_spp_dop) | Production of DOP by SPP (consumes t_po4) |
| − (2)*(p_assim_lip_dop) | Production of DOP by LIP (consumes t_po4) |
| continued on next page... | |





---

Tracer equations, continued from previous page

---

| | |
|---|---|
| - (2)*(p_po4_retent_ips)*(rfr_p) /(cgt_cellheight*cgt_density) | retention of phosphate in the sediment under oxic conditions (consumes t_po4) |

**Change of: sediment detritus**

$\frac{d}{dt}$ t_sed =

| | |
|---|---|
| + p_det_sedi_sed | detritus sedimentation |
| - p_sed_resp_nh4 | recycling of sedimentary detritus to ammonium using oxygen (respiration) |
| - p_sed_denit_nh4 | recycling of sedimentary detritus to ammonium using nitrate (denitrification) |
| - p_sed_sulf_nh4 | recycling of sedimentary detritus to ammonium using sulfate (sulfate reduction) |
| - p_sed_ero_det | sedimentary detritus erosion |
| - p_sed_biores_det | bio resuspension of sedimentary detritus |
| - p_sed_burial | burial of detritus deeper than max_sed |

**Change of: iron phosphate in sediment**

$\frac{d}{dt}$ t_ips =

| | |
|---|---|
| + (p_po4_retent_ips)*(rfr_p) | retention of phosphate in the sediment under oxic conditions |
| + p_ipw_sedi_ips | sedimentation of iron PO4 |

---

continued on next page...

---





| Tracer equations, continued from previous page | |
|---|---|
| − p_ips_liber_po4 | liberation of phosphate from the sediment under anoxic conditions |
| − p_ips_ero_ipw | erosion of iron PO4 |
| − p_ips_biores_ipw | bio resuspension of iron PO4 |
| − p_ips_burial | burial of iron PO4 |

**Change of: limnic phytoplankton**

$\frac{d}{dt}$ t_lip =

| | |
|---|---|
| + p_nh4_assim_lip | assimilation of ammonium by limnic phytoplankton |
| + p_no3_assim_lip | assimilation of nitrate by limnic phytoplankton |
| − p_lip_graz_zoo | grazing of zooplankton eating limnic phytoplankton |
| − p_lip_resp_nh4 | respiration of limnic phytoplankton |
| − p_lip_mort_det | mortality of limnic phytoplankton |

**Change of: dissolved organic carbon**

$\frac{d}{dt}$ t_doc =

| | |
|---|---|
| + p_assim_lpp_doc | Production of DOC by LPP |
| + p_assim_spp_doc | Production of DOC by SPP |
| + p_assim_lip_doc | Production of DOC by LPP |
| + p_assim_cya_doc | Production of DOC by CYA |





---

Tracer equations, continued from previous page

---

| | | |
|---|---|---|
| | − `p_doc2pco` | particle formation from DOC |
| | − `p_doc_resp` | respiration of DOC |
| | − `p_doc_denit` | recycling of DOC using nitrate (denitrification) |
| | − `p_doc_sulf` | Mineralization of DOC, e-acceptor sulfate (sulfate reduction) |

**Change of: phosphorus in dissolved organic carbon in Redfield ratio**

$\frac{d}{dt}$ `t_dop` =

| | | |
|---|---|---|
| | + `p_assim_lpp_dop` | Production of DOP by LPP |
| | + `p_assim_spp_dop` | Production of DOP by SPP |
| | + `p_assim_lip_dop` | Production of DOP by LIP |
| | − `p_dop2pocp` | particle formation from DOP |
| | − `p_dop_resp` | respiration of DOP |
| | − `p_dop_denit` | recycling of DOP using nitrate (denitrification) |
| | − `p_dop_sulf` | Mineralization of DOP, e-acceptor sulfate (sulfate reduction) |

**Change of: nitrogen in dissolved organic carbon in Redfield ratio**

$\frac{d}{dt}$ `t_don` =

| | | |
|---|---|---|
| | + `p_nh4_assim_lpp_don` | Production of DON by LPP |

---

continued on next page...

---





---

Tracer equations, continued from previous page

---

+ `p_no3_assim_lpp_don`   Production of DON by LPP

+ `p_nh4_assim_spp_don`   Production of DON by SPP

+ `p_no3_assim_spp_don`   Production of DON by SPP

+ `p_nh4_assim_lip_don`   Production of DON by LIP

+ `p_no3_assim_lip_don`   Production of DON by LIP

+ `(p_lpp_resp_nh4)*(don_fraction)`   respiration of large-cell phytoplankton

+ `(p_spp_resp_nh4)*(don_fraction)`   respiration of small-cell phytoplankton

+ `(p_lip_resp_nh4)*(don_fraction)`   respiration of limnic phytoplankton

+ `(p_cya_resp_nh4)*(don_fraction)`   respiration of diazotroph cyanobacteria

+ `(p_zoo_resp_nh4)*(don_fraction)`   respiration of zooplankton

- `p_don2pocn`   particle formation from DON

- `p_don_resp`   respiration of DON

- `p_don_denit`   recycling of DON using nitrate (denitrification)

---





---

| Tracer equations, continued from previous page | |
|---|---|
| − p_don_sulf | Mineralization of DON, e-acceptor sulfate (sulfate reduction) |

**Change of: sediment particular carbon**

$\frac{d}{dt}$ t_sed_poc =

| | |
|---|---|
| + p_poc_sedi_sed | poc sedimentation |
| − p_sed_poc_resp | recycling of sedimentary poc to dic using oxygen (respiration) |
| − p_sed_poc_denit | recycling of sedimentary poc to dic using nitrate (denitrification) |
| − p_sed_poc_sulf | recycling of sedimentary poc to dic using sulfate (sulfate reduction) |
| − p_sed_ero_poc | sedimentary poc erosion |
| − p_sed_biores_poc | bio resuspension of sedimentary poc |
| − p_poc_burial | burial of poc deeper than max_sed |

**Change of: sediment particular organic N+C**

$\frac{d}{dt}$ t_sed_pocn =

| | |
|---|---|
| + p_pocn_sedi_sed | pocn sedimentation |
| − p_sed_ero_pocn | sedimentary pocn erosion |
| − p_sed_biores_pocn | bio resuspension of sedimentary pocn |

---



---

| Tracer equations, continued from previous page | |
|---|---|
| - `p_pocn_burial` | burial of pocn deeper than max_sed |
| - `p_sed_pocn_resp` | recycling of sedimentary pocn to dic and NH4 using oxygen (respiration) |
| - `p_sed_pocn_denit` | recycling of sedimentary pocn to dic and NH4 using nitrate (denitrification) |

**Change of: sediment particular organic P+C**

$\frac{d}{dt}$ `t_sed_pocp` =

| | |
|---|---|
| + `p_pocp_sedi_sed` | pocp sedimentation |
| - `p_sed_ero_pocp` | sedimentary pocp erosion |
| - `p_sed_biores_pocp` | bio resuspension of sedimentary pocp |
| - `p_pocp_burial` | burial of pocp deeper than max_sed |
| - `p_sed_pocp_resp` | recycling of sedimentary pocp to dic and PO4 using oxygen (respiration) |
| - `p_sed_pocp_denit` | recycling of sedimentary pocp to dic and PO4 using nitrate (denitrification) |

**Change of: colored dissolved organic carbon**

$\frac{d}{dt}$ `t_cdom` =

| | |
|---|---|
| - `p_cdom_decay` | decay of cdom due to light |

**Change of: large-cell phytoplankton**

---

---



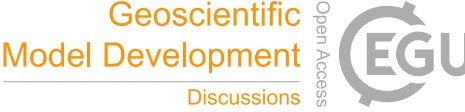

---

| Tracer equations, continued from previous page |
|---|

$\frac{d}{dt}$ `t_lpp =`

| | |
|---|---|
| + p_no3_assim_lpp | assimilation of nitrate by large-cell phytoplankton |
| + p_nh4_assim_lpp | assimilation of ammonium by large-cell phytoplankton |
| - p_lpp_graz_zoo | grazing of zooplankton eating large-cell phytoplankton |
| - p_lpp_resp_nh4 | respiration of large-cell phytoplankton |
| - p_lpp_mort_det | mortality of large-cell phytoplankton |

**Change of: suspended iron phosphate**

$\frac{d}{dt}$ `t_ipw =`

| | |
|---|---|
| + p_ips_ero_ipw/(cgt_cellheight* cgt_density) | erosion of iron PO4 |
| + p_ips_biores_ipw/(cgt_cellheight* cgt_density) | bio resuspension of iron PO4 |
| - p_ipw_sedi_ips/(cgt_cellheight* cgt_density) | sedimentation of iron PO4 |

**Change of: diazotroph cyanobacteria**

$\frac{d}{dt}$ `t_cya =`

| | |
|---|---|
| + p_n2_assim_cya | fixation of dinitrogen by diazotroph cyanobacteria |

---





---

| Tracer equations, continued from previous page | |
|---|---|
| – `p_cya_graz_zoo` | grazing of zooplankton eating diazotroph cyanobacteria |
| – `p_cya_resp_nh4` | respiration of diazotroph cyanobacteria |
| – `p_cya_mort_det` | mortality of diazotroph cyanobacteria |
| – `p_cya_mort_det_diff` | mortality of diazotroph cyanobacteria due to strong turbulence |

**Change of: detritus**

$\frac{d}{dt}$ `t_det` =

| | |
|---|---|
| + `p_lpp_mort_det` | mortality of large-cell phytoplankton |
| + `p_spp_mort_det` | mortality of small-scale phytoplankton |
| + `p_lip_mort_det` | mortality of limnic phytoplankton |
| + `p_cya_mort_det` | mortality of diazotroph cyanobacteria |
| + `p_cya_mort_det_diff` | mortality of diazotroph cyanobacteria due to strong turbulence |
| + `p_zoo_mort_det` | mortality of zooplankton |
| + `p_sed_ero_det/(cgt_cellheight* cgt_density)` | sedimentary detritus erosion |
| + `p_sed_biores_det/(cgt_cellheight* cgt_density)` | bio resuspension of sedimentary detritus |





---

Tracer equations, continued from previous page

---

| | |
|---|---|
| - `p_det_resp_nh4` | recycling of detritus using oxygen (respiration) |
| - `p_det_denit_nh4` | recycling of detritus using nitrate (denitrification) |
| - `p_det_sulf_nh4` | recycling of detritus using sulfate (sulfate reduction) |
| - `p_det_sedi_sed/(cgt_cellheight* cgt_density)` | detritus sedimentation |

**Change of: particulate organic carbon**

$\frac{d}{dt}$ `t_poc =`

| | |
|---|---|
| + `p_sed_ero_poc/(cgt_cellheight* cgt_density)` | sedimentary poc erosion |
| + `p_sed_biores_poc/(cgt_cellheight* cgt_density)` | bio resuspension of sedimentary poc |
| + `p_doc2pco` | particle formation from DOC |
| - `p_poc_resp` | respiration of POC |
| - `p_poc_denit` | recycling of POC using nitrate (denitrification) |
| - `p_poc_sulf` | Mineralization of POC, e-acceptor sulfate (sulfate reduction) |

---

---





---

Tracer equations, continued from previous page

|   |   |
|---|---|
| − p_poc_sedi_sed/(cgt_cellheight* cgt_density) | poc sedimentation |

**Change of: phosphorus in particulate organic carbon in Redfield ratio**

$\frac{d}{dt}$ t_pocp =

|   |   |
|---|---|
| + p_sed_ero_pocp/(cgt_cellheight* cgt_density) | sedimentary pocp erosion |
| + p_sed_biores_pocp/(cgt_cellheight cgt_density) | bio resuspension of sedimentary pocp |
| + p_dop2pocp | particle formation from DOP |
| − p_pocp_resp | respiration of POCP |
| − p_pocp_denit | recycling of POC using nitrate (denitrification) |
| − p_pocp_sulf | Mineralization of POC, e-acceptor sulfate (sulfate reduction) |
| − p_pocp_sedi_sed/(cgt_cellheight* cgt_density) | pocp sedimentation |
| − p_sed_pocp_sulf/(cgt_cellheight* cgt_density) | recycling of sedimentary pocp to dic and PO4 using sulfate (sulfate reduction) |

---





---

Tracer equations, continued from previous page

---

**Change of: nitrogen in particulate organic carbon in Redfield ratio**

$\frac{d}{dt}$ `t_pocn` =

| | |
|---|---|
| + `p_sed_ero_pocn/(cgt_cellheight* cgt_density)` | sedimentary pocn erosion |
| + `p_sed_biores_pocn/(cgt_cellheight* cgt_density)` | bio resuspension of sedimentary pocn |
| + `p_don2pocn` | particle formation from DON |
| − `p_pocn_resp` | respiration of POCN |
| − `p_pocn_denit` | recycling of POCN using nitrate (denitrification) |
| − `p_pocn_sulf` | Mineralization of POCN, e-acceptor sulfate (sulfate reduction) |
| − `p_pocn_sedi_sed/(cgt_cellheight* cgt_density)` | pocn sedimentation |
| − `p_sed_pocn_sulf/(cgt_cellheight* cgt_density)` | recycling of sedimentary pocn to dic and NH4 using sulfate (sulfate reduction) |

---

end of table **Tracer equations**

---



*Author contributions.* TN, HR, BC, and MS developed and implemented the model. TN performed the model simulations.
All authors contributed to writing the manuscript.

*Competing interests.* The authors declare that they have no conflict of interest.

*Acknowledgements.* Computational power was provided by the North-German Supercomputing Alliance (HLRN). Financial support by the BONUS INTEGRAL 363 project (Grant No. 03F0773A) is gratefully acknowledged. We wish to thank Bernd Schneider, and Henry Bitting for many advises and discussion. Gregor Rehder very much helped improving the manuscript
with many suggestions and careful revision.



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
