# Peer review of "Non-Redfield carbon model for the Baltic Sea (ERGOM version 1.2) – Implementation and Budget estimates"

_Geoscientific Model Development, 2022_

## Referee Comment (RC1)

**Peer review on Neumann et al. "Non-Redfield carbon model for the Baltic Sea (ERGOM version 1.2) – Implementation and Budget estimates." By Tatsuro Tanioka**

In this model description paper, the authors introduce non-Redfieldian C:N:P uptake stoichiometry into the biogeochemical model ERGOM v1.2 and study the model performance in the Baltic sea. By introducing a non-Redfieldian uptake ratio, the authors demonstrate that model estimates for pCO2, macronutrients, and oxygen improve compared to the previous version of ERGOM with strict Redfield C:N:P.

While I believe that the authors evaluated important issues for the modeling carbon cycle in the Baltic Sea, the study suffers from significant technical and model experimental design flaws. In particular, the assumption that cellular C:N:P of healthy phytoplankton is fixed at the Redfield ratio is not supported by either observation or previous modeling studies. Furthermore, almost identical issues in the Baltic Sea (underestimation of model pCO2) have been previously resolved by the model study (Fransner et al., 2018). It is not clear how the current study is a major improvement over the study by Fransner et al. and other numerous modeling studies (Kuznetsov et al., 2008; Kreus et al., 2015; Wan et al., 2011) that introduced non-Redfieldian stoichiometry to improve carbon cycle, nitrogen, phosphorus, and oxygen dynamics in the Baltic. In summary, this paper provides only an incremental advance in our understanding of biogeochemical cycling in the Baltic Sea, and I do not believe it is worthy of publication in GMD.

General Comments:
1. The assumption about fixed C/N/P of phytoplankton cells: the authors assume that "the stoichiometry of healthy phytoplankton cells follow the Redfield ratio." (line 45). However, this is likely not the case for phytoplankton in the Baltic, especially diazotrophs that thrive over summer following spring bloom (Larsson et al., 2001). C:N:P of suspended POM varies seasonally (Heiskanen et al., 1998). Previous non-Redfieldian models have considered these observations and described variable C/N/P of phytoplankton in the Baltic (Kuznetsov et al., 2008; Kreus et al., 2015; Wan et al., 2011). My enthusiasm for this study is dampened because the authors do not cite or show any evidence supporting their assumption that the C:N:P of phytoplankton in the Baltic is fixed at the Redfield ratio.
2. No rigorous assessment of how the model improved from the Redfield model: The authors do not fully quantitatively assess how incorporating non-Redfield C:N:P improves modeling nutrients and oxygen dynamics. In addition, a poor model fit of Alkalinity to observation by almost 200-300 uM raises concern about whether the carbon cycle model is correctly configured and parameters are tuned.

Other Comments:
1. L16: It may be more appropriate to call it "non-Redfieldian," not "non-Redfieldish" throughout the text.
2. L46: What is the justification that phytoplankton are healthy in the study area? Does healthy imply nutrient-replete? If so, please justify.

3. L47-48: The study by Martiny et al. (2013) clearly demonstrates significant deviation in C:N:P of suspended POM in the nutrient-deplete and nutrient-replete regions.
4. L56 "Strong observational evidence for evidence for an ER of DOM": Please provide ER evidence in Baltic.
5. Figure 3 and L 123-125: What exactly are these relationships between DIN/DIP and C:N:P in terms of equations? How are they derived?
6. Figure 4: Where are the heterotrophic bacteria that feed on DOM?
7. Figure 6: Please define what "organic matter" means? Is it DOM + POM or POM only?
8. L187-L188: "We have to note that our model approach does not allow for C:N and C:P ratio below Redfield ratios in the DOM and POM fractions." What is the justification for this? C:N and C:P in nutrient-rich regions such as the Southern Ocean are below Redfield ratios of 106 and 6.625 (Martiny et al., 2013).
9. Figure 13: It would be helpful to plot the Redfield case as a control.

Reference:

Fransner, F., Gustafsson, E., Tedesco, L., Vichi, M., Hordoir, R., Roquet, F., Spilling, K., Kuznetsov, I., Eilola, K., Mörth, C. M., Humborg, C., and Nycander, J.: Non-Redfieldian Dynamics Explain Seasonal pCO2Drawdown in the Gulf of Bothnia, J. Geophys. Res. Ocean., 12310th January, 166–188, https://doi.org/10.1002/2017JC013019, 2018.

Heiskanen, A.-S., Haapala, J., and Gundersen, K.: Sedimentation and Pelagic Retention of Particulate C, N and P in the Coastal Northern Baltic Sea, Estuar. Coast. Shelf Sci., 46, 703–712, https://doi.org/10.1006/ecss.1997.0320, 1998.

Kreus, M., Schartau, M., Engel, A., Nausch, M., and Voss, M.: Variations in the elemental ratio of organic matter in the central Baltic Sea: Part I—Linking primary production to remineralization, Cont. Shelf Res., 100, 25–45, https://doi.org/10.1016/j.csr.2014.06.015, 2015.

Kuznetsov, I., Neumann, T., and Burchard, H.: Model study on the ecosystem impact of a variable C:N:P ratio for cyanobacteria in the Baltic Proper, Ecol. Modell., 219, 107–114, https://doi.org/10.1016/j.ecolmodel.2008.08.002, 2008.

Larsson, U., Hajdu, S., Walve, J., and Elmgren, R.: Baltic Sea nitrogen fixation estimated from the summer increase in upper mixed layer total nitrogen, Limnol. Oceanogr., 46, 811–820, https://doi.org/10.4319/lo.2001.46.4.0811, 2001.

Martiny, A. C., Pham, C. T. A., Primeau, F. W., Vrugt, J. A., Moore, J. K., Levin, S. A., and Lomas, M. W.: Strong latitudinal patterns in the elemental ratios of marine plankton and organic matter, Nat. Geosci., 6, 279–283, https://doi.org/10.1038/ngeo1757, 2013.

Wan, Z., Jonasson, L., and Bi, H.: N/P ratio of nutrient uptake in the Baltic Sea, Ocean Sci., 7, 693–704, https://doi.org/10.5194/os-7-693-2011, 2011.

---

## Referee Comment (RC2)

**General comment:**
The manuscript proposed a non-Redfield carbon fixation in the biogeochemical model which improved significantly the carbon cycle in terms of *spCO2*. The advantage of this implantation is clearly demonstrated. It is quite meaningful for the general model development on the Baltic Sea basin scale, especially with focus on carbon dynamics under different forcing conditions. It has also a potential implementation in the central North Sea, where the simulated *spCO2* is easily overestimated with the Redfield carbon fixation (Prowe eat al., 2009 and Lorkowski et .,2012).

I recommend the publication of the manuscript after a minor revision based on the following comments.

**Specific comments:**
1. The introduction needs to be reorganized:
   The first paragraph put to the end of the introduction part
   The paragraph stating from line 45 move to the beginning of the intro, which is followed by the paragraph starting from line 20.

2. Line 47: (Martiny et ., 2016) change to Martiny et ., (2016)
3. Line 52: 'a variation between adapted…. a global scale' does this mean that different groups has different N:P ratio?
4. Line 53, state to stated
5. Line 73-81, 'A prominent…mixing occurs.' This is not model description, consider to move them to the introduction or discussion part.
6. Line 82, in situ conditions (Fig.1)
7. Line 82-84, 'The missing organic carbon …. concentrations'. This is not model description, consider to move them to the introduction or discussion part.
8. Line 84: 'decided to' deleted, 'extend' to 'extended'
9. Line 85: 'an' to 'a', 'not limited' change to 'allow extra'
10. Line 86: 'ultimately by' to 'beyond the part limited by'
11. Ling 101-102,'When nutreints……. biomass', Does that mean in this cast, equation 1 does not take place?
12. Line 103-104, 'if both N …. produced', how to define both N and P are limiting? Could you give an explicate definition of N limitation/P limitation and N,P limitation since those terms are pretty important throughout the paper.
13. Figure 2, for nutrient rich conditions, Does that mean only when min(ln,lp,IL)=IL in equation 1, the phytoplankton growth takes place?
14. Figure 3, So the production of DOP/DON/DOC can not happen at the same time? if so, how could that condition with N,P limitation happen (DOC production)? Or how to define N limitation/P limitation and N,P limitation? And DOC change to DOM (DOC)
15. Line 113: 'a doubling of rates within' to 'doubling of growth rate with'
16. Line 119: 'we divide the nutrient …carbon assimilation'. This is quite misleading. If the value is one as shown in Fig.3, which means only equation 1 takes place. We take P: C uptake ratio as example, it is 1 P*106/C=1. When goes to the extreme cases when there is no Nitrogen in the system and very high P concentrations (lp=1,ln=0), then equation 3 and 4 take place if I'm correct? If yes, then in equation 3, dDOP/dt=ro*PY*IL*IT (in P unit as shown in Table 1) in equation4, dDOC/dt=ro*PY*IL*IT (in C unit) then further dDOP/dt=dDOC/dt. we assume dDOP/dt=dDOC/dt=1, which means by producing 1 DOP, 1P and 106 C are assimilated, and by producing 1 DOC, only 1 DOC is assimilated, then the P:C uptake ratio is 1P*106/(106+1)==> it is not 0.5. Is my understanding wrong? could you please correct me?
17. Line 120: equations '1 to 4' change to '2 to 4'
18. Figure 3: 'Nutrients (N,P) to carbon uptake ratios', This is also quite misleading. When talking about N (or P) to C uptake ratio, one always tends to think about the comparison with the Redfield numbers. So can you give it a more accurate defination?
19. Line 125: transparent exopolymer particle (TEP) to TEP
20. Line 137, any reference?
21. Figure 4: What's the mean by f(R,N,P)

22. Figure 4: arrows: respiration→ no-Redfield assimilation and O2, why? What process is this standing for? I did not get it from the process description in appendix.
23. Figure 5: Could you give the colorbar for the bathymetry?
24. Table 2: Can you illuminate these region divisions in Figure 5?
25. Line 254-255: 'the closed budget … should be zero': In Fig.15a, how is the sum of fluxes calculated? is it sum(Riverine load+air->sea flux+transport from north sea+burial+ocean change+sediment change) or sum(Riverine load+air->sea flux+transport from north sea+burial)? If the former, then the term 'sum of fluxes' is rather misleading especially without any explicit expression (change of inventory is not flux). If the later, why it is necessarily zero? Is there any net gain due to anthropogenic influence?
26. Line 260: 'In the water column… 70s': I can not see an obvious increase of nutrient loads in the 1960s and 1970s (especially compared with 1980s). Instead, the increase of water column carbon inventory is more coincident with the change of transport from North Sea (Fig.15a, see for example the sharp increase in 1963-1965).
    Another questions, over the time 1963-1975, the total inventory (water+sediment) shows a constant increase, which means there should be a net carbon flux into the system, but why this is not reflected from the sum of fluxes in Fig.15c, which instead shows a constant slight decrease?
    Could you also re-plot the Fig.15a so that the fluxes are not shielded by the legends?
27. Line 264-265: 'The increasing sum…. source': I can not understand this sentence even though I agree an internal source. If the sum of fluxes (river and transport) is zero, while there are changes in inventory, it indicates an internal source. But as stated here, the increase of fluxes and inventory change is not necessary an internal source. Please re-write this sentence to give a clear argument about the implication of internal source, for instance, smaller riverine load compared with transport but a long-term constant inventory. Probably an additional plot of the sum of river and transport in Fig.16b, which facilitate the comparison with the yellow line.
28. Line 266: ??--> Appendix B4?
29. Line 270-271: denitrification…. Cyanobacteria→ (denitrification…. Cyanobacteria)
30. Figure 6: c) residual of the budget which can be attributed to alkalinity generation: How was this calculated? And is this a cumulated result?
31. Line 276:boundary fluxes: what do 'boundary fluxes' constitute of? likely redundant here, can it be deleted? the same for nitrogen budget.
32. Figure 17: b) legend of 'Denit Ocean' is wrong (line type)
             c) Suggest to make the line type (and legend) of 'Denit Ocean' identical with the other two plots (a and b)
33. Line 345: 'reasonable' to 'reasonably'

**References:**
Prowe, A.F., Thomas, H., Pätsch, J., Kühn, W., Bozec, Y., Schiettecatte, L.S., Borges, A.V. and de Baar, H.J., 2009. Mechanisms controlling the air–sea CO2 flux in the North Sea. *Continental Shelf Research*, *29*(15), pp.1801-1808.

Lorkowski, I., Pätsch, J., Moll, A. and Kühn, W., 2012. Interannual variability of carbon fluxes in the North Sea from 1970 to 2006–Competing effects of abiotic and biotic drivers on the gas-exchange of CO2. Estuarine, Coastal and Shelf Science, 100, pp.38-57.

---

## Author Comment (AC1)

First of all, we would like to thank two referees for the thorough reviews of our manuscript. We followed the suggestions and think that the manuscript has improved considerably.

In the following, we respond to the referee's remarks. Remarks are shown in blue and our response in black.

**Referee #1**

While I believe that the authors evaluated important issues for the modeling carbon cycle in the Baltic Sea, the study suffers from significant technical and model experimental design flaws. In particular, the assumption that cellular C:N:P of healthy phytoplankton is fixed at the Redfield ratio is not supported by either observation or previous modeling studies. Furthermore, almost identical issues in the Baltic Sea (underestimation of model pCO2) have been previously resolved by the model study (Fransner et al., 2018). It is not clear how the current study is a major improvement over the study by Fransner et al. and other numerous modeling studies (Kuznetsov et al., 2008; Kreus et al., 2015; Wan et al., 2011) that introduced non-Redfieldian stoichiometry to improve carbon cycle, nitrogen, phosphorus, and oxygen dynamics in the Baltic.

We disagree that the described changes of ERGOM are not a major step forward in the description of the C-cycle of the Baltic Sea. However, we are thankful for the comments, as it clearly shows that we fell short in explaining why the model set up was chosen, and what is different - and progress - in comparison to earlier approaches. In the revised version, we provide an additional section ("Differences to earlier Approaches") which is drafted below:

The study Fransner et al. (2018) is limited to the northern Baltic (Gulf of Bothnia). Therefore, it has not been shown that the model works reasonably for the whole Baltic Sea. The simulation period in Fransner et al. is 21 years which may be sufficient for the Gulf of Bothnia (GoB) but is much too short for the whole Baltic Sea. Radtke et al. (2012) show a residence time for phosphorus of more than 35 years.

Fransner et al. use both, non-Redfield phytoplankton biomass and extracellular release of DOC. They found that for the GoB "A substantial part of the fixed carbon is directly exuded as semilabile extracellular DOC" (26%-52%). In order to keep the model as simple as possible and with it the computational effort as low as possible, we decided to include extracellular release of carbon only. We see this fact as an advantage over Fransner et al., especially for ensemble simulations on climate scales. Unfortunately, the authors do not show any deep water properties like oxygen which may be impacted by the increased downward carbon flux.

For Kuznetsov et al. (2008), a follow up study Kuznetsov et al. (2011), demonstrate that non-Redfield biomass, at least during summer since cyanobacteria only are considered, is by far not sufficient to reproduce observed pCO2. We use this result also as an argument to focus on extracellular release.

The model used in Kreus et al. (2015) is applied at a station in the central Baltic Sea. Thus, it is not shown that the model gives reasonable results in a 3D environment. It uses a similar approach as in Fransner et al. with a flexible elemental ratio in phytoplankton and extracellular release of DOM. From our point of

view, it involves the same disadvantage concerning the computational effort and do not proof that cell quotas improve the carbon cycle dynamics.

Wan et al. (2011) changed N/P uptake and mineralization ratios but did not introduce a flexible elemental uptake ratio. This approach implies violation of mass conservation.

We think that the question to what extent the elemental ratio (C/N/P) in healthy phytoplankton cells can deviate from Redfield is not answered yet. There are some studies claiming that the elemental ratio remains close to Redfield which we refer to in the introduction. On the other hand, some studies report strong deviations in POM. However, in most studies no separation of living cells and other particles have been performed (e.g. Nausch et al., 2009). Therefore, we make the alternative proposal to realize a non-Redfield uptake by introducing extracellular release. The advantage is a simpler model with less computational demands.

General Comments:

1. The assumption about fixed C/N/P of phytoplankton cells: the authors assume that "the stoichiometry of healthy phytoplankton cells follow the Redfield ratio." (line 45). However, this is likely not the case for phytoplankton in the Baltic, especially diazotrophs that thrive over summer following spring bloom (Larsson et al., 2001).

This comment is also related to the above raised concern by the referee #1. We address the rationale for the way we chose our parameterization in an additional section "Rationale for model setup" in the revised version with the below drafted content:

We do not doubt flexibility in phytoplankton stoichiometry. However, from a modeler's point of view, we consider a fixed elemental ratio in phytoplankton as a reasonable simplification with the advantage of less model complexity. We proved this concept by the application for the Baltic Sea. Measurable state variables agree well with the model data. Especially for surface pCO2, we achieved a considerable improvement. Improvement of the carbon cycle mass balances was the main focus of our model development since it plays a vital role in the energy cascade of the marine ecosystem.

For this reason, we decided to transfer the intracellular deviation from a fixed elemental ratio into dissolved organic matter with a flexible ratio as extracellular release. We justify this assumption by the little effect of intracellular flexibility on the carbon uptake which is a focus of our model development. Furthermore, observations of C/N/P ratios which distinguish between living cells and POM are still missing in the Baltic Sea area. In the following, we review literature supporting our assumptions.

In Kuznetsov et al. (2008 and 2011), we applied the Larsson et al. findings for diazotrophs. However, these elemental ratios do not explain observed pCO2, although the C/P ratio in diazotrophs increase up to fourfold and an additional, artificial, in spring blooming, diazotrophs species was introduced.

Larsson et al. did their study with filamentous cyanobacteria. Filaments consist not only of vegetative cells but also of akinetes, heterocysts, and vacuoles which in sum not necessarily are composed according the Redfield ratio. Especially, vacuoles develop in a later state of the bloom and may explain

an increasing C:P ratio. These mechanisms are not explicitly formulated in our model and parametrized instead by extracellular release.

Nausch et al. (2009) show that the elemental C:P ratio (up to 400) is elevated especially in cyanobacteria (their Fig. 7) similar to Larsson et al. However, the C:P ratio (100-200) in POM at the same stations is much lower (same Fig.). Taking the high C:P ratio of cyanobacteria into account, the C:P ratio of the remaining POM is close to the Redfield ratio (~100). In their Tab. 2, C:N ratios in POM are given (7-9) which appear close to Redfield. The slight C enrichment in POM cannot explain observed pCO2 (see also Kuznetsov, 2011).

Kreus et al. (2015) introduce extracellular release and cell quota into their model and run it in 1D environment in the central Baltic Sea. Two experiments have been performed (a) variable quotas, and (b) fixed quotas. POC:PON ratios are virtually the same for both experiments while POC:POP ratios show a different seasonality. However, they conclude that fueling the summer cyanobacteria bloom controlling the carbon cycle and nitrogen dynamics is determined by DOM which is also part of our model. The shortcoming in the DIP cycle in experiment (b) of Kreus et al. has been solved with our approach. In summary, one can conclude that cell quotas do not have an impact on the nitrogen and carbon cycle (their Fig. 5).

2. No rigorous assessment of how the model improved from the Redfield model: The authors do not fully quantitatively assess how incorporating non-Redfield C:N:P improves modeling nutrients and oxygen dynamics.

We do not claim to improve nutrient and oxygen dynamics but the carbon dynamics, clearly shown in Figs. 1 and 14. For nutrients and oxygen, we show that the concentrations almost match with observations at selected stations.

In addition, a poor model fit of Alkalinity to observation by almost 200-300 uM raises concern about whether the carbon cycle model is correctly configured and parameters are tuned.

We can dispel the referee's doubts: Alkalinity is a largely conservative tracer and its concentration is the result of boundary fluxes and mixing of water masses. That is, there is no freedom for model tuning. The underestimated alkalinity concentration is a well-known issue in Baltic Sea models (see discussion and references in our manuscript) and hypotheses to additional alkalinity sources exist but are lacking a proof by observations so far.

Other Comments:

1. L16: It may be more appropriate to call it "non-Redfieldian," not "non-Redfieldish" throughout the text.

We followed the referee's suggestion.

2. L46: What is the justification that phytoplankton are healthy in the study area? Does healthy imply nutrient-replete? If so, please justify.

No. Healthy cells are able to divide (grow) if conditions allow.

3. L47-48: The study by Martiny et al. (2013) clearly demonstrates significant deviation in C:N:P of suspended POM in the nutrient-deplete and nutrient-replete regions.

Variations may be significant in a statistical sense but in our opinion moderate in terms of absolute values. Their Fig. 2 gives for POC:PON = ~5-9, and for POC:POP = ~80-150 neglecting the last 3 month (out of 32). The POC:POP increase at the end of the time series cannot be attributed to a phosphate limitation (their Fig. 1) as a quota model would suggest. Furthermore, the authors note in their discussion section the analyzed POM constitutes not solely of phytoplankton.

4. L56 "Strong observational evidence for evidence for an ER of DOM": Please provide ER evidence in Baltic.

Evidence is given e.g. in Hoikkala et al. (2015) and references therein, now in the list of references.

5. Figure 3 and L 123-125: What exactly are these relationships between DIN/DIP and C:N:P in terms of equations? How are they derived?

We use eq. 1-4 as stated in L 124. The paragraph has been re-phrased.

6. Figure 4: Where are the heterotrophic bacteria that feed on DOM?

Bacteria are not explicitly simulated but are parameterized by the degradation rates, as many models do. As already discussed above, we follow the principle to keep the model as simple as possible. This is necessary to handle large simulations in a 3D environment. Complexity of a model depends on the research question and if bacteria are in the focus, of course they have to be considered explicitly.

7. Figure 6: Please define what "organic matter" means? Is it DOM + POM or POM only?

It is DOM+POM. See line 189 in the revised version or 180 in the initial version.

8. L187-L188: "We have to note that our model approach does not allow for C:N and C:P ratio below Redfield ratios in the DOM and POM fractions." What is the justification for this? C:N and C:P in nutrient-rich regions such as the Southern Ocean are below Redfield ratios of 106 and 6.625 (Martiny et al., 2013).

This is a simplification of the nature in our model approach avoiding computationally expensive cell quotas. A justification is *inter alia* the moderate deviations shown in Martiny et al. (2013). The impact on the carbon cycle is negligible as demonstrated in e.g. Kreus et al. (2015).

9. Figure 13: It would be helpful to plot the Redfield case as a control.

Fig. 1

**Referee #2**

1. The introduction needs to be reorganized:

The first paragraph put to the end of the introduction part

This is a request by the editor. We have a first paper on ERGOM 1.2 and were asked to make a note in the very beginning that it is the same model with a focus on a different part of the model.

The paragraph stating from line 45 move to the beginning of the intro, which is followed by the paragraph starting from line 20.

We re-organized the introduction as the referee recommended.

2. Line 47: (Martiny et ., 2016) change to Martiny et ., (2016)

Done.

3. Line 52: 'a variation between adapted…. a global scale' does this mean that different groups has different N:P ratio?

Yes, this is how we understand the study. The four phytoplankton groups have fixed but slightly different elemental ratios. This setup gives best accordance with observed POM ratios due to different composition of the model groups.

4. Line 53, state to stated

We used recommendations in https://www.languageediting.com/verb-tenses-in-scientific-writing/.

Done.

5. Line 73-81, 'A prominent…mixing occurs.' This is not model description, consider to move them to the introduction or discussion part.

The referee is right, this is not pure model description. Thus, we integrated the paragraph in the introduction section.

6. Line 82, in situ conditions (Fig.1)

Done.

7. Line 82-84, 'The missing organic carbon …. concentrations'. This is not model description, consider to move them to the introduction or discussion part.

We like to thank the referee for this hint. Same as for #5.

8. Line 84: 'decided to' deleted, 'extend' to 'extended'

Done.

9. Line 85: 'an' to 'a', 'not limited' change to 'allow extra'

Done.

10. Line 86: 'ultimately by' to 'beyond the part limited by'

Done.

11. Ling 101-102,'When nutreints……. biomass', Does that mean in this cast, equation 1 does not take place?

That is true. Instead, eq. 2-4 become "active". However, there is a smooth transition from eq. 1 to eqs. 2-4 controlled by the limitation functions $l_N$ and $l_P$. See als our comment to #13.

12. Line 103-104, 'if both N …. produced', how to define both N and P are limiting? Could you give an explicate definition of N limitation/P limitation and N,P limitation since those terms are pretty important throughout the paper.

The referee is right. Our formulation was a bit sloppy. We re-phrased the sentence to: "If both N and P becoming exhausted, the fraction of produced DOC increases." In addition, we introduced a figure demonstrating the succession of production rates for phytoplankton, DON, DOP, and DOC. See also our comment to #14.

13. Figure 2, for nutrient rich conditions, Does that mean only when min(ln,lp,lL)=lL in equation 1, the phytoplankton growth takes place?

Nutrient limitations are formulated as a Michaelis-Menten or Monod kinetic. $l_N = N/(N_h + N)$. Depending on the half saturation constant $N_h$ and the ambient nutrient concentration N, growth rate gradually decreases with decreasing N.  According to Liebigs law, we use the minimum of all limitation functions. It gives a value between 0 and unity and is then multiplied with the maximum growth. If $l_N$ becomes lower, then $(1-l_N)$ in eq. 3 increases and DOP production starts. Back to your question, as long as $\min(l_N, l_P, l_L) > 0$, phytoplankton growth and the minimum value of the limitation functions control the uptake. However, simultaneous respiration and mortality decrease phytoplankton.

14. Figure 3, So the production of DOP/DON/DOC can not happen at the same time? if so, how could that condition with N,P limitation happen (DOC production)? Or how to define N limitation/P limitation and N,P limitation? And DOC change to DOM (DOC)

Production of phytoplankton, DOC, PON, and DON is simultaneous and controlled by the limitation functions. That is, the actual production rate of each OM compartment is variable and depends on nutrient concentrations. If e.g. N decreases DOP production increases (if P is sufficiently available) and phytoplankton growth decreases. We included an additional figure (4) demonstrating the production rates of PY, DON, DOP, and DOC.

15. Line 113: 'a doubling of rates within' to 'doubling of growth rate with'

Done.

16. Line 119: 'we divide the nutrient …carbon assimilation'. This is quite misleading. If the value is one as shown in Fig.3, which means only equation 1 takes place. We take P: C uptake ratio as example, it is 1 P*106/C=1. When goes to the extreme cases when there is no Nitrogen in the system and very high P concentrations (lp=1,ln=0), then equation 3 and 4 take place if I'm correct? If yes, then in equation 3, dDOP/dt=ro*PY*lL*lT (in P unit as shown in Table 1) in equation4, dDOC/dt=ro*PY*lL*lT (in C unit) then further dDOP/dt=dDOC/dt. we assume dDOP/dt=dDOC/dt=1, which means by producing 1 DOP, 1P and 106 C are assimilated, and by producing 1 DOC, only 1 DOC is assimilated, then the P:C uptake ratio is 1P*106/(106+1)==> it is not 0.5. Is my understanding wrong? could you please correct me?

Eqs. 1-4 show relative rates, which are independent on OM stoichiometry. Building biomass (OM), the elemental composition has to be considered. That is, P, N, and C are taken up with different speeds. This is part of the uptake processes in the model as a factor.

Assuming, the model "currency" is phosphorus and 1 mol P is taken up for OM building, then e.g. 16 mol N and 106 mol C are take up as well. In the example you raised, where d/dt(DOP)=d/dt(DOC)=r0, DOP will increase by r0 moles P and 106*r0 moles C per time unit while DOC will increase by 106*r0 mol C. This will give for the uptake ratio: r0*P(DOP)/(r0*106C(DOP) + r0*106C(DOC)) = 1P/212C, and normalized by 1P/106C yields 0.5. We get the same result from the relative rates. Therefore, we will stick to the relative rates, normalized with the uptake stoichiometry.

17. Line 120: equations '1 to 4' change to '2 to 4'

Done.

18. Figure 3: 'Nutrients (N,P) to carbon uptake ratios', This is also quite misleading. When talking about N (or P) to C uptake ratio, one always tends to think about the comparison with the Redfield numbers. So can you give it a more accurate defination?

We think all information is given in the figure's caption. Nutrient concentrations are normalized with their half saturation constant, that is, a concentration of one gives a limitation of 0.5. The N/C and P/C ratios are normalized by the classical Redfield numbers 16/106 and 1/106, respectively. Thus, a ratio of one refers to the Redfield ratio.

19. Line 125: transparent exopolymer particle (TEP) to TEP

Changed

20. Line 137, any reference?

There is no reference for our model because we introduced it in the presented development step.

21. Figure 4: What's the mean by f(R,N,P)

We are thankful for the hint. It should be f(N,P). We modified the figure.

22. Figure 4: arrows: respiration🞂 no-Redfield assimilation and O2, why? What process is this standing for? I did not get it from the process description in appendix.

Many thanks for discovering this bug. The arrow should end in the nutrient pool and of course oxygen or other electron acceptors are used. However, for clarity of the schematic, we decided not to draw arrows for all oxygen demanding processes. We made a note in the figure's caption.

23. Figure 5: Could you give the colorbar for the bathymetry?

Done.

24. Table 2: Can you illuminate these region divisions in Figure 5?

Done.

25. Line 254-255: 'the closed budget … should be zero': In Fig.15a, how is the sum of fluxes calculated? is it sum(Riverine load+air->sea flux+transport from north sea+burial+ocean change+sediment change) or sum(Riverine load+air->sea flux+transport from north sea+burial)? If the former, then the term 'sum of fluxes' is rather misleading especially without any explicit expression (change of inventory is not flux). If the later, why it is necessarily zero? Is there any net gain due to anthropogenic influence?

Thanks for the hint. Term "sum of fluxes" is too sloppy. We changed it into sum of boundary fluxes and inventory changes. We did the correction throughout the text.

26. Line 260: 'In the water column… 70s': I can not see an obvious increase of nutrient loads in the 1960s and 1970s (especially compared with 1980s). Instead, the increase of water column carbon inventory is more coincident with the change of transport from North Sea (Fig.15a, see for example the sharp increase in 1963-1965).

Loads are shown in Figs 18 and 19. They show a clear increase between 1960 until 1980. We refer now to the figures showing loads in the discussion of the increased carbon inventory. We think that nutrient loading is the main reason for carbon increase in the Baltic. The increase continues until about 1980 together with the loads and decreases with decreasing loads afterwards. However, it is still on a high level mirroring the eutrophic state.

Another questions, over the time 1963-1975, the total inventory (water+sediment) shows a constant increase, which means there should be a net carbon flux into the system, but why this is not reflected from the sum of fluxes in Fig.15c, which instead shows a constant slight decrease?

Thanks for pointing at this issue. It is because of our (sloppy) use of fluxes which lumps together boundary fluxes and inventory changes. This should be zero in a mass conserving system. We corrected the terms fluxes and inventory changes.

The inventory increase is due to higher carbon fixation initiated by increased nutrient loads. It becomes not obvious in the air-sea fluxes because changes in the inventory are very small compared to the air-sea fluxes.

Could you also re-plot the Fig.15a so that the fluxes are not shielded by the legends?

Done.

27. Line 264-265: 'The increasing sum…. source': I can not understand this sentence even though I agree an internal source. If the sum of fluxes (river and transport) is zero, while there are changes in inventory, it indicates an internal source. But as stated here, the increase of fluxes and inventory change is not necessary an internal source. Please re-write this sentence to give a clear argument about the implication of internal source, for instance, smaller riverine load compared with transport but a long-term constant inventory. Probably an additional plot of the sum of river and transport in Fig.16b, which facilitate the comparison with the yellow line.

We re-phrased the paragraph and modified figure 17.

28. Line 266: ??--> Appendix B4?

Done.

29. Line 270-271: denitrification…. Cyanobacteria (denitrification…. Cyanobacteria)

Done.

30. Figure 6: c) residual of the budget which can be attributed to alkalinity generation: How was this calculated? And is this a cumulated result?

Yes, it is a cumulated result, now noted in the figure. The alkalinity generation is calculated indirectly as the sum of boundary fluxes and inventory changes. In a mass conserving system it should be zero like it is for carbon. This deviation from zero, the residual, we attribute to alkalinity generation. We think this is better explained now by more carefully using the terms fluxes and inventory change.

31. Line 276:boundary fluxes: what do 'boundary fluxes' constitute of? likely redundant here, can it be deleted? the same for nitrogen budget.

We think "boundary fluxes" is the more precise term which is different from "internal fluxes" describing e.g. phosphorus fluxes between model state variables.

32. Figure 17: b) legend of 'Denit Ocean' is wrong (line type)

Corrected.

c) Suggest to make the line type (and legend) of 'Denit Ocean' identical with the other two plots (a and b)

Corrected

33. Line 345: 'reasonable' to 'reasonably'

Corrected.

**Technical evaluation**

A request from the technical evaluation was to ensure that the used color schemes allow readers with color vision deficiencies to correctly interpret figures. We use different line styles and checked the figures with the Color Blindness Simulator.

Literature:

L. Hoikkala, P. Kortelainen, H. Soinne, H. Kuosa, Dissolved organic matter in the Baltic Sea: Journal of Marine Systems, Volume 142, 2015, Pages 47-61, https://doi.org/10.1016/j.jmarsys.2014.10.005.

Kuznetsov, Ivan; Neumann, Thomas; Schneider, Bernd;  Yakushev, Evgeny: Processes regulating pCO2 in the surface waters of the central eastern Gotland Sea: a model study. Oceanologia, Volume 53, Issue 3, 2011, Pages 745-770, https://doi.org/10.5697/oc.53-3.745. (https://www.sciencedirect.com/science/article/pii/S0078323411500223)

Markus Kreus, Markus Schartau, Anja Engel, Monika Nausch, Maren Voss: Variations in the elemental ratio of organic matter in the central Baltic Sea: Part I—Linking primary production to remineralization, Continental Shelf Research, Volume 100, 2015, Pages 25-45, https://doi.org/10.1016/j.csr.2014.06.015.

Monika Nauscha, Günther Nausch, Hans Ulrich Lass, Volker Mohrholz, Klaus Nagel, Herbert Siegel, Norbert Wasmund: Phosphorus input by upwelling in the eastern Gotland Basin (Baltic Sea) in summer and its effects on filamentous cyanobacteria. Estuarine, Coastal and Shelf Science 83 (2009) 434–442, https://doi.org/10.1016/j.ecss.2009.04.031

Radtke, H., Neumann, T., Voss, M., and Fennel, W. (2012), Modeling pathways of riverine nitrogen and phosphorus in the Baltic Sea, J. Geophys. Res., 117, C09024, doi:10.1029/2012JC008119.

---

## Author Response (AR2)

IOW, Seestraße 15, 18119 Rostock

Dr. Thomas Neumann

Senior Scientist

Seestraße 15
D-18119 Rostock
phone: +49 381 51 97 130
fax: +49 381 51 97 440
www.io-warnemuende.de
thomas.neumann@
io-warnemuende.de

Rostock, 26.10.2022

Andrew Yool
Topical Editor
GMD

**Production files gmd-2022-79**

Dear Andrew Yool (GM editor)

Hereby, I submit the files needed for production of the manuscript gmd-2022-79

**Non-Redfield carbon model for the Baltic Sea (ERGOM version 1.2) -- Implementation and Budget estimates**

Thomas Neumann, Hagen Radtke, Bronwyn Cahill, Martin Schmidt, and Gregor Rehder

First of all, we would like to thank you and the referees again for the careful consideration of the manuscript. We updated the Zenodo repository. In particular, the data used for the figures were updated because a new figure was introduced in the revised version of the manuscript.

A request from the technical evaluation was to ensure that the used color schemes allow readers with color vision deficiencies to correctly interpret figures. We use different line styles and checked the figures with the Color Blindness Simulator.

With best regards, Thomas Neumann